# Evaluating one-loop string amplitudes

**Lorenz Eberhardt⋆ and Sebastian Mizera†**

Institute for Advanced Study, Einstein Drive, Princeton, NJ 08540, USA

⋆ elorenz@ias.edu ,   † smizera@ias.edu

## Abstract

We evaluate one-loop open-string amplitudes at finite $\alpha'$ for the first time. Our method involves a deformation of the integration contour over the modular parameter $\tau$ to a fractal contour introduced by Rademacher in the context of analytic number theory. This procedure leads to explicit and practical formulas for the one-loop four-point amplitudes in type-I superstring theory, amenable to numerical evaluation. We plot the amplitudes as a function of the Mandelstam invariants $s$ and $t$ and directly verify long-standing conjectures about their behaviour at high energies.

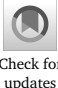

# 1 Introduction

Superstring perturbation theory instructs us to compute scattering amplitudes $\mathcal{A}_{g,n}$ as integrals of correlations functions of vertex operators over the moduli space $\mathcal{M}_{g,n}$ of genus-$g$ Riemann surfaces with $n$ punctures. Schematically, the formula encountered in textbooks on string theory is

$$\mathcal{A}_{g,n} \sim \int_{\mathcal{M}_{g,n}} \langle \mathcal{V}_1(z_1)\mathcal{V}_2(z_2)\cdots\mathcal{V}_n(z_n)\rangle \, \mathrm{d}\mu_{g,n}\,, \tag{1.1}$$

where $\mathcal{V}_i(z_i)$ are vertex operators inserted at positions $z_i$ and $\mathrm{d}\mu_{g,n}$ denotes the measure on $\mathcal{M}_{g,n}$ involving $z_i$'s and the surface moduli [1, 2]. Recall that the moduli space $\mathcal{M}_{g,n}$ is $(3g+n-3)$-dimensional, real or complex depending on whether we deal with open or closed strings respectively. It is well-known, albeit not often emphasized, that the above prescription is only approximately correct and (1.1) is ill-defined.

The problem with (1.1) has a physical origin. It can be traced back to the fact that the target space is Lorentzian, while the worldsheet theory is Euclidean. Of course, the reason to insist on a Euclidean worldsheet is so that we can use the powerful tools of two-dimensional CFTs and avoid spurious singularities that would come with a Lorentzian worldsheet [3]. The price we have to pay, however, is that certain ambiguities related to causal and unitary propagation of strings in space-time (which in quantum field theory are addressed by the Feynman $i\varepsilon$ prescription) remain unresolved. This can be already seen on explicit examples, such as $g = 1$ and $n = 4$, where (1.1) is purely real and hence cannot be consistent with the unitarity via the optical theorem. Recall that the question of unitarity in the target space is separate from that of unitarity of the worldsheet theory.

Witten proposed to cure this problem by zooming in on the boundaries of the moduli space $\mathcal{M}_{g,n}$ corresponding to Riemann surfaces degenerating to Feynman diagrams and resolve the aforementioned ambiguities by requiring consistency with the field-theory $i\varepsilon$ prescription [4]. Let us focus on the open-string case, where one can think of the open-string moduli space $\mathcal{M}_{g,n}$ as a contour embedded in its complexification $\mathcal{M}_{g,n}(\mathbb{C})$.[1] The task is then to prescribe an integration contour that coincides approximately with $\mathcal{M}_{g,n}$ on the bulk of the integration contour, but is otherwise designed to implement the Witten $i\varepsilon$ near the boundaries. The subject of this paper is a concrete realization of this idea.

Problems with the integration contour are somewhat milder after viewing string theory as an effective field theory and committing to the $\alpha'$-expansion. In fact, virtually all computations of string amplitudes are done this way, see, e.g., [5–12] for reviews. By contrast, in this work, we are interested in exploring intrinsically stringy properties of amplitudes and hence work at finite $\alpha'$, where we need to face the aforementioned difficulties.

The simplest case in which (1.1) needs to be corrected is already at genus zero (for any $n \geqslant 4$), but in a sense "anything goes" and the precise rerouting of the contour does not affect the final answer. This is related to fact that $\mathcal{A}_{0,n}$ is a tree-level amplitude and hence does not have any branch cuts. Likewise, at higher genus the part of the integration contour lying in the $z_i$ coordinates is easily fixable, but the part lying in the directions of the Riemann surface moduli needs additional work. The simplest interesting case is therefore $g = 1$ and $n = 4$, where $\mathcal{M}_{1,4}$ depends on the modular parameter $\tau$ in addition to the positions of the punctures. Describing and manipulating this contour will be the main results of this paper. We focus on the simplest case of open strings, including annulus and Möbius strip topologies.

Let us first recall the textbook definition of the integration contour used in (1.1) for the planar one-loop open-string amplitude. The integrand of (1.1) is known explicitly and will be

---

[1]Here, $\mathcal{M}_{g,n}(\mathbb{C})$ denotes the complexification of the open string moduli space. It is in general a cover of the corresponding closed string moduli space.

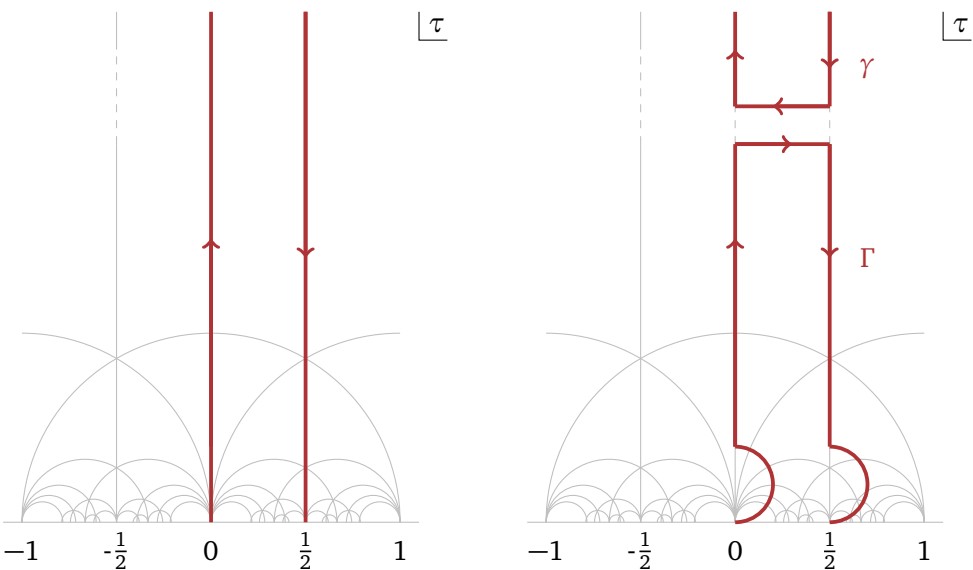

Figure 1: Contours of integration in the upper half-plane of the modular parameter $\tau$. **Left:** Textbook contour corresponding to integration over the annulus ($i\mathbb{R}$) and Möbius strip ($\frac{1}{2} + i\mathbb{R}$) topologies. The gray lines carving out the fundamental domain and its images are there only to guide the eye. **Right:** The integration contour consistent with causality and unitarity. The part approaching infinity can be evaluated exactly and integrating over $\Gamma$ is the main challenge addressed in this paper.

reviewed later in Section 2.1. As mentioned above, the most critical part of the contour lies in the $\tau$-plane, see Figure 1 (left). The integration over the annulus corresponds to the contour on the imaginary axis $i\mathbb{R}$ directed upwards. By itself, the integral over this contour is divergent as $\tau \to i\infty$, which corresponds to the annulus becoming thick and looking like a disk with a single closed-string emission at zero momentum. In type I superstring, this divergence is cured by the Möbius strip topology. It turns out that it can be described by exactly the same integrand as the annulus, except the contour needs to be shifted to $\frac{1}{2} + i\mathbb{R}$ and its orientation reversed. The two parts of the contour meet at infinity and cancel the divergence provided the gauge group is chosen to be SO(32).

The problems with the integration contour occur near $\tau = 0$ and $\tau = \frac{1}{2}$, where the annulus and the Möbius strip become very thin and look like Feynman diagrams. This is precisely the part of the contour that has to be appropriately modified. It turns out that the prescription consistent with the Witten $i\varepsilon$ is to choose the contour with two semi-circles illustrated in Figure 1 (right). Details behind this constructions will be given in Section 3.1. Since the integrand is holomorphic in the $\tau$ upper half-plane, the resulting contour can be freely deformed. We used this fact to split it into the part $\gamma$ enclosing the $i\infty$ and the rest which we call $\Gamma$. The former can be easily evaluated in terms of derivatives of the tree-level amplitude, so the main focus will be on $\Gamma$. Note that the size of the semi-circles in $\Gamma$ does not matter. A similar contour can be designed for the non-planar scattering amplitudes, see Section 3.1 for details.

We stress that this contour is *not* related by a contour deformation to the original contour. The reason is that the correlation function (the integrand) has essential singularities at every rational $\tau$ and hence the usual Cauchy contour deformation arguments do not apply there. Instead, $\gamma \cup \Gamma$ should be treated as a new proposal for the integration contour in the $\tau$-space. A more precise description on the whole complex moduli space $\mathcal{M}_{1,n}(\mathbb{C})$ will be given in [13]. As a matter of fact, the essential singularities are another way of determining that the contour on the left of Figure 1 could not have been the correct one: it simply gives a divergence close

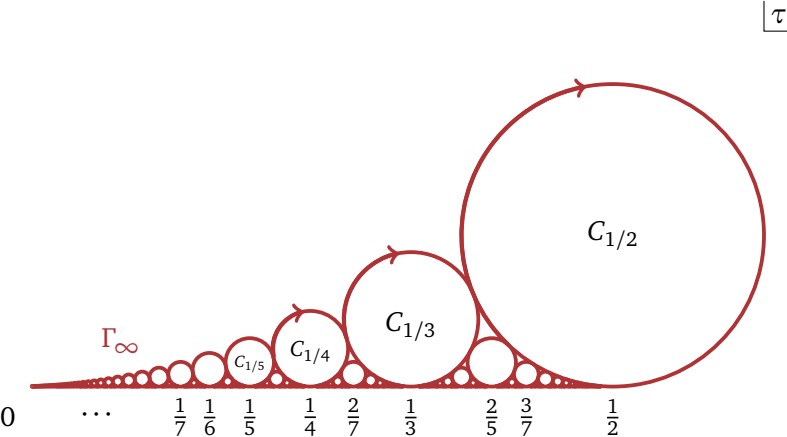

Figure 2: Rademacher contour $\Gamma_\infty$ in the $\tau$-plane is a sum of infinitely many Ford circles $C_{a/c}$ for all irreducible fractions $\frac{a}{c} \in (0, \frac{1}{2}]$. Each circle touches the real axis at $\tau = \frac{a}{c}$, has radius $\frac{1}{2c^2}$, and is oriented clockwise.

to the real axis because of the bad direction of approach at the singularities.

In a previous publication [14], we have already checked implicitly that the above contour is needed for consistency with unitarity at $n = 4$. More precisely, if $\Gamma$ corresponds to the choice of $+i\varepsilon$ prescription, the $-i\varepsilon$ version would be described by a similar contour with both half-circles bulging to the left. The imaginary part of the amplitude, with is proportional to the difference between the two choices, would be given by integrating over two circles anchored at $\tau = 0$ and $\tau = \frac{1}{2}$. After additional massaging, this contour gives rise to an explicit and convergent integral representation of the imaginary part $\mathrm{Im}\,\mathcal{A}_{1,4}$. We checked that this answer is in perfect agreement with computing the imaginary part using unitarity cuts. We refer the reader to [14] for more details.

Direct integration over the $n$-dimensional contour involving $\Gamma$ and the $z_i$ moduli using generalizations of the Pochhammer contour will be presented elsewhere [13]. For $n = 4$, this gives a 4-dimensional contour. But the bottom line of the above discussion of the imaginary part is that, in a sense, integrating over the circles anchored at any rational $\tau$ is already a solved problem and it leads to 2-dimensional integrals. Computing $\mathcal{A}_{1,n}$ can therefore be made more efficient if we managed to deform $\Gamma$ to a collection of circles.

A realization of this idea is inspired by the beautiful work of Rademacher [15,16], who employed a similar deformation to provide convergent infinite-sum representations of the Fourier coefficients of certain modular forms. Starting with $\Gamma$, an iterative process described in Section 3.2 allows us to deform it into the contour $\Gamma_\infty$ shown in Figure 2. We call it the *Rademacher contour*.[2] It consists of an infinite number of Ford circles $C_{a/c}$ touching the real axis at rational points $\tau = \frac{a}{c}$ for every such fraction between 0 and $\frac{1}{2}$. Each circle has a radius $\frac{1}{2c^2}$. The point is that each of them can be manipulated into a convergent integral expression. Roughly speaking, the smaller the circle, the smaller its contribution. Therefore, however insane the Rademacher contour might look like, it leads to an explicit formula for computing the amplitude.

The analysis becomes rather complicated due to the infinite number of branches of the integrand. After the dust settles, the final answer for the planar type I superstring amplitude

---

[2]Versions of the Rademacher contour appeared previously in many other areas of high-energy theory, see e.g. [17–20] for an incomplete cross-section.

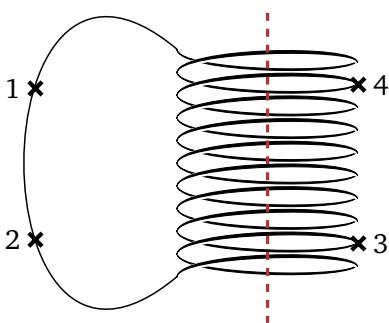

Figure 3: Example contribution to the sum (1.2) with $(n_{\text{L}}, n_{\text{D}}, n_{\text{R}}, n_{\text{U}}) = (0, 1, 7, 1)$ and hence $c = 10$. Time flows to the right and the cut (dashed line) separates the punctures with incoming (1 and 2) and outgoing (3 and 4) momenta. The four integers $(n_{\text{L}}, n_{\text{D}}, n_{\text{R}}, n_{\text{U}})$ count the number of extra windings across the cut between the punctures 1 and 2, 2 and 3, 3 and 4, 4 and 1, respectively.

$A^{\text{p}}$ takes the following form:

$$A^{\text{p}}(s,t) = \Delta A^{\text{p}}(s,t) + \sum_{\substack{\text{irreducible} \\ \text{fractions} \\ 0 < \frac{a}{c} \leqslant \frac{1}{2}}} \sum_{\substack{\text{windings} \\ n_{\text{L}}, n_{\text{D}}, n_{\text{R}}, n_{\text{U}} \geqslant 0 \\ n_{\text{L}} + n_{\text{D}} + n_{\text{R}} + n_{\text{U}} = c-1}} A_{a/c}^{n_{\text{L}}, n_{\text{D}}, n_{\text{R}}, n_{\text{U}}}(s,t). \tag{1.2}$$

The first contribution is the result of integrating over $\gamma$ and can be written explicitly in terms of derivatives of the Veneziano amplitude, see eq. (3.5). The second term involves a sum over all irreducible fractions $\frac{a}{c}$ between 0 and $\frac{1}{2}$, as dictated by the Rademacher contour $\Gamma_\infty$. For each of them, we sum over a finite number of integers $n_{\text{L}}$, $n_{\text{D}}$, $n_{\text{R}}$, $n_{\text{U}}$ that depend on how the four punctures are placed relative to one another. We can interpret these terms more physically as winding numbers in the following way.

Close to the real axis of $\tau$, Riemann surfaces become very skinny and look like worldlines. Consider the $s$-channel, in which the external particles 1 and 2 are incoming and hence are placed at past infinity, and likewise 3 and 4 are at future infinity since they are outgoing. We can separate them by an imagined space-like cut surface. Embedding the worldline in spacetime, the color lines of the annulus (near $\tau = \frac{0}{1}$) would go through the cut surface twice: on the way in and out. However, those of the Möbius strip (near $\tau = \frac{1}{2}$) would do it four times since it needs an extra winding. As a generalization, close to the point $\tau = \frac{a}{c}$, the color lines do exactly $c-1$ extra windings. Moreover, we can count how many extra windings occur between every pair of punctures, starting from the $(1, 2)$ and ending on $(4, 1)$. Let us call these numbers $(n_{\text{L}}, n_{\text{D}}, n_{\text{R}}, n_{\text{U}})$. They have to add up to $c-1$. An example is given in Figure 3. This gives an interpretation of every term in the sum (1.2).

The explicit expressions for $A_{a/c}^{n_{\text{L}}, n_{\text{D}}, n_{\text{R}}, n_{\text{U}}}$ are given in Section 3.3. In addition to the aforementioned winding numbers, they depend on the kinematic variables $(s, t)$ and we find they are proportional to $\frac{1}{c^5}$. Every such term is given by a convergent two-dimensional integral similar to the ones originating from the phase space integration for unitarity cuts. We argue that the whole sum in (1.2) convergent for $s > 0$, although the convergence is sufficiently fast for useful numerical evaluation when $s \gtrsim \frac{1}{\alpha'}$. For $s \to 0$, convergence breaks down. To understand the origin of this behavior, consider the toy model $A_{a/c}^{n_{\text{L}}, n_{\text{D}}, n_{\text{R}}, n_{\text{U}}}(s,t) = \frac{1}{c^5} e^{i\alpha' s\phi}$, where $\phi$ is a "random" phase that can depend on the winding numbers, $\frac{a}{c}$, and the kinematics. This is a good model for the actual formula for $A_{a/c}^{n_{\text{L}}, n_{\text{D}}, n_{\text{R}}, n_{\text{U}}}(s,t)$ that we derive in this paper, see eq. (3.25). As we take $\alpha' s \ll 1$, the phases stop mattering and the number of terms in the four-fold sum (over $a, n_{\text{L}}, n_{\text{D}}, n_{\text{R}}, n_{\text{U}}$ subject to one constraint) in (1.2) is $\mathcal{O}(c^4)$. Therefore,

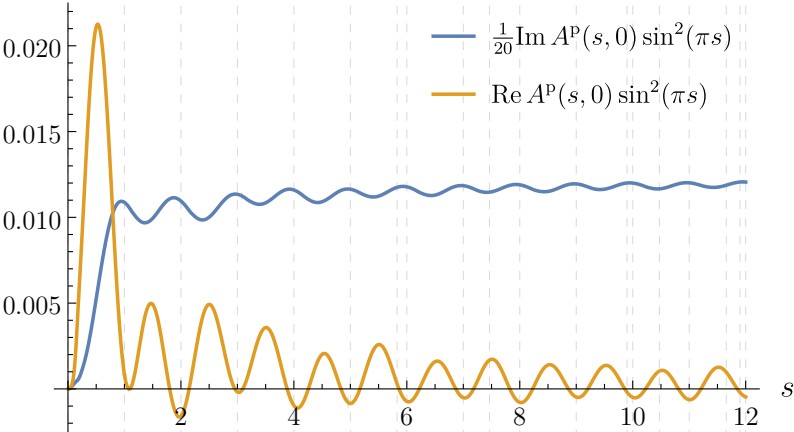

Figure 4: Planar open-string amplitude $A^{\mathrm{p}}(s,t)$ in the forward limit $t=0$, plotted as a function of $s$. For the purpose of the plot, the amplitude is multiplied by $\sin^2(\pi s)$ in order to remove double poles. The real part is given in orange and the imaginary part (rescaled by $\frac{1}{20}$) is in blue. Faint vertical lines indicate values of $s$ at which a new threshold opens up. The error bars from extrapolation $c \to \infty$ are smaller than the line widths.

in the strict $\alpha' = 0$ limit, we end up with a harmonic sum of the type $\sum_{c=1}^{\infty} \frac{1}{c}$, which indeed diverges. Increasing $s$ helps with convergence. For example, if the phases $\phi$ were sufficiently random (as they seem to be in practice), they would give $\mathcal{O}(\frac{1}{\sqrt{c}})$ enhancement to each sum, thus making the series converge pretty well. Indeed, we prove that (1.2) converges for every $s \in \mathbb{Z}_{>0}$. Analogous formulas can be derived in the non-planar case.

The formula (1.2) allows us to compute $A^{\mathrm{p}}(s,t)$ for given $s$ and $t$ and finally plot the amplitude. As an example, the results in the forward limit, $t=0$, are shown in Figure 4. We plotted $A^{\mathrm{p}}(s,0)\sin^2(\pi s)$ with the additional factor that removes double poles at every integer $s$ to make the plot readable. One can notice a few interesting features. The imaginary part dominates by around two orders of magnitude over the real part (in the figure, the former is multiplied by $\frac{1}{20}$ for readability). The imaginary part remains constant or slightly increasing, while the real part oscillates around zero. We cross-checked our results with the unitarity cut computation [14], explicit evaluation using the generalized Pochhammer contour [13], and computations of mass shifts, finding agreement in all cases. Details of the computations that went into Figure 4 are given in Section 3.5.

This interpretation in terms of winding numbers allows us to predict possible singularities of $A^{\mathrm{p}}$. For example, a pole in the $s$-channel can only happen when the punctures 1 and 2 (or 3 and 4) are brought together. But this can only happen when $n_{\mathrm{L}} = 0$ (or $n_{\mathrm{R}} = 0$). Hence if one is interested in averaged mass shifts and decay widths of strings, which can be read off from the coefficient of the double pole at integer $s$, the computation simplifies quite dramatically. We use this fact to compute mass shifts and decay widths up to $s \leqslant 16$, see Appendix D.

Moreover, we observe that mass shifts and decay widths provide a rough estimate for the average behaviour of the amplitude. As a practical application, we used them to compute the high-energy fixed-angle behavior. This limit was previously studied using saddle-point methods in [21] (see also [22, 23] for closed strings), who found evidence that the amplitude is suppressed as $\mathrm{e}^{-\alpha' S_{\mathrm{tree}}}$ as $\alpha' \to \infty$ with $s/t$ fixed, where $S_{\mathrm{tree}}$ is the tree-level on-shell action. But it is actually not possible to perform saddle-point analysis correctly without knowing the original integration contour, so the discussion of [21–23] should be viewed only as a heuristic. It is thus interesting to study how the high-energy behavior looks like in practice and now we have a great tool to do so.

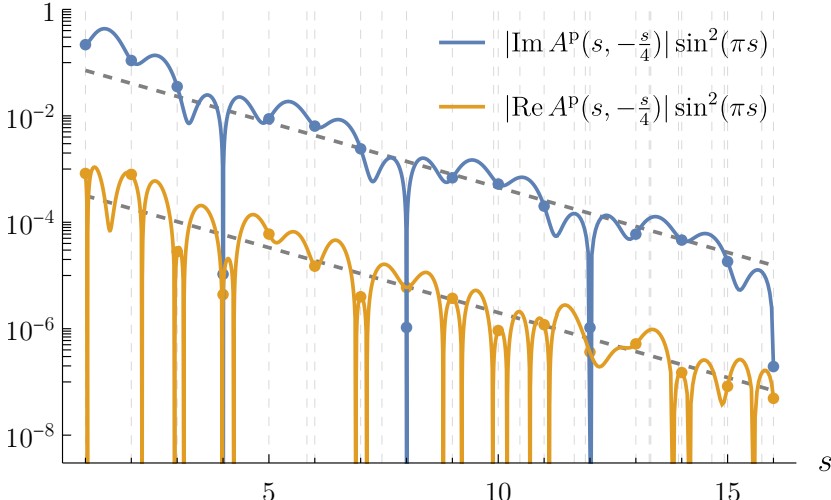

Figure 5: Exponential decay of the planar open-string amplitude in the high-energy fixed-angle limit. We plot $A^{\mathrm{p}}(s, -\frac{s}{4})\sin^2(\pi s)$ with the absolute values of real and imaginary parts in orange and blue respectively. The data for $s \leqslant 16$ is computed using (1.2) (with $c \leqslant 10$) and for all integer $s$ using mass shifts and decay widths (with $c \leqslant 1000$). Faint vertical lines indicate energies at which a new threshold opens up. The gray dashed lines correspond to exponential suppression $\mathrm{e}^{-S_{\mathrm{tree}}}$ with $S_{\mathrm{tree}} = s\log(s) + t\log(-t) + u\log(-u) \approx 0.56s$ and are plotted with two different constants to guide the eye. The data confirms exponential decay.

In Figure 5 we plot an example numerical evaluation of $A^{\mathrm{p}}(s, t)\sin^2(\pi s)$ at $60°$ scattering angle (translating to $t = -\frac{s}{4}$) for a range of energies $s$ on a logarithmic scale. The data spans roughly 8 orders of magnitude. We plot a continuous curve up to $s \leqslant 16$ and also high-precision values at all integer $s$ (since the plot is logarithmic, the spikes indicate zeros of the amplitude). The latter are computed using mass shifts and decay widths. The gray dashed lines indicate the exponential decay $\mathrm{e}^{-\alpha' S_{\mathrm{tree}}}$ and are in perfect agreement with the numerical data. We reaffirm this result by extracting the exponent of the exponential decay for a couple of scattering angles in Figure 6. Details of these computations are given in Section 3.5. For the imaginary part, this behavior was already verified in [14], where the coefficient of the exponential was also proposed, explaining departure from a pure exponential in Figure 5. Despite following the exponential envelope, the data has a jagged behavior, which is a result of receiving contributions from an infinite number of saddle points with the same exponential decay but different phases. We leave a more careful study of the high-energy behavior of the string amplitudes, where the integration contour $\Gamma$ is taken as a starting point, until future work.

A lot of physical aspects of open-string amplitudes, including computing the cross-section, dominance of low partial-wave spins, and low-energy expansions, were already discussed in [14], where we refer the reader for details.

This paper is organized as follows. In Section 2, we review the definitions of one-loop open string amplitudes and how Riemann surface degenerations encode different singularities of the amplitudes. We advise readers familiar with the subject to skip directly ahead to Section 3, where the main results of this paper are summarized in Section 3, including the proposal for the integration contour, the Rademacher procedure, and the numerical calculations. The curious reader then hopefully wants to know how to derive these results, which we explain in the following three Sections. In Section 4, we study the warm-up example of the two-point

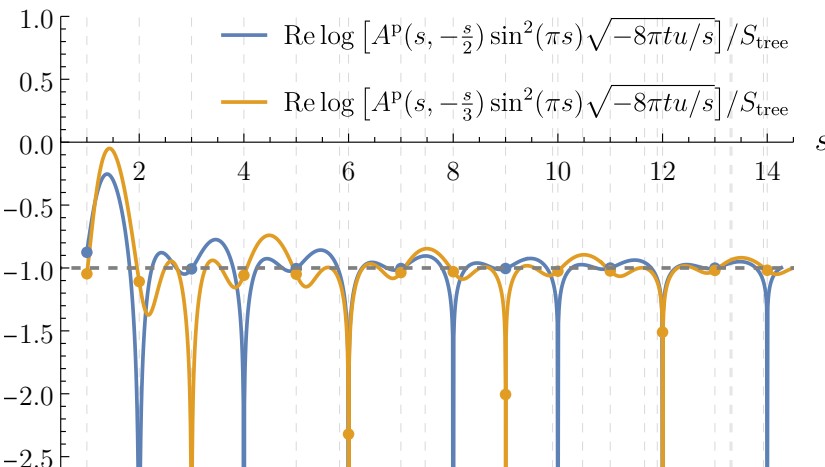

Figure 6: Coefficient of the exponential suppression for two different scattering angles, corresponding to setting $t = -\frac{s}{2}$ and $t = -\frac{s}{3}$. After normalizing the amplitude by $\sin^2(\pi s)\sqrt{-8\pi tu/s}$, we plot the real part of its exponent in the units of the tree-level action $S_{\text{tree}}$. The data for $s \lesssim 14.5$ is computed using (1.2) with $c \leqslant 10$ and for all integer $s$ with $c \leqslant 1000$ using mass shifts and decay widths. The gray dashed line indicates the exponential suppression $\mathrm{e}^{-S_{\text{tree}}}$ with which we find perfect agreement.

function, which illustrates most of the ideas employed in the full computation in a simplified setting. In Section 5, we give details of the manipulations leading to the formula (1.2) in the planar case and in Section 6 we treat the non-planar one. Computations of mass shifts and decay widths are given in Section 7. Finally, we conclude with a list of future directions in Section 8. This paper comes with a number of appendices. In Appendix A we review the computation of the cusp contribution to the planar amplitude. In Appendix B, we discuss convergence of the Rademacher method. In Appendix C we explain how to count the solutions to quadratic equations modulo prime powers, which is an ingredient in the computations of mass shifts. In Appendix D we tabulate the results for the numerical evaluation of mass shifts and decay widths.

## 2 Review: One-loop open string amplitudes

### 2.1 Annulus and Möbius strip topologies

In this work, we will consider one-loop open string four-point amplitudes in type I superstring theory. There are three possible diagrams for the scattering of four gluons that are depicted in Figure 7. The three basic cases are: the planar annulus diagram, where all four vertex operators are inserted on one boundary of the annulus, the (planar) Möbius strip diagram and the non-planar annulus diagram, where two vertex operators are one boundary and the other two are on the other boundary of the annulus. Of course, all these diagrams also exist with the roles of the punctures 1, 2, 3 and 4 permuted. We should immediately mention that the non-planar annulus diagram with one vertex operator on one boundary and all three others on the other boundary vanishes identically. The reason is that one has to trace over the Chan–Paton group factors and a single vertex operator insertion on one boundary leads to the factor $\mathrm{Tr}(t^a) = 0$, where $t^a$ are the adjoint generators for $\mathrm{SO}(N)$ with $N = 32$.

The color structure of the planar amplitudes is hence of the form $\mathrm{Tr}(t^{a_1} t^{a_2} t^{a_3} t^{a_4})$, whereas the color structure of the non-planar amplitudes is $\mathrm{Tr}(t^{a_1} t^{a_2})\mathrm{Tr}(t^{a_3} t^{a_4})$. Note that, relative

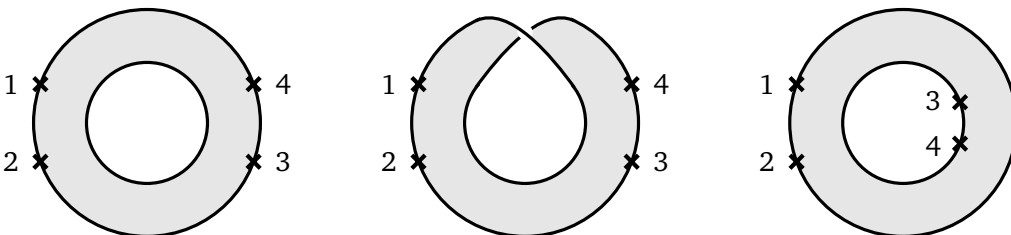

Figure 7: Three open string topologies at the one-loop level. **Left:** Planar annulus. **Middle:** Möbius strip. **Right:** Non-planar annulus.

to the Möbius strip, the annulus contribution has an additional factor of $N$ coming from the Chan–Paton trace $\mathrm{Tr}(\varnothing) = N$ on the empty boundary. The integrands for these open string amplitudes are well-known [24, 25]. Setting $\alpha' = 1$, they are given respectively by

$$\mathcal{A}_{\mathrm{an}}^{\mathrm{p}} = 2^9 \pi^2 g_{\mathrm{s}}^4 N \, t_8 \, \mathrm{Tr}(t^{a_1} t^{a_2} t^{a_3} t^{a_4}) \frac{(-i)}{32} \int_{\mathcal{M}_{1,4}^{\mathrm{p,an}}} \mathrm{d}\tau \, \mathrm{d}z_1 \, \mathrm{d}z_2 \, \mathrm{d}z_3 \prod_{j<i} \vartheta_1(z_{ij}, \tau)^{-s_{ij}} \,, \tag{2.1a}$$

$$\mathcal{A}_{\mathrm{M\ddot{o}b}} = 2^9 \pi^2 g_{\mathrm{s}}^4 \, t_8 \, \mathrm{Tr}(t^{a_1} t^{a_2} t^{a_3} t^{a_4}) \, i \int_{\mathcal{M}_{1,4}^{\mathrm{M\ddot{o}b}}} \mathrm{d}\tau \, \mathrm{d}z_1 \, \mathrm{d}z_2 \, \mathrm{d}z_3 \prod_{j<i} \vartheta_1(z_{ij}, \tau)^{-s_{ij}} \,, \tag{2.1b}$$

$$\mathcal{A}_{\mathrm{an}}^{\mathrm{n\text{-}p}} = 2^9 \pi^2 g_{\mathrm{s}}^4 \, t_8 \mathrm{Tr}(t^{a_1} t^{a_2}) \mathrm{Tr}(t^{a_3} t^{a_4}) \frac{(-i)}{32} \int_{\mathcal{M}_{1,4}^{\mathrm{n\text{-}p,an}}} \mathrm{d}\tau \, \mathrm{d}z_1 \, \mathrm{d}z_2 \, \mathrm{d}z_3 \prod_{j=1}^{2} \prod_{i=3}^{4} \vartheta_4(z_{ij}, \tau)^{-s_{ij}}$$
$$\times \left(\vartheta_1(z_{21}, \tau) \vartheta_1(z_{43}, \tau)\right)^{-s} \,. \tag{2.1c}$$

The planar amplitude is a sum of the first two contributions. The amplitudes depend on the Mandelstam invariants $s_{ij} = -(p_i + p_j)^2$, where $p_i$ denotes the external momentum associated to the vertex operator at position $z_i$. Only two kinematic invariants, say $(s, t)$ with $s = s_{12}$ and $t = s_{23}$ are independent. We use the mostly-minus signature in which, e.g., the $s$-channel kinematics is described by $-s < t < 0$. We denote $z_{ij} = z_i - z_j$. We use the following standard conventions for the Jacobi theta function

$$\vartheta_1(z, \tau) = i \sum_{n \in \mathbb{Z}} (-1)^n \mathrm{e}^{2\pi i (n - \frac{1}{2}) z + \pi i (n - \frac{1}{2})^2 \tau} \,, \tag{2.2a}$$

$$\vartheta_4(z, \tau) = \sum_{n \in \mathbb{Z}} (-1)^n \mathrm{e}^{2\pi i n z + \pi i n^2 \tau} \,. \tag{2.2b}$$

$g_{\mathrm{s}}$ is the string coupling constant. The prefactor $t_8$ depends on the polarizations and kinematics and will be spelled out below.

The moduli spaces $\mathcal{M}_{1,4}$ are real four-dimensional and consists of the purely imaginary modular parameter $\tau$ (for the Möbius strip we have $\tau \in \frac{1}{2} + i\mathbb{R}$) and purely real $z_i$, which are appropriately ordered, i.e., $0 \leqslant z_1 \leqslant z_2 \leqslant z_3 \leqslant z_4 = 1$ in the planar case and $-1 \leqslant z_1 \leqslant z_2 \leqslant 1$, $z_{21} \leqslant 1$, $0 \leqslant z_3 \leqslant z_4 = 1$ in the non-planar case. The U(1) isometry group of the annulus and the Möbius strip allows us to pick the location of one puncture, say $z_4$, arbitrarily and we will choose $z_4 = 1$ in the following. The factor of $-i$ in front of the expressions compensates for our choice to integrate over purely imaginary $\tau$ which supplies a further $i$ from the Jacobian. As written, the amplitudes are hence real. This immediately tells us that the above description of $\mathcal{M}_{1,4}$ cannot be quite correct: scattering amplitudes at loop level need imaginary parts for consistency with unitarity. We will temporarily ignore this issue and come back to it in Section 3.1, where we will define a prescription for the integration contour similar to the causal Feynman $i\varepsilon$ in quantum field theory.

Geometrically, these formulas arise as follows. For an open string worldsheet such as the annulus and the Möbius strip, there is always a corresponding closed surface that double covers the original open surface. The covering is branched along the boundaries of the Riemann surface. It is given in both cases by the torus. It can be constructed by taking the orientation double cover of the original surface and gluing the boundaries. For example, the orientation double cover of the Möbius strip is an annulus and the aforementioned torus is obtained by gluing its two boundaries. As an orientation double cover, the covering surface admits an orientation-reversing involution $\Phi$ such that the original surface is given by the respective quotient where one identifies $z \sim \Phi(z)$. In the case at hand, the torus is as usual realized by $\mathbb{T} = \mathbb{C}/\Lambda$ with $\Lambda = \langle 1, \tau \rangle$. Taking then $z \in \mathbb{T}$, the orientation-reversing map can be chosen as $\Phi(z) = \bar{z}$. For this to yield a well-defined map on the torus, we need $\Lambda = \bar{\Lambda}$. This is true in two distinct cases:

1. $\tau \in i\mathbb{R}_{>0}$. In this case, the resulting surface has two boundaries, namely the boundary corresponding to $z \in \mathbb{R} + \mathbb{Z}\,\mathrm{Im}(\tau)$ and the boundary corresponding to $z \in \mathbb{R} + (\mathbb{Z} + \frac{1}{2})\,\mathrm{Im}(\tau)$. The resulting geometry is an annulus.

2. $\tau \in \frac{1}{2} + i\mathbb{R}_{>0}$. In this case, there is only a single boundary given by the translates of the real line and we hence obtain a Möbius strip as the quotient surface.

In particular, vertex operators for the annulus can be either inserted on the real line, $z_j \in \mathbb{R}$, or on the line $z_j \in \mathbb{R} + \frac{1}{2}i\tau$. For the planar case, we will always choose the real line.

The close connection to the torus explains the appearance of Jacobi theta functions in (2.1). In fact, the Green's function on the torus is given by

$$G(z_i, z_j) = \log \left| \frac{\vartheta_1(z_{ij}, \tau)}{\vartheta_1'(\tau)} \right|^2 - \frac{2\pi [\mathrm{Im}(z_{ij})]^2}{\mathrm{Im}(\tau)}. \tag{2.3}$$

The non-holomorphic piece is necessary because of the constant function that is a zero mode of the Laplacian. Consequently, the Green's function satisfies

$$\Delta_{z_i} G(z_i, z_j) = 2\pi \delta^2(z_i - z_j) - \frac{2\pi}{\mathrm{Im}(\tau)}, \tag{2.4}$$

and the right-hand side has a vanishing integral over the torus. Now the free boson propagator on the quotient surface is simply given by

$$\frac{1}{2} G(z_i, z_j) = \begin{cases} \log \frac{\vartheta_1(z_{ji}, \tau)}{\vartheta_1'(\tau)}, & \text{if } z_i \text{ and } z_j \text{ are on the same boundary,} \\ \log \frac{\vartheta_4(z_{ji}, \tau)}{\vartheta_1'(\tau)}, & \text{if } z_i \text{ and } z_j \text{ are on different boundaries.} \end{cases} \tag{2.5}$$

This explains the various $\vartheta_i$-factors appearing in (2.1). It also explains why the planar annulus amplitude has exactly the same form as the Möbius strip amplitude, except that $\tau$ is shifted to $\tau + \frac{1}{2}$ in the latter one. The relative overall normalization between the two diagrams requires a more careful discussion, see [26].

These are the amplitudes for type I superstrings which are traditionally derived in the RNS formalism, although the pure-spinor superstring is actually more effective in deriving these formulas [27]. From the perspective of the RNS formalism, it is surprising that one ends up with a simple integral over the *bosonic* moduli space of Riemann surfaces. In general, super-string amplitudes involve integrals over supermoduli space: a supermanifold of dimension $3g - 3 + n|2g - 2 + n$ (for $n$ NS-punctures). Hence for a four-point function, there are four

fermionic integrals to be done. In general, there is no canonical way of performing integrals over the fermionic directions since there is no preferred choice of such "fermionic directions". Correspondingly, superstring theory usually does not give canonical integrands over the moduli space of Riemann surfaces.

At genus one, one is however in luck, since there is a very natural choice for how to do the fermionic integrals. In the traditional formalisms, fermionic integrals are performed by inserting picture-changing operators and on a genus-one surface we need exactly $n$ of them. Hence we can consider the correlation function of picture 0 NS-sector vertex operator, which leads to a well-defined integrand on the reduced space of supermoduli: the moduli space of spin-curves.[3] One then has to sum over the non-trivial spin-structures of the respective surfaces to finally reduce the amplitude to an integral over ordinary moduli space. Through various miraculous cancellations, one ends up with the simple integrands given in (2.1). In particular, the one-loop determinants of the worldsheet fields together with the additional contributions from the vertex operators (beyond the Green's function explained above) essentially cancel out in the end. They produce the coordinate-independent factor $t_8$. It is given by

$$
\begin{aligned}
t_8 = {} & \mathrm{tr}_\nu(F_1 F_2 F_3 F_4) + \mathrm{tr}_\nu(F_1 F_3 F_2 F_4) + \mathrm{tr}_\nu(F_1 F_2 F_4 F_3) \\
& - \tfrac{1}{4}\Big( \mathrm{tr}_\nu(F_1 F_2)\,\mathrm{tr}_\nu(F_3 F_4) + \mathrm{tr}_\nu(F_1 F_3)\,\mathrm{tr}_\nu(F_2 F_4) + \mathrm{tr}_\nu(F_1 F_4)\,\mathrm{tr}_\nu(F_2 F_3) \Big),
\end{aligned}
\tag{2.6}
$$

where the linearized field strengths are $F_i^{\mu\nu} = p_i^\mu \epsilon_i^\nu - \epsilon_i^\mu p_i^\nu$ with polarization vectors $\epsilon_i^\mu$ and the traces $\mathrm{tr}_\nu$ are taken over the Lorentz indices. It equals the Yang–Mills numerator and is the unique permutation-invariant structure consistent with gauge invariance and supersymmetry. Consequently, the four-gluon amplitude at any genus is guaranteed to have this universal factor present.

For convenience, we will strip off the prefactors from (2.1) that do not affect the analysis and denote the resulting amplitudes with non-curly symbols. In particular, after simplifying the integrand, the planar annulus amplitude is given by

$$
A_{\mathrm{an}}^{\mathrm{p}} = -i \int_{i\mathbb{R}_{\geqslant 0}} \mathrm{d}\tau \int_{0 \leqslant z_1 \leqslant z_2 \leqslant z_3 \leqslant 1} \mathrm{d}z_1\, \mathrm{d}z_2\, \mathrm{d}z_3 \left( \frac{\vartheta_1(z_{21},\tau)\vartheta_1(z_{43},\tau)}{\vartheta_1(z_{31},\tau)\vartheta_1(z_{42},\tau)} \right)^{-s} \left( \frac{\vartheta_1(z_{32},\tau)\vartheta_1(z_{41},\tau)}{\vartheta_1(z_{31},\tau)\vartheta_1(z_{42},\tau)} \right)^{-t},
\tag{2.7}
$$

and similarly the Möbius strip contribution $A_{\mathrm{Möb}}$ is obtained by replacing the $\tau$ integration with $\frac{1}{2} + i\mathbb{R}_{\geqslant 0}$ and multiplying by $-1$. The total planar amplitude is then

$$
A^{\mathrm{p}} = A_{\mathrm{an}}^{\mathrm{p}} + A_{\mathrm{Möb}}.
\tag{2.8}
$$

We also set $N = 32$, which is required for this combination to be well-defined. Finally, the non-planar amplitude is given by

$$
A^{\mathrm{n\text{-}p}} = \frac{-i}{32} \int_{i\mathbb{R}_{\geqslant 0}} \mathrm{d}\tau \int_{\substack{0 \leqslant z_1 \leqslant z_2 \leqslant 1 \\ 0 \leqslant z_3 \leqslant 1}} \mathrm{d}z_1\, \mathrm{d}z_2\, \mathrm{d}z_3 \prod_{j=1}^{2} \prod_{i=3}^{4} \vartheta_4(z_{ij},\tau)^{-s_{ij}} \big( \vartheta_1(z_{21},\tau)\vartheta_1(z_{43},\tau) \big)^{-s}.
\tag{2.9}
$$

All conventions agree with those used in [14].

## 2.2 Singularities of open strings

As it stands, the integrals in eq. (2.1) are all divergent. There are relatively benign divergences such as the collision of $z_1$ and $z_2$. We are already used to dealing with such singularities in the tree-level disk amplitude. They lead to the poles of the string amplitude corresponding

---

[3]Strictly speaking this distribution of picture changing operators (PCOs) is not entirely consistent, but it seems that one can get away with it at genus 1.

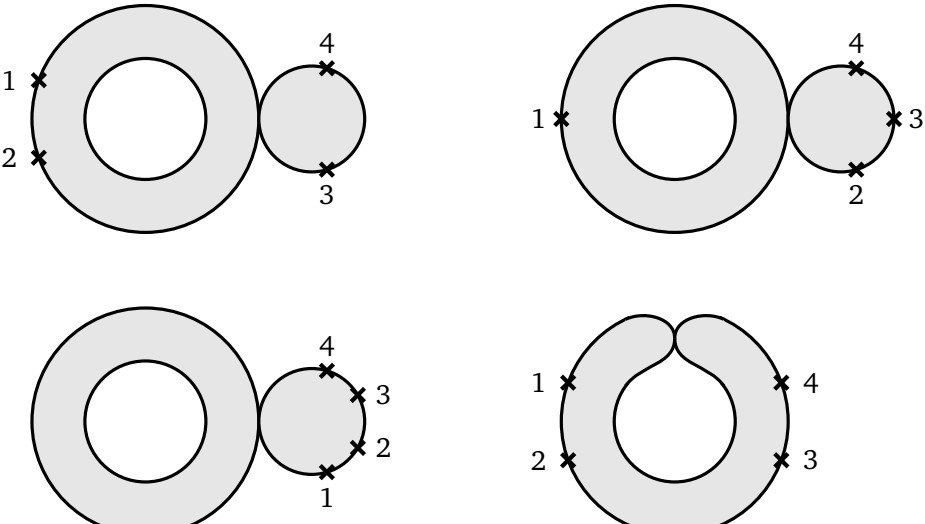

Figure 8: Four basic degenerations of the planar annulus amplitude. **Top left:** Massive pole exchange. **Top right:** Wave function renormalization. **Bottom left:** Tadpole. **Bottom right:** Non-separating degeneration.

to the exchange of massive string-modes, e.g., collision of $z_1$ and $z_2$ gives poles at $s \in \mathbb{Z}_{\geqslant 0}$. However, the story becomes more complicated at one-loop because the amplitude will also have discontinuities which are reflected in more intricate singular behaviours.

Before discussing them, we should recall some basic features of the moduli space of open Riemann surfaces. We will mostly just discuss the planar annulus case, since all other cases are similar. There are various degenerations of the surface that are familiar from the Deligne–Mumford compactification of the closed string moduli space. There is one additional type of boundary in the open string that appears when closing a hole without vertex operators, which we discuss below. The basic single degenerations of the planar annulus diagrams are depicted in Figure 8. Of course, these also exist with punctures relabelled in all ways such that the original permutation is preserved alonge the outer boundary.

Let us discuss the physical meaning and behaviour of the integrand near these degenerations in turn. Near degenerations, the worldsheet develops a very long "neck" connecting the two parts of the surface. This situation is conformally equivalent to a pinched cycle as drawn in Figure 8. Hence, near the degeneration, the string worldsheet collapses to a worldline and one makes contact with the effective field-theory description, at least for that part of the diagram. To fully reduce to field-theory amplitudes, one has to completely degenerate the surface, i.e., pinch four compatible cycles.

### 2.2.1 Massive pole exchange

For example, one such degeneration corresponding to a field-theory bubble diagram is depicted in Figure 9. Such Feynman diagrams correspond to a field theory with only cubic vertices and every cubic vertex is identified with a three-punctured disk in the string worldsheet. In fact, this relation can be made precise for the open string via Witten's open cubic string field theory.

In this way, we obtain one field-theory-like propagator for every pinched cycle in the open string worldsheet and the singularities that various regions in moduli space produce are expected to reproduce the standard behaviors in field theory as review below. In particular, the first picture in Figure 8 corresponds to the exchange of a massive intermediate particle. It leads to poles in the amplitude for $s \in \mathbb{Z}_{\geqslant 0}$ since there are physical particles in string theory that can go on-shell in this case. In fact, the amplitude will have double poles at $s \in \mathbb{Z}_{\geqslant 0}$ since

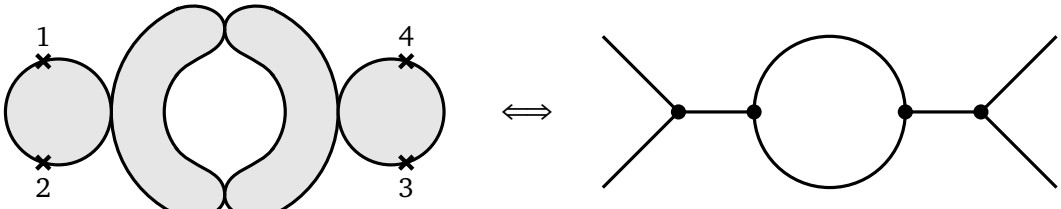

Figure 9: The maximal degeneration corresponding to a bubble diagram in the *s*-channel.

we can consider the double degeneration where also punctures 1 and 2 split off and produce a further propagator which can go on-shell.

As we shall discuss below, the existence of double poles at $s \in \mathbb{Z}_{\geqslant 0}$ is directly tied to the mass renormalization of the intermediate massive states. Since the mass of the massless particles is protected by gauge invariance, the double pole is actually absent for $s = 0$.

### 2.2.2 Wave function renormalization

The second picture in Figure 8 corresponds to the one-loop wave-function renormalization of the disk four-point amplitude. Let us see this explicitly. The degeneration is obtained by changing the coordinates to

$$z_2 = 1 - \lambda, \qquad z_3 = 1 - \lambda x, \qquad (2.10)$$

for $0 < x < 1$ and small $\lambda$. Then $x$ will become a cross-ratio on the disk that splits off from the annulus as depicted in the figure. We can see this directly at the level of the integrand. In this limit $\vartheta_1(z_{ij}, \tau) \sim z_{ij} \vartheta_1'(0)$ for $i, j \in \{2, 3, 4\}$ and hence the amplitude becomes

$$A_{\text{an}}^{\text{p}} = -i \int_{i\mathbb{R}_{\geqslant 0}} \mathrm{d}\tau \int_0^1 \mathrm{d}z_1 \int_0^\delta \mathrm{d}\lambda \int_0^1 \mathrm{d}x \, \lambda \, x^{-s} (1-x)^{-t}. \qquad (2.11)$$

This result is indeed proportional to the disk amplitude, which is obtained by integrating over $x$. We cut off the integral over $\lambda$ off at some small positive $\delta$, which is where the approximation of the degeneration breaks down. For fixed $\tau$, $z_1$ and $x$, the resulting integral over $\lambda$ is convergent as $\delta \to 0$. This means that the integrand is non-singular as we approach the degeneration corresponding to the wave function renormalization and thus the wave function renormalization actually vanishes. This is a consequence of supersymmetry: we are considering the scattering of massless gauge bosons which sit in a $\frac{1}{2}$-BPS multiplet of the spacetime supersymmetry algebra. This protects them from wave function renormalization.

The upshot of this discussion is that we do not have to worry about the degenerations corresponding to wave function renormalizations: nothing special is happening there.

### 2.2.3 Tadpoles

In a similar vein, the third diagram in Figure 8 represents the tadpole diagram of string theory. Consistency of the theory requires the vanishing of this diagram. This is indeed the case as one can see by a similar scaling as in (2.11), now setting

$$z_1 = 1 - \lambda, \qquad z_2 = 1 - \lambda x_2, \qquad z_3 = 1 - \lambda x_1, \qquad (2.12)$$

with $0 < x_1 < x_2 < 1$ being the two cross-ratios on the disk with five marked points. Thus also the regions in moduli space corresponding to the tadpole do not deserve special attention.

### 2.2.4 Non-separating degenerations

Finally, the most subtle degeneration is the non-separating degeneration depicted in the last picture of Figure 8. The name indicates that the resulting nodal surface is still connected: it is topologically a six-punctured disk with two punctures glued together. Non-separating degenerations cause the appearance of discontinuities and branch cuts in the string amplitude. In fact, the singularity structure near such a discontinuity is more complicated than the Deligne–Mumford compactification of moduli makes one suspect.

In particular, the string integrand does actually not extend to a smooth function over all of the compactified moduli space. The depicted degeneration corresponds to $\tau \to 0$ with $\frac{z_{ij}}{\tau}$ fixed, meaning that also all $z_{ij} \to 0$ at the same rate. It will turn out that it is actually more convenient to just take $\tau \to 0$ and keep all $z_i$'s fixed. The string amplitude automatically singles out the correct degenerations. To investigate the behaviour of the integrand near this degeneration, one uses the modular covariance of the integrand:

$$\left( \frac{\vartheta_1(z_{21}, \tau)\vartheta_1(z_{43}, \tau)}{\vartheta_1(z_{31}, \tau)\vartheta_1(z_{42}, \tau)} \right)^{-s} = e^{-\pi i s \tilde{\tau}(z_{21}^2 + z_{43}^2 - z_{42}^2 - z_{31}^2)} \left( \frac{\vartheta_1(z_{21}\tilde{\tau}, \tilde{\tau})\vartheta_1(z_{43}\tilde{\tau}, \tilde{\tau})}{\vartheta_1(z_{31}\tilde{\tau}, \tilde{\tau})\vartheta_1(z_{42}\tilde{\tau}, \tilde{\tau})} \right)^{-s} \tag{2.13}$$

$$= e^{2\pi i s \tilde{\tau} z_{32} z_{41}} \left( \frac{\vartheta_1(z_{21}\tilde{\tau}, \tilde{\tau})\vartheta_1(z_{43}\tilde{\tau}, \tilde{\tau})}{\vartheta_1(z_{31}\tilde{\tau}, \tilde{\tau})\vartheta_1(z_{42}\tilde{\tau}, \tilde{\tau})} \right)^{-s}, \tag{2.14}$$

where $\tilde{\tau} = -\frac{1}{\tau}$. Since $0 < z_{ij}\tilde{\tau} < \tilde{\tau}$ and $\tilde{\tau}$ has large imaginary part, we can actually use that the $n = 0$ term in the definition of the Jacobi theta function (2.2a) will dominate in this limit, i.e., $\vartheta_1(z_{ij}\tilde{\tau}, \tilde{\tau}) \sim i e^{-\pi i \tilde{\tau} z_{ij} + \frac{1}{4}\pi i \tilde{\tau}}$. This yields

$$A_{\text{an}}^{\text{p}} = -i \int_{i/\delta}^{i\infty} \frac{d\tilde{\tau}}{-\tilde{\tau}^2} \int dz_1 \, dz_2 \, dz_3 \, \tilde{q}^{-s(1-z_{41})z_{32} - t z_{43} z_{21}} + \text{higher order in } \tilde{q}, \tag{2.15}$$

where $\tilde{q} = e^{2\pi i \tilde{\tau}}$. We again cut off the integral over $\tau$ (and hence over $\tilde{\tau}$) off at small $\delta$ where our approximations break down.

One notices that, e.g., in the $s$-channel with large $s > 0$ and small $t < 0$, the exponent of $\tilde{q}$ can be negative and hence the integral over $\tilde{\tau}$ is generically divergent. For fixed $s$ and $t$, there is a fixed number of terms in the expansion of the $\vartheta_1$-function that yield divergent contributions as $\tilde{\tau} \to i\infty$. These are precisely the terms that can contribute to the imaginary part of the amplitude, since positive exponents of $\tilde{q}$ come with manifestly real coefficients and hence cannot contribute to $\text{Im} A_{\text{an}}^{\text{p}}$. Below, we will discuss how to properly deal with the divergent contributions. For now, let us note that the imaginary part of the amplitude is much simpler than the real part because only finitely many terms in the $\tilde{q}$-expansion contribute to it. This is of course expected physically because the imaginary part of the amplitude can in principle be computed by the optical theorem from tree-level amplitudes, which was discussed in detail in [14].

Let us also mention that this discussion immediately implies that the string integrand does *not* extend to a well-defined smooth function on the Deligne–Mumford compactification of moduli space. Indeed, which term in the $\tilde{q}$-expansion in (2.15) dominates for $\tilde{q} \to 0$ depends on the choice of the other moduli $z_i$ and consequently is not a smooth function of the $z_i$'s in this limit. Contrary to what is often stated, it means that the string integrands have a more complicated singularity structure than that predicted by the Deligne–Mumford compactification and to properly characterize them one would also need to consider a more complicated compactification of the moduli space. As far as we are aware, this has not been made precise in the literature.

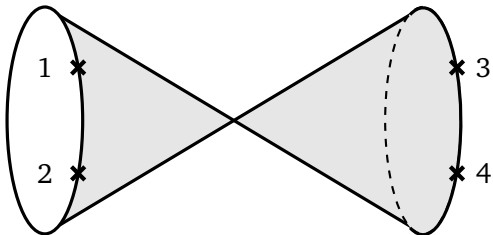

Figure 10: The degeneration leading to the closed string pole.

### 2.2.5 Closed string pole

There is one final singularity of the integrand that is special to the open string. If one views the annulus as a cylinder, then one can make the cylinder very long and pinch the corresponding closed string cycle. For the non-planar diagram, this leads to two disks connected at a single joint node as illustrated in Figure 10.

This degeneration is actually of real codimension-2, in contrast with codimension-1 cases encountered above, since it corresponds to a closed string degeneration and indeed each of the two disks has only one real modulus. Explicitly, this corresponds to taking $\tau \to i\infty$, where to leading order $\vartheta_4(z_{ij}, \tau) \sim 1$ and $\vartheta_1(z_{ij}, \tau) = -2e^{\frac{1}{4}\pi i \tau} \sin(\pi z_{ij})$. The amplitude becomes

$$A^{\text{n-p}} \sim -i \int_{i/\delta}^{i\infty} \mathrm{d}\tau \int_{0 \leqslant z_1 \leqslant z_2 \leqslant 1} \mathrm{d}z_1 \, \mathrm{d}z_2 \int_0^1 \mathrm{d}z_3 \, \left(4\sin(\pi z_{21}) \sin(\pi z_{43})\right)^{-s} e^{-\frac{1}{2}\pi i s \tau} \,. \tag{2.16}$$

The integral over $\tau$ is again divergent, but can in this case defined in an ad-hoc manner, e.g., by analytic continuation in $s$. The fact that the integrand only depends on $z_{21}$ and $z_{43}$ corresponds to the fact that this degeneration is really codimension-2. We can hence fix $z_2 = 1$ and compute

$$A^{\text{n-p}} \sim -\frac{2}{\pi s} \left( \int_0^1 \mathrm{d}z \, (2\sin(\pi z))^{-s} \right)^2 \,. \tag{2.17}$$

The remaining integral corresponds to the disk two-point function with one closed-string vertex operator. Higher closed-string string poles can be observed by keeping more terms in the $q$-expansion of the Jacobi theta functions. Since $\vartheta_4(z, \tau)$ has a $q$-expansion with half-integer powers of $q$, it leads to the following integrals over $\tau$:

$$-i \int_{i/\delta}^{i\infty} \mathrm{d}\tau \, e^{-\frac{1}{2}\pi i s \tau + \pi i k \tau} \sim -\frac{2}{\pi(s - 2k)} \,, \tag{2.18}$$

with integer $k$. Thus, the amplitude has a closed-string pole at every even integer. The factor of 2 compared to the normalization of the open-string spectrum corresponds to the usual relative normalization between the open and closed string spectrum.

For the planar amplitudes, this region in the moduli space also exists. However, the corresponding closed-string exchange happens at zero momentum and thus corresponds to a closed string tadpole. Explicitly, we have for the planar annulus amplitude for $\tau \to i\infty$

$$A^{\text{p}}_{\text{an}} \sim -i \int_{i/\delta}^{i\infty} \mathrm{d}\tau \int \mathrm{d}z_1 \, \mathrm{d}z_2 \, \mathrm{d}z_3 \, \left( \frac{\sin(\pi z_{21}) \sin(\pi z_{43})}{\sin(\pi z_{31}) \sin(\pi z_{42})} \right)^{-s} \left( \frac{\sin(\pi z_{32}) \sin(\pi z_{41})}{\sin(\pi z_{31}) \sin(\pi z_{42})} \right)^{-t} \,. \tag{2.19}$$

Here, we again replaced the Jacobi theta functions by their leading behaviour as $\tau \to i\infty$. The integrand becomes completely $\tau$-independent and hence the integral over $\tau$ diverges.

Nevertheless, we can observe that the Möbius strip has a similar divergence as $\tau \to i\infty$ and since $\tau$ and $\tau + \frac{1}{2}$ become indistinguishable for large $\text{Im}(\tau)$, the integrands approach exactly the same value, except for the overall minus sign that is present in (2.1b) compared to (2.1a) when $N = 32$. Hence the two divergences can be cancelled against each other in the full planar string amplitude. This is known as the Fischler–Susskind–Polchinski mechanism [28,29]. In the next section, we will see an elegant way of combining the annulus and the Möbius strip diagrams naturally.

# 3 Summary of the results

In this section, we explain all the conceptual ideas utilized in this paper without going too much into technical details. The full derivation of our formulas for the scattering amplitudes is given in Section 5 and Section 6, after a simpler warmup example that we discuss in Section 4. Consequences for the mass-shifts of the string spectrum are explored in Section 7. In this section, we also demonstrate the usefulness of the Rademacher expansion by explicitly evaluating it numerically.

## 3.1 Integration contour

After having discussed various singularities of the open string integrands, we move on to making sense of the integrals near the degenerations. One obvious idea for evaluating string amplitudes is to chop up the integration domain, compute the individual pieces in (typically disjoint) kinematic regions where they converge, and define the final result via analytic continuation back into a single kinematic point. This approach was used in the old literature on string amplitudes, see, e.g., [30–32], but is difficult to make practical beyond genus zero. The simple reason is that in order to perform analytic continuation, one needs an analytic expression to begin with and these are very hard to find for such an intricate object as a string scattering amplitude. However, such an analytic continuation was carried out for the imaginary part of the one-loop amplitude in type II strings [32, Theorem 2] and heterotic strings [32, Theorem 4], and produced explicit expressions for the decay widths of massive string states [32, Theorem 5]. One might define analytic continuation using dispersive methods such as using Mandelstam representation. But in contrast with quantum field theory, these are difficult to use due to the exponential divergences at infinity. Likewise, $\alpha'$-expansion is not an option because it would only provide an asymptotic series. Hence, we are lead to the conclusion that in order to evaluate string amplitudes at finite $\alpha'$, one has to face the problem of constructing the correct integration contour.

A physical picture for deciding how to construct the contour was proposed more recently by Witten [4]. He pointed out that the reason for the divergences is that we treat the worldsheet as a Euclidean 2D CFT, whereas the target space is Lorentzian. To get the correct causal structure of the amplitude we would hence like to perform the computations on a Lorentzian worldsheet. While this is not possible globally on the moduli space without introducing spurious UV divergences, it is possible near the degenerations where long tubes or strips develop on the worldsheet which can be endowed with a Lorentzian metric.

For a codimension-1 degeneration, there is always a modulus $\tilde{q}$ that represents a good local coordinate on moduli space such that $\tilde{q} = 0$ is the degenerate locus. For example, in the four degenerations depicted in Figure 8, $\tilde{q}$ corresponds to:

$$z_{43}, \quad z_{42}, \quad z_{41}, \quad \text{and} \quad e^{-\frac{2\pi i}{\tau}}, \tag{3.1}$$

respectively. Here, $\tilde{q}$ measures the width of the neck connecting the two parts of the surface, which is conformally equivalent to saying that $\tilde{q} = e^{-t}$, where $t$ is the proper Euclidean length

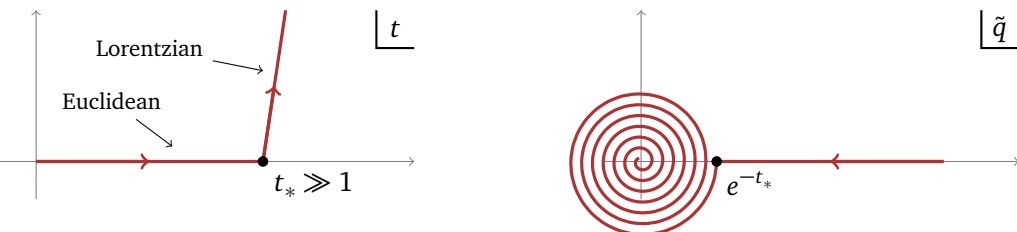

Figure 11: The integration contour in the neighborhood of the divisor at small $\tilde{q} = e^{-t}$. **Left:** In the $t$-plane, the Euclidean contour running along the real axis is rotated into a Lorentzian one after some large proper time $t_*$. The fact that the Lorentzian contour retains a small positive real part is equivalent to the Feynman $i\varepsilon$ prescription. **Right:** The image of the contour in the variable $\tilde{q}$ involves an infinite spiral onto the divisor at $\tilde{q} = 0$ with radius $e^{-t_*}$.

of the tube. Thus the Lorentzian analytic continuation corresponds to rotating into the upper half-plane after some large proper time $t_* \gg 1$, corresponding to the swirl contour in the $\tilde{q}$-plane, see Figure 11.

This necessitates the definition of a complexification of the open-string moduli space $\mathcal{M}_{1,n}$. We already discussed this complexification in Section 2.1. It is induced from the underlying complex moduli space of the torus that appears as the orientation double cover. In practice, it just corresponds to allowing complex $z_i$'s and arbitrary $\tau$ in the upper half-plane $\mathbb{H}$. The contour of integration is most interesting for the $\tau$-part of the integral, since it leads to the discontinuities of the amplitude. Here, we describe it for any $n$.

As we approach the $\tau \to 0$ region, the local parameter is given by $\tilde{q} = e^{-\frac{2\pi i}{\tau}}$. From the above discussion, the relevant contour hence takes the form

$$\frac{2\pi i}{\tau} = t_* + it\,, \tag{3.2}$$

for some large real constant $t_*$ and $t \in \mathbb{R}_{\geqslant 0}$ describing the Wick-rotated part of the contour, i.e.,

$$\tau = \frac{2\pi i}{t_* + it}\,. \tag{3.3}$$

This contour maps to a semi-circle in the complex $\tau$-plane with radius $\frac{\pi}{t_*}$ and centered at $\frac{\pi i}{t_*}$, bulging to the right, see Figure 12. The actual contour itself is of course not important since we can always deform it. The only important content of it is that we approach $\tau = 0$ from the right and not from the top. The direction in which we approach $\tau = 0$ matters since $\tau = 0$ is an essential singularity of the integrand.

The other interesting place of the $\tau$-integration is $\tau \to i\infty$, where in the planar case we expect the Fischler–Susskind–Polchinski mechanism to cancel the respective singularities and in the non-planar case the closed string pole originates from. Let us start with the planar case. Recall that the Möbius strip case is obtained by shifting $\tau \to \tau + \frac{1}{2}$ up to a minus sign compared to the annulus case, so the part of the contour close to $\tau = \frac{1}{2}$ looks the same as near $\tau = 0$, except for reversed orientation, see Figure 12. Most of the two contours cancel out and the end result is simply that the annulus and Möbius strip contours get connected, up to a contribution from $i\infty$. The resulting contour $\Gamma$ is displayed in Figure 12.

We should mention that there is a bit of choice involved in this contour. We could have for example also declared that the Möbius strip corresponds to $\tau \in \frac{3}{2} + i\mathbb{R}_{\geqslant 0}$ (since the integrands are periodic in $\tau$) and hence the horizontal connection between the contours would have been longer. Another common definition is to take the principal value of the integral in $q = e^{2\pi i\tau}$ that runs through the pole at $q = 0$. This corresponds to putting no horizontal part in the

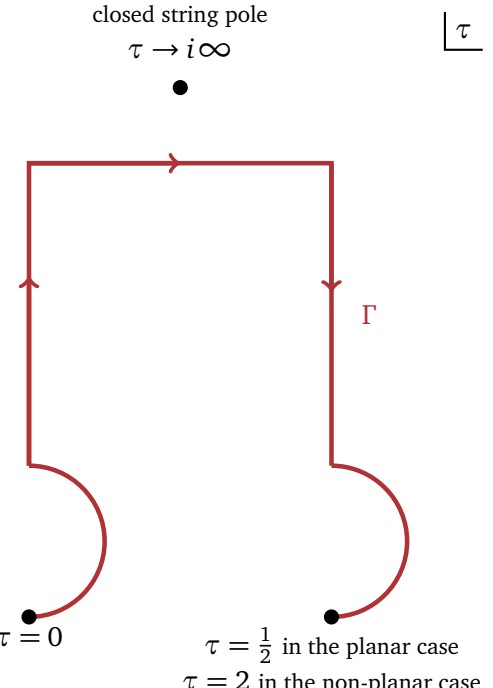

Figure 12: Contour of integration in the $\tau$-plane for open string one-loop amplitudes. The endpoint of the contour is $\tau = \frac{1}{2}$ in the planar case and $\tau = 2$ in the non-planar case.

contour. All these definitions differ by a multiple of the residue of the pole at infinity. More precisely, for the four-point amplitude, the principal value definition and the definition in terms of a closed contour differ by

$$\Delta A^{\mathrm{p}} = \frac{i}{2} \int_{0 \leqslant z_1 \leqslant z_2 \leqslant z_3 \leqslant 1} \mathrm{d}z_1 \, \mathrm{d}z_2 \, \mathrm{d}z_3 \left( \frac{\sin(\pi z_{21}) \sin(\pi z_{43})}{\sin(\pi z_{31}) \sin(\pi z_{42})} \right)^{-s} \left( \frac{\sin(\pi z_{32}) \sin(\pi z_{41})}{\sin(\pi z_{31}) \sin(\pi z_{42})} \right)^{-t}, \quad (3.4)$$

as worked out in eq. (2.19). The remaining integral is the disk four-point function with an additional closed-string dilaton vertex operator at zero momentum inserted. This follows directly from the geometry of the degeneration: for large $\tau$, the hole of the annulus closes and is replaced by a puncture with no momentum inflow. The leading term comes from the massless level and the Lorentz index structure only allows for a scalar, i.e., the dilaton. The dilaton vertex operator at zero momentum is in fact equal to the action itself and hence an insertion of this operator simply renormalizes $\alpha'$. As one can check the residue is indeed proportional to the $\alpha'$-derivative of the tree-level amplitude, explicitly

$$\Delta A^{\mathrm{p}} = \frac{i}{(2\pi)^2} \left[ \frac{\mathrm{d}}{\mathrm{d}s} \left( \frac{\Gamma(1-s)\Gamma(-t)}{\Gamma(1-s-t)} \right) + \frac{\mathrm{d}}{\mathrm{d}t} \left( \frac{\Gamma(-s)\Gamma(1-t)}{\Gamma(1-s-t)} \right) \right]. \quad (3.5)$$

We check this in Appendix A. This fact is known as the soft-dilaton theorem in string theory [33, 34].

In practice, we will hence compute with the closed contour $\Gamma$ as in Figure 12, but will then subtract the contribution from the closed string pole in the end in order to obtain the actual amplitude with correct reality properties. We will however mostly suppress this in the following discussion. Thus, the combined planar amplitude is given by

$$A^{\mathrm{p}} = \Delta A^{\mathrm{p}} - i \int_{\Gamma} \mathrm{d}\tau \int \mathrm{d}z_1 \, \mathrm{d}z_2 \, \mathrm{d}z_3 \left( \frac{\vartheta_1(z_{21}, \tau)\vartheta_1(z_{43}, \tau)}{\vartheta_1(z_{31}, \tau)\vartheta_1(z_{42}, \tau)} \right)^{-s} \left( \frac{\vartheta_1(z_{32}, \tau)\vartheta_1(z_{41}, \tau)}{\vartheta_1(z_{31}, \tau)\vartheta_1(z_{42}, \tau)} \right)^{-t}. \quad (3.6)$$

We did not spell out explicitly the appropriate contours for the $z_i$-integration. We will analyze them further below.

A very similar contour can be defined for the non-planar case. In this case, the $\tau$-contour as defined via the $i\varepsilon$ prescription is the same as the one for the planar annulus. However, we can notice that the integrand is quasi-periodic in $\tau \to \tau + 2$, which leads to the simple phase $e^{-\pi i s}$ of the integrand. This means that we can compute the expression

$$(1 - e^{-\pi i s})A^{\text{n-p}} \tag{3.7}$$

by subtracting from the original contour a contour that is shifted as $\tau \to \tau + 2$. This then cancels the infinite horizontal tail that runs horizontally to the left as $\tau = it_* - t$. The contour is then essentially identical to the planar contour, except that it ends at $\tau = 2$ instead of $\tau = \frac{1}{2}$, see Figure 12.

## 3.2 Rademacher contour

The Rademacher contour is a general method to compute contour integrals over modular objects. It was historically used to derive an asymptotic expansion for the partition numbers, which appear as the Fourier coefficients of the Dedekind eta-function $\eta(\tau)^{-1}$, see [35]. Let us explain the basic method at the simple example $\eta(\tau)^{-24}$, which gives the bosonic open-string partition function and hence is interesting in its own right. We will consider the more complicated examples relevant for the open superstring amplitudes below. Suppose we want to compute

$$\int_\Gamma \frac{d\tau}{\eta(\tau)^{24}}, \tag{3.8}$$

where the contour is identical to the one displayed in Figure 12 with the endpoint at $\tau = \frac{1}{2}$.[4]

One deforms the contour in a series of steps as follows. Let us first recall the *Farey sequence* $F_n$, which consists of all fractions $0 < \frac{a}{c} \leqslant 1$ such that $a$ and $c$ are coprime integers, $(a, c) = 1$, and $c \leqslant n$. By convention, we do not include 0, even though it is often included in the literature. The first few terms are

$$F_1 = \left(\tfrac{1}{1}\right), \tag{3.9a}$$

$$F_2 = \left(\tfrac{1}{2}, \tfrac{1}{1}\right), \tag{3.9b}$$

$$F_3 = \left(\tfrac{1}{3}, \tfrac{1}{2}, \tfrac{2}{3}, \tfrac{1}{1}\right), \tag{3.9c}$$

$$F_4 = \left(\tfrac{1}{4}, \tfrac{1}{3}, \tfrac{1}{2}, \tfrac{2}{3}, \tfrac{3}{4}, \tfrac{1}{1}\right), \tag{3.9d}$$

$$F_5 = \left(\tfrac{1}{5}, \tfrac{1}{4}, \tfrac{1}{3}, \tfrac{2}{5}, \tfrac{1}{2}, \tfrac{3}{5}, \tfrac{2}{3}, \tfrac{3}{4}, \tfrac{4}{5}, \tfrac{1}{1}\right). \tag{3.9e}$$

It is a non-trivial fact that one can draw *Ford circles* $C_{a/c}$ around the points $\tau = \frac{a}{c} + \frac{i}{2c^2}$ such that none of the circles overlap and two of them touch only if they are neighbors in the Farey sequence $F_n$ for some $n$.

We now construct a series of Rademacher contours $\Gamma_n$ as follows. For $n = 2$, we start with the contour that follows the arc of the Ford circle $C_{0/1}$ until the common point of $C_{0/1}$ and $C_{1/2}$ is reached, where we start following the arc of the Ford circle $C_{1/2}$ as depicted in Figure 13. The resulting contour is called $\Gamma_2$. It is equivalent to the original contour $\Gamma$ we described in Figure 12.

---

[4]In the open bosonic string, we face the same problem as in the planar four-point function discussed above, that we should really take the principal value at the pole $\tau = i\infty$ in order to get a physical result. For the bosonic string, there is a double pole at $\tau = i\infty$ and hence a correction term as in (3.5) does not exist. This makes the present computation unphysical.

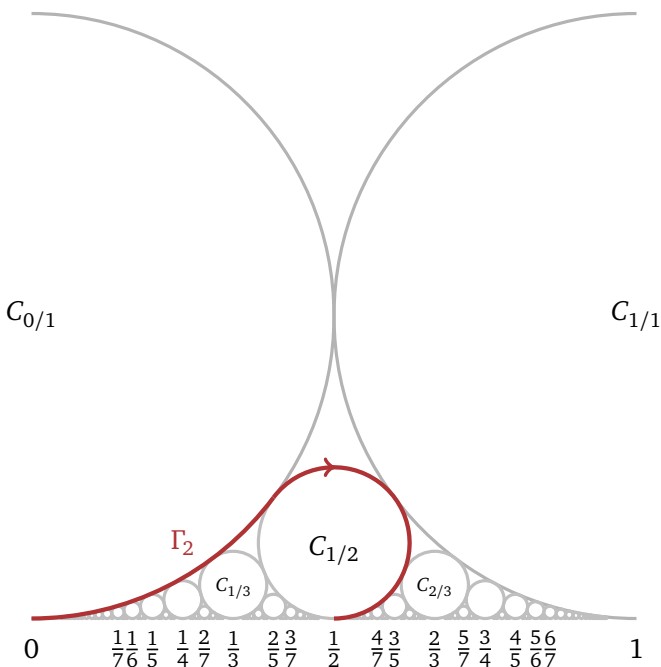

Figure 13: The Ford circles $C_{a/c}$ in the $\tau$ upper half-plane. The original contour of integration $\Gamma$ can be deformed to the second Rademacher contour $\Gamma_2$.

The contour $\Gamma_3$ includes the following modification to $\Gamma_2$: we follow the arc of $C_{0/1}$ only until it touches the circle $C_{1/3}$, which we then follow until we touch the circle $C_{1/2}$, which we then follow until $\tau = \frac{1}{2}$. We iteratively modify the contour further in the same way so that the contour $\Gamma_n$ describes an arc of $C_{0/1}$ until we meet $C_{1/n}$. We then follow the arcs of all the Ford circles in the Farey sequence $F_n$ until we reach the endpoint at $\tau = \frac{1}{2}$. Hence all Ford circles $C_{a/c}$ with $\frac{a}{c} \leqslant \frac{1}{2}$ and $c \leqslant n$ appear in the contour $\Gamma_n$. For example, the contour $\Gamma_5$, following the circles listed in (3.9e), is depicted in Figure 14.

The obvious idea is now to take the limiting contour $\Gamma_\infty$ that encircles every Ford circle $C_{a/c}$ with $0 < \frac{a}{c} \leqslant \frac{1}{2}$ precisely once and hence

$$\int_{\Gamma_\infty} \frac{\mathrm{d}\tau}{\eta(\tau)^{24}} = \sum_{c=1}^{\infty} \sum_{\substack{1 \leqslant a \leqslant \frac{c}{2} \\ (a,c)=1}} \int_{C_{a/c}} \frac{\mathrm{d}\tau}{\eta(\tau)^{24}} \,. \tag{3.10}$$

The contour $\Gamma_\infty$ was already illustrated in Figure 2. Of course it is not obvious from our discussion that this procedure converges, but we will find that it does in all the cases of interest. This can be proved rigorously by estimating the contributions to the integral from the remaining small arcs on the contour $\Gamma_n$. We do this in Appendix B. In fact, this procedure always converges when the modular weight of the integrand is negative.

So far, it may seem like this procedure has not gained us much. However, due to modular invariance, it is much simpler to compute the integral over the Ford circle than over the original contour. Indeed, consider the following modular transformation

$$\gamma(\tau) = \frac{a\tau + b}{c\tau + d}\,, \tag{3.11}$$

where $b$ and $d$ are chosen such that $ad - bc = 1$. Then

$$\eta\left(\frac{a\tau + b}{c\tau + d}\right)^{24} = (c\tau + d)^{12}\, \eta(\tau)^{24}\,. \tag{3.12}$$

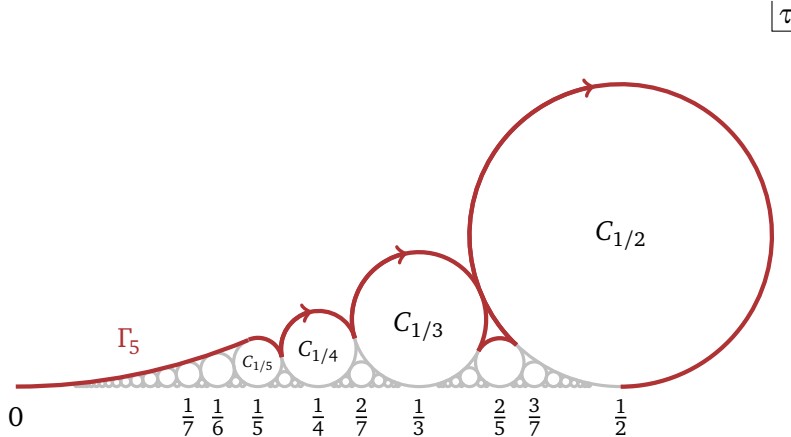

Figure 14: The fifth Rademacher contour $\Gamma_5$ obtained by following the Ford circles in the fifth Farey sequence (3.9e) according to the rules given in the text.

We can use this modular transformation to change variables in the integral over $C_{a/c}$ and obtain

$$\int_{C_{a/c}} \frac{\mathrm{d}\tau}{\eta(\tau)^{24}} = -\int_{\longrightarrow} \frac{\mathrm{d}\tau}{(c\tau + d)^{14}\,\eta(\tau)^{24}}. \tag{3.13}$$

The additional two powers of $c\tau + d$ come from the Jacobian. Due to our judicious choice of the modular transformation, the new contour runs now horizontally, i.e., we mapped the circle touching the real axis at $\tau = \frac{a}{c}$ to the circle at $\tau = i\infty$. After the modular transformation, the contour starts at $i - \infty$ and runs to $i + \infty$. This is opposite to the natural orientation of the circle at $i\infty$ and leads to the additional minus sign. The new integrand is holomorphic in the upper half-plane, except for a singularity at $\tau = i\infty$. We may hence deform the contour from $i + \mathbb{R}$ to $iL + \mathbb{R}$ for arbitrarily large $L$. We will frequently denote such a horizontal contour by $\longrightarrow$.

For large imaginary parts of $\tau$, it is then advantageous to use the Fourier expansion of $\eta(\tau)^{-24}$, which gives

$$\int_{C_{a/c}} \frac{\mathrm{d}\tau}{\eta(\tau)^{24}} = -\int_{\longrightarrow} \frac{\mathrm{d}\tau}{(c\tau + d)^{14}} \left(\mathrm{e}^{-2\pi i\tau} + 24 + \mathcal{O}(\mathrm{e}^{2\pi i\tau})\right). \tag{3.14}$$

For large $L$, all the contributions coming from $\mathcal{O}(\mathrm{e}^{2\pi i\tau})$ are exponentially suppressed and do not contribute. Similarly, the contribution from the constant term 24 does not contribute since it is polynomially suppressed thanks to the prefactor $(c\tau + d)^{-14}$. We thus conclude that we have the exact equality

$$\int_{C_{a/c}} \frac{\mathrm{d}\tau}{\eta(\tau)^{24}} = -\int_{\longrightarrow} \frac{\mathrm{d}\tau}{(c\tau + d)^{14}}\,\mathrm{e}^{-2\pi i\tau}. \tag{3.15}$$

The fact that we can reduce integrals of modular objects along the Ford circles back to integrals over elementary functions in this way is at the heart of the power of the Rademacher method.

After we have argued for the vanishing of the higher Fourier terms in the expansion, we can deform the contour back to finite $L$. In fact, we would like to deform it to large *negative* values of $L$, since the integrand is exponentially suppressed there. The only obstruction to this procedure is the $14^{\text{th}}$ order pole at $\tau = -\frac{d}{c}$ and hence its residue is the only contributing factor. Thus we get

$$\int_{C_{a/c}} \frac{\mathrm{d}\tau}{\eta(\tau)^{24}} = 2\pi i \operatorname*{Res}_{\tau=-\frac{d}{c}} \frac{\mathrm{e}^{-2\pi i\tau}}{(c\tau + d)^{14}} = \frac{(2\pi)^{14}\mathrm{e}^{\frac{2\pi id}{c}}}{13!\,c^{14}}, \tag{3.16}$$

where we recall that $d$ was determined by $a$ through $ad \equiv 1 \bmod c$. Let us write $d = a^*$ for the inverse mod $c$. Thus we find for the bosonic open string partition function

$$Z_{\text{open}} = -i \int_{\Gamma_\infty} \frac{d\tau}{\eta(\tau)^{24}} = \frac{-i(2\pi)^{14}}{13!} \sum_{c=1}^{\infty} \frac{1}{c^{14}} \sum_{\substack{1 \leqslant a \leqslant \frac{c}{2} \\ (a,c)=1}} e^{\frac{2\pi i a^*}{c}}. \tag{3.17}$$

This is a very fast-converging infinite sum representation of the partition function and trivial to evaluate numerically to very high accuracy.

## 3.3 Results for the four-point amplitudes

Using the same basic idea, one can also evaluate the integrals

$$\int_{C_{a/c}} d\tau \, dz_1 \, dz_2 \, dz_3 \left( \frac{\vartheta_1(z_{21},\tau)\vartheta_1(z_{43},\tau)}{\vartheta_1(z_{31},\tau)\vartheta_1(z_{42},\tau)} \right)^{-s} \left( \frac{\vartheta_1(z_{32},\tau)\vartheta_1(z_{41},\tau)}{\vartheta_1(z_{31},\tau)\vartheta_1(z_{42},\tau)} \right)^{-t}, \tag{3.18}$$

and similarly for the non-planar amplitude. Detailed derivations will be given in Section 5 and Section 6. Here, we simply present the results of this computation and highlight some of its features.

### 3.3.1 Planar amplitude in the $s$-channel with $s \leqslant 1$

Let us first explain the simplest case of interest: the planar amplitude $A^{\text{p}}$ in the $s$-channel for $0 < s \leqslant 1$ and $t < 0$. The amplitude, and hence also our formula, behaves discontinuously as we cross the normal thresholds of the string that are located at

$$s = (\sqrt{m_{\text{D}}} + \sqrt{m_{\text{U}}})^2, \tag{3.19}$$

for integers $m_{\text{D}}, m_{\text{U}} \in \mathbb{Z}_{\geqslant 0}$ corresponding to mass levels of string states in the units $\alpha' = 1$. Each threshold is connected with a new two-particle exchange being kinematically allowed. After the massless threshold, the next one appears at $s = 1$ corresponding to $(m_{\text{D}}, m_{\text{U}}) = (0, 1)$ or $(1, 0)$. Hence the formula for the amplitude is going to be particularly simple when $s \leqslant 1$.

Evaluating the integrals over the circle $C_{a/c}$ always leads to further sums which we label by integers $n_{\text{L}}, n_{\text{D}}, n_{\text{R}}, n_{\text{U}}$ satisfying the constraint $n_{\text{L}} + n_{\text{D}} + n_{\text{R}} + n_{\text{U}} = c - 1$. These integers are associated to particular winding numbers which we explain further below. We hence write

$$A^{\text{p}} = \Delta A^{\text{p}} + \sum_{c=1}^{\infty} \sum_{\substack{1 \leqslant a \leqslant \frac{c}{2} \\ (a,c)=1}} \sum_{\substack{n_{\text{L}}, n_{\text{D}}, n_{\text{R}}, n_{\text{U}} \geqslant 0 \\ n_{\text{L}}+n_{\text{D}}+n_{\text{R}}+n_{\text{U}}=c-1}} A_{a/c}^{n_{\text{L}}, n_{\text{D}}, n_{\text{R}}, n_{\text{U}}}, \tag{3.20}$$

where $\Delta A^{\text{p}}$ is given by (3.5). The individual contributions $A_{a/c}^{n_{\text{L}}, n_{\text{D}}, n_{\text{R}}, n_{\text{U}}}$ are given by

$$A_{a/c}^{n_{\text{L}}, n_{\text{D}}, n_{\text{R}}, n_{\text{U}}} = -\frac{16\pi i \, e^{-\pi i \sum_{a=\text{L,R}, b=\text{D,U}} \left[ s \sum_{m=n_a+1}^{n_a+n_b} + t \sum_{m=n_b+1}^{n_a+n_b} \right] ((\frac{md}{c}))}}{15 c^5 \sqrt{stu}} \int_{P>0} dt_{\text{L}} \, dt_{\text{R}}$$

$$\times P(s, t, t_{\text{L}}, t_{\text{R}})^{\frac{5}{2}} \left( \frac{\Gamma(-t_{\text{L}})\Gamma(s+t_{\text{L}})}{\Gamma(s)} \begin{cases} e^{2\pi i t_{\text{L}}((\frac{dn_{\text{L}}}{c}))}, & \text{if } n_{\text{L}} > 0, \\ \frac{\sin(\pi(s+t_{\text{L}}))}{\sin(\pi s)}, & \text{if } n_{\text{L}} = 0 \end{cases} \right) (\text{L} \leftrightarrow \text{R}). \tag{3.21}$$

Let us dissect this formula one by one. The following number-theoretic (discontinuous) *sawtooth* function makes an appearance:

$$((x)) = \begin{cases} x - \lfloor x \rfloor - \frac{1}{2}, & \text{if } x \notin \mathbb{Z}, \\ 0, & \text{if } x \in \mathbb{Z}. \end{cases} \tag{3.22}$$

As in the open-string partition function example discussed in Section 3.2, $d$ denotes the inverse of $a \bmod c$, i.e., $ad \equiv 1 \bmod c$. Here, $P(s, t, t_{\mathrm{L}}, t_{\mathrm{R}})$ is the following polynomial in $t_{\mathrm{L}}$ and $t_{\mathrm{R}}$, also known as the Baikov polynomial:

$$P(s, t, t_{\mathrm{L}}, t_{\mathrm{R}}) = \frac{s^2 t^2 - 2s^2 t t_{\mathrm{L}} + s^2 t_{\mathrm{L}}^2 - 2s^2 t t_{\mathrm{R}} - 2s^2 t_{\mathrm{L}} t_{\mathrm{R}} - 4s t t_{\mathrm{L}} t_{\mathrm{R}} + s^2 t_{\mathrm{R}}^2}{4st(s+t)}. \tag{3.23}$$

It measures the volume of the two-particle phase space: it is equal to $\ell_\perp^2$, where $\ell_\perp$ is the transverse part of the loop momentum that is orthogonal to all external momenta. The integration bounds are therefore impose only by taking $P > 0$.

Note that the two factors in the second line of this "generalized Baikov representation" are simply the tree-level Veneziano amplitudes decorated with extra phases. The left one depends only on "left" variables such as $t_{\mathrm{L}}$ and $n_{\mathrm{L}}$, while the other one only on the "right" variables.

The reader may rightfully ask why the representation (3.21) is better than the original integrals from (2.1), given that it still involves infinite sums over $a$, $c$, and $n_a$'s, as well as two integrals over $t_{\mathrm{L}}$ and $t_{\mathrm{R}}$. In the regime $s < 1$, interest in (3.21) is indeed rather of theoretical nature. While we believe that the representation is convergent, it does so very slowly and is also not absolutely convergent. Indeed, it is precisely on the cusp of convergence.

Let us understand why. The factor $A_{a/c}^{n_{\mathrm{L}}, n_{\mathrm{D}}, n_{\mathrm{R}}, n_{\mathrm{U}}}$ depends on $a$, $n_{\mathrm{L}}$, $n_{\mathrm{D}}$, $n_{\mathrm{R}}$ and $n_{\mathrm{U}}$ only via phases (disregarding for the moment the case distinction in (3.21)). Thus naively analyzing convergence of the whole sum (3.20) would lead one to the rough estimate

$$|A^{\mathrm{p}} - \Delta A^{\mathrm{p}}| \leqslant F(s, t) \sum_{c=1}^\infty \sum_{\substack{1 \leqslant a \leqslant \frac{c}{2} \\ (a,c)=1}} \sum_{\substack{n_{\mathrm{L}}, n_{\mathrm{D}}, n_{\mathrm{R}}, n_{\mathrm{U}} \geqslant 0 \\ n_{\mathrm{L}} + n_{\mathrm{D}} + n_{\mathrm{R}} + n_{\mathrm{U}} = c-1}} \frac{1}{c^5}. \tag{3.24}$$

The latter sum is logarithmically divergent because there are $\mathcal{O}(c^3)$ choices for $n_{\mathrm{L}}$, $n_{\mathrm{D}}$, $n_{\mathrm{R}}$ and $n_{\mathrm{U}}$ and $\mathcal{O}(c)$ choices for $a$ and thus we end up with a harmonic series. However, at least heuristically, we expect that the sum converges, albeit not absolutely. The reasoning is that for very large values of $c$, the phases in (3.21) look completely random and thus we expect that even though there are $\mathcal{O}(c^4)$ choices for $(a, n_{\mathrm{L}}, n_{\mathrm{D}}, n_{\mathrm{R}}, n_{\mathrm{U}})$, each sum only leads to a $\mathcal{O}(\sqrt{c})$ enhancement. This would lead to a convergent sum. Since the phases involve $s$ and $t$, convergence becomes worse for small values of $s$ and $t$ and completely breaks down if we try to approach $s \to 0$. This is the manifestation of a the massless branch cut in our formula.

However, while convergence of (3.21) for $s \leqslant 1$ is slow, it becomes much faster for the generalization of the formula for $s \geqslant 1$ that we describe below. In this regime, it is actually very practical and allows to directly evaluate the amplitude.

Let us now explain the geometrical interpretation of $n_{\mathrm{L}}$, $n_{\mathrm{D}}$, $n_{\mathrm{R}}$ and $n_{\mathrm{U}}$ as winding numbers. For this we should recall that we analytically continued $\tau$ inside the complexified moduli space, which we can identify with the moduli space of a torus (without invariance under modular transformations). For $\tau \sim \frac{a}{c}$, the relevant torus becomes very thin and the $z_i$'s are all on a line that winds around the long cycle of the torus $c$ times. For the case $\frac{a}{c} = \frac{0}{1}$ and $\frac{a}{c} = \frac{1}{2}$, this is just the fact that the boundary of the annulus goes once around the annulus, while the boundary of the Möbius strip winds twice around, see Figure 7.

Geometrically, we can hence think of a loop that winds $c$ times around itself with the four vertex operators on it. Every term $A_{a/c}^{n_{\mathrm{L}}, n_{\mathrm{D}}, n_{\mathrm{R}}, n_{\mathrm{U}}}$ corresponds to a consistent way of cutting this diagram in the $s$-channel. The integers $n_{\mathrm{L}}$, $n_{\mathrm{D}}$, $n_{\mathrm{R}}$ and $n_{\mathrm{U}}$ correspond to the amount of windings that separate the four vertex operators. We displayed the four possibilities for $c = 2$ in Figure 15.

In general, there are $\frac{1}{6} c(c+1)(c+2)$ ways to distribute the four vertex operators like this. This also explains why the case $n_{\mathrm{L}} = 0$ or $n_{\mathrm{R}} = 0$ plays a special role in the formula (3.21).

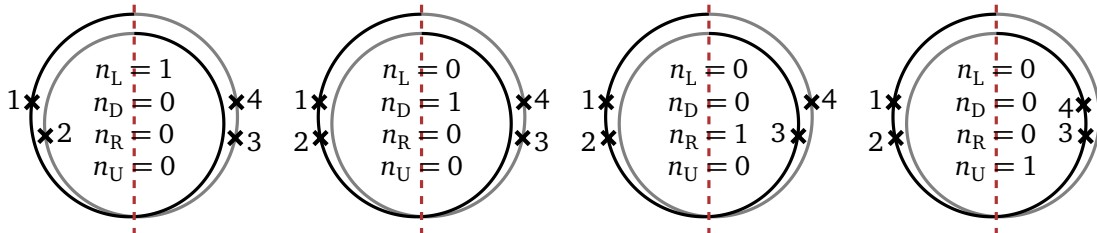

Figure 15: Four possibilities for windings of vertex operators for $c = 2$ that correspond to all the generalized $s$-channel cuts. For example, in the first case $(n_L, n_D, n_R, n_U) = (1, 0, 0, 0)$, because going from the puncture 1 to 2 requires one winding and no windings are necessary to travel between the other pairs of punctures.

In this case, two vertex operators can collide which manifests itself as poles in $s \in \mathbb{Z}_{>0}$ in the amplitude. The amplitude has in fact double poles at every positive integer $s$ corresponding to the mass renormalization of massive states. We can read-off the mass shifts from the prefactors of these double poles. The only diagrams that contribute to these prefactors are those with $n_L = n_R = 0$. Thus the mass shifts are much simpler physical quantities than the full amplitude and we also analyze them extensively in this paper. Our results for them are discussed in Section 3.4.

### 3.3.2 Higher values of $s$

Equation (3.21) can be systematically extended to higher values of $s$. In this case, we get contributions from each mass-level that can be exchanged in the scattering process labelled by the integers $m_D$ and $m_U$ mentioned above. The generalization of (3.21) now reads

$$
A_{a/c}^{n_L, n_D, n_R, n_U} = - \frac{16\pi i\, e^{-\pi i \sum_{a=L,R,\, b=D,U} \left[ s \sum_{m=n_a+1}^{n_a+n_b} + t \sum_{m=n_b+1}^{n_a+n_b} \right] ((\frac{md}{c}))}}{15 c^5 \sqrt{stu}} \sum_{\substack{m_D, m_U \geqslant 0 \\ (\sqrt{m_D}+\sqrt{m_U})^2 \leqslant s}}
$$

$$
\times\, e^{\frac{2\pi i d}{c}(m_D n_D + m_U n_U)} \int_{P_{m_D,m_U}>0} dt_L\, dt_R\, P_{m_D,m_U}(s,t,t_L,t_R)^{\frac{5}{2}}\, Q_{m_D,m_U}(s,t,t_L,t_R)
$$

$$
\times \left( \frac{\Gamma(-t_L)\Gamma(s+t_L-m_D-m_U)}{\Gamma(s)} \begin{cases} e^{2\pi i t_L ((\frac{d n_L}{c}))}, & \text{if } n_L > 0, \\ \frac{\sin(\pi(s+t_L))}{\sin(\pi s)}, & \text{if } n_L = 0 \end{cases} \right) (L \leftrightarrow R). \quad (3.25)
$$

The polynomials $P_{m_D,m_U}$ still have a purely kinematical interpretation of $\ell_\perp^2$. They are explicitly given by

$$
P_{m_D,m_U}(s,t,t_L,t_R) = -\frac{1}{4stu} \det \begin{bmatrix} 0 & s & u & m_U - s - t_L \\ s & 0 & t & t_L - m_D \\ u & t & 0 & m_D - t_R \\ m_U - s - t_L & t_L - m_D & m_D - t_R & 2m_D \end{bmatrix}, \quad (3.26)
$$

where the determinant arises kinematically as a certain Gram determinant.

Moreover, the factors $Q_{m_D,m_U}(s,t,t_L,t_R)$ are polynomials in the four arguments. Physically, they appear from the sum over polarizations and degeneracies of the internal states. They are defined as follows. Take

$$Q_{m_{\rm L},m_{\rm D},m_{\rm R},m_{\rm U}}(s,t) = [q_{\rm L}^{m_{\rm L}} q_{\rm D}^{m_{\rm D}} q_{\rm R}^{m_{\rm R}} q_{\rm U}^{m_{\rm U}}] \prod_{\ell=1}^{\infty} \prod_{a={\rm L,R}} (1-q^\ell q_a^{-1})^{-s}(1-q^\ell q_a)^{-s}$$

$$\times \prod_{a={\rm D,U}} (1-q^\ell q_a^{-1})^{-t}(1-q^{\ell-1} q_a)^{-t}$$

$$\times \prod_{a={\rm L,R}} (1-q^\ell q_a^{-1} q_{\rm D}^{-1})^{-u}(1-q^{\ell-1} q_a q_{\rm D})^{-u}\,, \tag{3.27}$$

where $q = q_{\rm L} q_{\rm D} q_{\rm R} q_{\rm U}$ and $[q_{\rm L}^{m_{\rm L}} q_{\rm D}^{m_{\rm D}} q_{\rm R}^{m_{\rm R}} q_{\rm U}^{m_{\rm U}}]$ denotes the coefficient of the relevant term in the series expansion around each $q_a = 0$. We then have

$$Q_{m_{\rm D},m_{\rm U}}(s,t,t_{\rm L},t_{\rm R}) = \sum_{m_{\rm L},m_{\rm R}=0}^{m_{\rm D}+m_{\rm U}} Q_{m_{\rm L},m_{\rm D},m_{\rm R},m_{\rm U}}(s,t)(-t_{\rm L})_{m_{\rm L}}(-s-t_{\rm L}+m_{\rm L}+1)_{m_{\rm D}+m_{\rm U}-m_{\rm L}}$$

$$\times (-t_{\rm R})_{m_{\rm R}}(-s-t_{\rm R}+m_{\rm R}+1)_{m_{\rm D}+m_{\rm U}-m_{\rm R}}\,, \tag{3.28}$$

where $(a)_n = a(a+1)\cdots(a+n-1)$ is the rising Pochhammer symbol. In practice, we computed all the polynomials $Q_{m_{\rm D},m_{\rm U}}$ with $(\sqrt{m_{\rm D}}+\sqrt{m_{\rm U}})^2 \leqslant s \leqslant 39$. They rapidly grow in the number of terms. In the ancillary file Q.txt we included all the ones needed to reproduce our results up to $s \leqslant 16$.

In the language of the Rademacher expansion, the sum over $m_{\rm D}$ and $m_{\rm U}$ corresponds to the sum over so-called polar terms in the modular integrand. When crossing one of the production thresholds, a new polar term arises and contributes to the integral.

### 3.3.3   Imaginary part

As explained in Section 3.1 and [14], the imaginary part is much simpler to compute. To be precise, we have

$$\operatorname{Im} A^{\rm p} = -\frac{1}{2i}\left( A_{0/1}^{0,0,0,0} - \sum_{\substack{n_{\rm L},n_{\rm D},n_{\rm R},n_{\rm U}\geqslant 0 \\ n_{\rm L}+n_{\rm D}+n_{\rm R}+n_{\rm U}=1}} A_{1/2}^{n_{\rm L},n_{\rm D},n_{\rm R},n_{\rm U}}\right). \tag{3.29}$$

The first term corresponds to the $s$-channel cut of the annulus computed as a circle anchored at $\tau = \frac{0}{1}$ (in this edges case, we set $a^* = 0$). The other four terms correspond to the four possible ways that we can cut the Möbius strip in the $s$-channel, computed using a Ford circle at $\tau = \frac{1}{2}$. The overall minus sign comes about because of the orientation of the contour and $\frac{1}{2i}$ is the normalization extracting the imaginary part, see [14] for details.

It is a very non-trivial identity that the imaginary part of eq. (3.20) recovers indeed eq. (3.29). While our derivation shows that this indeed holds, we do not have a direct proof of this fact (see however below for some special cases).

### 3.3.4   Other channels and the non-planar case

We also derive the corresponding formulas for the $u$-channel of the planar amplitude and for the $s$- and $u$-channel of the non-planar amplitude. The reader can find the results in these three cases in eq. (5.48), (6.33) and (6.42). The formulas are essentially all identical, except that the allowed range of $(n_{\rm L},n_{\rm D},n_{\rm R},n_{\rm U})$ is different and the appearing phases are slightly different. The $u$-channel formulas also do not exhibit poles because the corresponding vertex operators are not allowed to collide. We also remark again that the non-planar formula does not need a correction from the cusp, i.e. there is no $\Delta A^{\rm n\text{-}p}$. For the non-planar amplitude, the range of fractions runs from $0 < \frac{a}{c} \leqslant 2$, since the endpoint of the integration contour is different, see Figure 12. We should also remember that the non-planar amplitude has an additional factor of $(1-{\rm e}^{-\pi i s})A^{\rm n\text{-}p}$ in front, see eq. (3.7). Thus naively, the amplitude has triple poles at every even integer $s$. One can however easily check that they cancel out of the final expression.

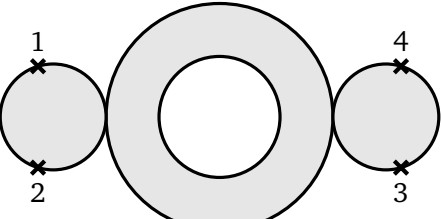
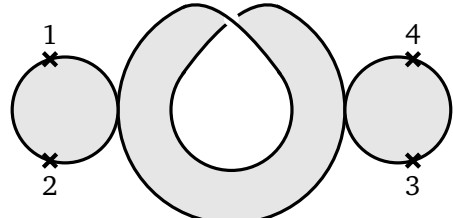

Figure 16: Worldsheet configurations leading to double poles at every $s \in \mathbb{Z}_{>0}$ for the planar annulus and Möbius strip topologies respectively.

### 3.4 Results for the mass-shifts

As already mentioned, the above formulas allow us to compute mass-shifts in a convenient way. Recall that they originate from the worldsheet degenerations illustrated in Figure 16. Mass shifts are given by the coefficient of the double pole $\mathrm{DRes}_{s=s_*}$ in (3.25) at every positive integer $s_*$. Only the terms with $n_\mathrm{L} = n_\mathrm{R} = 0$ contribute and for them we have

$$
\mathrm{DRes}_{s=s_*} A^{0,n_\mathrm{D},0,n_\mathrm{U}}_{a/c} = -\frac{16\pi i \, e^{\frac{2\pi i s_* d}{c} n_\mathrm{D} n_\mathrm{U}}}{15 c^5 \sqrt{-s_* t(s_* + t)} \, \Gamma(s_*)^2} \sum_{\substack{m_\mathrm{D}, m_\mathrm{U} \geqslant 0 \\ (\sqrt{m_\mathrm{D}} + \sqrt{m_\mathrm{U}})^2 \leqslant s_*}} e^{\frac{2\pi i d}{c}(m_\mathrm{D} n_\mathrm{D} + m_\mathrm{U} n_\mathrm{U})}
$$
$$
\times \int_{P_{m_\mathrm{D},m_\mathrm{U}} > 0} dt_\mathrm{L} \, dt_\mathrm{R} \, P_{m_\mathrm{D},m_\mathrm{U}}(s_*, t, t_\mathrm{L}, t_\mathrm{R})^{\frac{5}{2}} \, Q_{m_\mathrm{D},m_\mathrm{U}}(s_*, t, t_\mathrm{L}, t_\mathrm{R})
$$
$$
\times (t_\mathrm{L} + 1)_{s_* - m_\mathrm{D} - m_\mathrm{U} - 1}(t_\mathrm{R} + 1)_{s_* - m_\mathrm{D} - m_\mathrm{U} - 1}. \tag{3.30}
$$

For every mass-level, the integral over $t_\mathrm{L}$ and $t_\mathrm{R}$ can be explicitly evaluated and gives a polynomial of degree $s_*-1$ in $t$. In particular, in the simplest mass-shift at $s = 1$ takes the simple form

$$
\mathrm{DRes}_{s=1} A^\mathrm{p} = \frac{i}{(2\pi)^2} - \frac{\pi^2 i}{210} \sum_{c=1}^{\infty} \frac{1}{c^5} \sum_{\substack{1 \leqslant a \leqslant \frac{c}{2} \\ (a,c)=1}} \sum_{n=0}^{c-1} e^{-\frac{2\pi i n(n+1)a^*}{c}}, \tag{3.31}
$$

where $d = a^*$ denotes again the inverse mod $c$. Such sums are classical objects in number theory. In particular, the sum over $n$ is known as a *Gauss sum* and can be explicitly evaluated in terms of the Jacobi symbol (a generalization of the Legendre symbol).

One can also perform strong tests on these formulas by invoking some further number theory. There are two ways to compute the imaginary part of the mass-shift: either take the imaginary part directly in (3.30) or take the imaginary part in (3.29). For the mass-shifts we can check the equality of the two formulas directly. We show that

$$
F^{m_\mathrm{D},m_\mathrm{U}}_s(c) = \frac{2}{c} \mathrm{Im}\left[ i \sum_{\substack{1 \leqslant a \leqslant \frac{c}{2} \\ (a,c)=1}} \sum_{\substack{n_\mathrm{D},n_\mathrm{U} \geqslant 0 \\ n_\mathrm{D}+n_\mathrm{U}=c-1}} e^{\frac{2\pi i d}{c}(s n_\mathrm{D} n_\mathrm{U} + m_\mathrm{D} n_\mathrm{D} + m_\mathrm{U} n_\mathrm{U})} \right] \tag{3.32}
$$

is almost a multiplicative function, meaning that (suppressing other labels)

$$
F(c) = F(p_1^{\ell_1}) \cdots F(p_k^{\ell_k}), \tag{3.33}
$$

where $c = p_1^{\ell_1} \cdots p_k^{\ell_k}$ is the prime factorization of $c$. The identity (3.33) can fail for finitely many prime numbers, but they can be treated separately. For example, in the simplest case for

the mass-shift at $s = 1$, we have

$$F_1^{0,0}(c) = \begin{cases} 0, & \text{if } c = 1 \text{ or } c \geqslant 3 \text{ contains a square,} \\ 2, & \text{if } c = 2, \\ 1, & \text{if } c \geqslant 3 \text{ is square-free,} \end{cases} \tag{3.34}$$

where we recall that square-free means that the number has no repeated prime factor. Using properties of Euler products, one has

$$\frac{105}{\pi^4} = \frac{\zeta(4)}{\zeta(8)} = \sum_{c=1}^{\infty} \frac{1}{c^4} \begin{cases} 0, & \text{if } c \geqslant 3 \text{ contains a square,} \\ 1, & \text{if } c \geqslant 3 \text{ is square-free,} \end{cases} \tag{3.35}$$

which leads to the exact evaluation

$$\mathrm{Im}\,(3.31) = \frac{\pi^2}{448}. \tag{3.36}$$

This result agrees with the one obtained by directly extracting the double pole from (3.29). Similarly, one can check that the corresponding equalities hold for higher values of $s$, where they involve more non-trivial $L$-functions that generalize the Riemann zeta-function appearing in (3.35). This computation provides a completely independent check of (parts of) our formula (3.25).

The generalization to higher $s = s_*$ is quite straightforward. In the ancillary file `DRes.txt`, we included the expressions in terms of Gauss sums needed that compute them up to $s \leqslant 16$. To highlight the main result, we find

$$\underset{s=1}{\mathrm{DRes}}\, A^{\mathrm{p}} = d_1 + \frac{\pi^2}{448} i, \tag{3.37a}$$

$$\underset{s=2}{\mathrm{DRes}}\, A^{\mathrm{p}} = (1+t)\left(d_2 + \frac{17\pi^2}{7560} i\right), \tag{3.37b}$$

$$\underset{s=3}{\mathrm{DRes}}\, A^{\mathrm{p}} = (1+t)(2+t)\left(d_3 + \frac{167341\pi^2}{143700480} i\right). \tag{3.37c}$$

The imaginary parts can be evaluated exactly and the real parts $d_{s_*}$ are constants we can compute with arbitrary precision, but did not manage to express in terms of known quantities:

$$d_1 \approx 8.36799 \cdot 10^{-5}, \qquad d_2 \approx -1.61091 \cdot 10^{-4}, \qquad d_3 \approx -9.05359 \cdot 10^{-6}. \tag{3.38}$$

The neat factorization pattern into $(1+t)(2+t)\cdots(s_*-1+t)$ breaks down starting with $s_* = 4$ (but as noticed in [14], it holds approximately for the imaginary parts). The reason is that the spectrum no longer consists of a single supermultiplet and particles of varies spins at the same mass level get different mass shifts and decay widths. Numerical evaluation of higher $\mathrm{DRes}_{s=s_*}$ up to $c \leqslant 1000$ is given in App. D.

Since the mass shifts are easier to evaluate, we can also use them to explore the behaviour of the amplitude at high energies. Since they control the value of the amplitude at all integer values of $s \in \mathbb{Z}$, they give an approximate idea of the high-energy behaviour. However, as we shall see the real part of the amplitude oscillates and thus only knowing integer values can be somewhat misleading.

## 3.5 Numerical computations and convergence

Let us explain how to use the Rademacher formula in practical computations and summarize our observations on the convergence in $c$. To take specific examples, we will discuss the type of manipulations that went into producing the plots shown in Section 1.

The first step is to simplify the integration domain, which can be done by a change of variables from $(t_{\mathrm{L}}, t_{\mathrm{R}})$ to $(x, y)$ as follows:

$$t_{\mathrm{L/R}} = \frac{\sqrt{\Delta_{m_{\mathrm{D}}, m_{\mathrm{U}}}}}{2\sqrt{s}}(\sqrt{-u}x \pm \sqrt{-t}y) + \frac{1}{2}(m_{\mathrm{D}} + m_{\mathrm{U}} - s), \qquad (3.39)$$

where we introduced

$$\Delta_{m_{\mathrm{D}}, m_{\mathrm{U}}}(s) = \left[s - (\sqrt{m_{\mathrm{D}}} + \sqrt{m_{\mathrm{U}}})^2\right]\left[s - (\sqrt{m_{\mathrm{D}}} - \sqrt{m_{\mathrm{U}}})^2\right]. \qquad (3.40)$$

This change leads to the Jacobian $\sqrt{tu}\,\Delta_{m_{\mathrm{D}}, m_{\mathrm{U}}}/2s$. In addition, the polynomial $P_{m_{\mathrm{D}}, m_{\mathrm{U}}}$ simplifies to

$$P_{m_{\mathrm{D}}, m_{\mathrm{U}}} = \frac{\Delta_{m_{\mathrm{D}}, m_{\mathrm{U}}}(s)}{4s}(1 - x^2 - y^2), \qquad (3.41)$$

and hence the overall powers of $\Delta_{m_{\mathrm{D}}, m_{\mathrm{U}}}$ can be pulled out of the integral. We thus obtain the simplified formula for (5.33):

$$A_{a/c}^{n_{\mathrm{L}}, n_{\mathrm{D}}, n_{\mathrm{R}}, n_{\mathrm{U}}} = -\frac{\pi i \, e^{-\pi i \sum_{a=\mathrm{L,R},\, b=\mathrm{D,U}}\left[s \sum_{m=n_a+1}^{n_a+n_b} + t \sum_{m=n_b+1}^{n_a+n_b}\right](\!(\frac{md}{c})\!)}}{60 c^5 s^4 \Gamma^2(s)\sin^2(\pi s)} \sum_{\substack{m_{\mathrm{D}}, m_{\mathrm{U}} \geqslant 0 \\ (\sqrt{m_{\mathrm{D}}} + \sqrt{m_{\mathrm{U}}})^2 \leqslant s}} \Delta_{m_{\mathrm{D}}, m_{\mathrm{U}}}^{\frac{7}{2}}(s)$$

$$\times e^{\frac{2\pi i d}{c}(m_{\mathrm{D}} n_{\mathrm{D}} + m_{\mathrm{U}} n_{\mathrm{U}})} \int_{\mathbb{D}} \mathrm{d}x\,\mathrm{d}y\,(1 - x^2 - y^2)^{\frac{5}{2}} Q_{m_{\mathrm{D}}, m_{\mathrm{U}}}(s, t, t_{\mathrm{L}}, t_{\mathrm{R}})$$

$$\times \left(\Gamma(-t_{\mathrm{L}})\Gamma(s + t_{\mathrm{L}} - m_{\mathrm{D}} - m_{\mathrm{U}}) \begin{cases} e^{2\pi i t_{\mathrm{L}}(\!(\frac{dn_{\mathrm{L}}}{c})\!)}\sin(\pi s), & \text{if } n_{\mathrm{L}} > 0, \\ \sin(\pi(s + t_{\mathrm{L}})), & \text{if } n_{\mathrm{L}} = 0 \end{cases}\right)(\mathrm{L} \leftrightarrow \mathrm{R}).$$

$$(3.42)$$

The integration domain $\mathbb{D}$ is the unit disk, $x^2 + y^2 < 1$. Written this way, every integral in the sum is convergent and the only singularities come from the explicit sine function out in the front. This is the reason, in all computations we multiply out by $\sin^2(\pi s)$ in order to remove these divergences from plots. As mentioned before, these double poles at every positive integer $s$ are associated with worldsheet degenerations illustrated in Figure 16 and they come from the terms $n_{\mathrm{L}} = 0$ and $n_{\mathrm{R}} = 0$ where two punctures can collide. Analogous formulas can be derived in other channels and the non-planar case.

In the forward limit, $t = 0$, the expression undergoes some simplifications because $t_{\mathrm{L}} = t_{\mathrm{R}}$ and consequently the variable $y$ can be integrated out. The result is

$$A_{a/c}^{n_{\mathrm{L}}, n_{\mathrm{D}}, n_{\mathrm{R}}, n_{\mathrm{U}}}\bigg|_{t=0} = -\frac{i\pi^2 e^{-\pi i s \sum_{a=\mathrm{L,R},\, b=\mathrm{D,U}} \sum_{m=n_a+1}^{n_a+n_b}(\!(\frac{md}{c})\!)}}{192 c^5 s^4 \Gamma^2(s)\sin^2(\pi s)} \sum_{\substack{m_{\mathrm{D}}, m_{\mathrm{U}} \geqslant 0 \\ (\sqrt{m_{\mathrm{D}}} + \sqrt{m_{\mathrm{U}}})^2 \leqslant s}} \Delta_{m_{\mathrm{D}}, m_{\mathrm{U}}}^{\frac{7}{2}}(s)\, e^{\frac{2\pi i d}{c}(m_{\mathrm{D}} n_{\mathrm{D}} + m_{\mathrm{U}} n_{\mathrm{U}})}$$

$$\times \int_{-1}^{1} \mathrm{d}x\,(1 - x^2)^3 Q_{m_{\mathrm{D}}, m_{\mathrm{U}}}(s, 0, t_{\mathrm{L}}, t_{\mathrm{L}})\Gamma^2(-t_{\mathrm{L}})\Gamma^2(s + t_{\mathrm{L}} - m_{\mathrm{D}} - m_{\mathrm{U}})$$

$$\times \left(\begin{cases} e^{2\pi i t_{\mathrm{L}}(\!(\frac{dn_{\mathrm{L}}}{c})\!)}\sin(\pi s), & \text{if } n_{\mathrm{L}} > 0, \\ \sin(\pi(s + t_{\mathrm{L}})), & \text{if } n_{\mathrm{L}} = 0 \end{cases}\right)(\mathrm{L} \leftrightarrow \mathrm{R}). \qquad (3.43)$$

In practice, we perform the sums over the winding numbers $n_{\mathrm{L}}$, $n_{\mathrm{D}}$, $n_{\mathrm{R}}$, $n_{\mathrm{U}}$ and fractions $\frac{a}{c}$ within the integrand. Notice that $a$ never appears explicitly, so the sum can be expressed as that over $d$, running over the range $\{1, 2, \ldots, \lfloor\frac{c}{2}\rfloor\}^*$, where the star $*$ denotes the inverse mod $c$.

To perform a computation in a finite amount of time, we need to truncate the sum in (3.20) at some $c$. As highlighted before, due to the oscillating terms in the sums, it is difficult

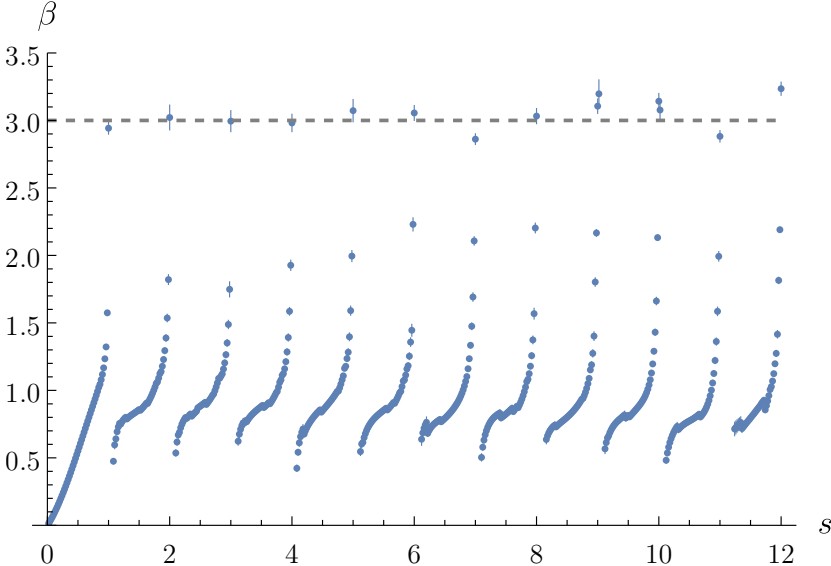

Figure 17: Fitted values of the exponent $\beta$ in (3.44) together with their standard deviation error bars. The gray dashed line corresponds to $\beta = 3$ expected at positive integers $s$.

to accurately estimate the truncation errors analytically. As an alternative, we can fit the dependence on $c$ and extrapolate the data to $c \to \infty$, thus obtaining some error bars on the amplitude computed using the Rademacher method.

Let us first use fitting to get an estimate on the rate of convergence of the Rademacher method. This can be done quite reliably for the imaginary part, since in that case we can compute the exact value with arbitrary precision using (3.29). On the other hand, we take the imaginary part of (3.20) computed up to $c \leqslant c_*$ and fit the result to the simple ansatz

$$(3.29) - \alpha \, c_*^{-\beta} \,. \tag{3.44}$$

The exponent $\beta$ in principle depends on the kinematics. Note that it corresponds to the convergence of partial sums, not individual terms in the $c$-sum. Any positive $\beta$ indicates convergence. The "random phase" model explained in Section 3.3.1 corresponds to $\beta \approx 2$. In Section 7, we will prove that for positive integers $s$, we have $\beta = 3$. In Appendix B, we further argue that that $\beta > 0$ for every $s > 0$. The goal of the following discussion is to extract $\beta$ directly from the data. We focus on the forward limit case, $t = 0$, corresponding to the amplitude plotted in Figure 4.

After taking a logarithm, (3.44) can be fitted using linear regression. In practice, it is difficult to control the systematic errors coming from the fact we might not have reached the asymptotic regime in $c$. In practice we have access to data $c_* \leqslant c_{max}$, where $c_{max} = 68$ for data points with $s \leqslant 1$ and $c_{max} = 40$ for those with $s \leqslant 12$. It is beneficial to drop multiple data points at low $c_*$, but not too many to reduce the statistics. To find a balance, we scan over multiple cutoffs $c_{min}$ such that fitting only the data $c_{min} \leqslant c_* \leqslant c_{max}$ gives the most accurate fit, quantified by the highest value of the adjusted coefficient of determination $\bar{R}^2$. We also discard all unreliable fits with $\bar{R}^2 < 0.99$. The resulting $\beta$'s together with the error bars are plotted in Figure 17.

As anticipated, at positive integers $s$, convergence reaches $\beta \approx 3$ predicted by the Gauss sum formula. We observe that for $s$ just above an integer, convergence changes drops drastically. As expected, when $s \to 0$, the value $\beta \approx 0$ is reached indicating poorer and poorer convergence due to lack of cancellations between terms in the Rademacher expansion. Across

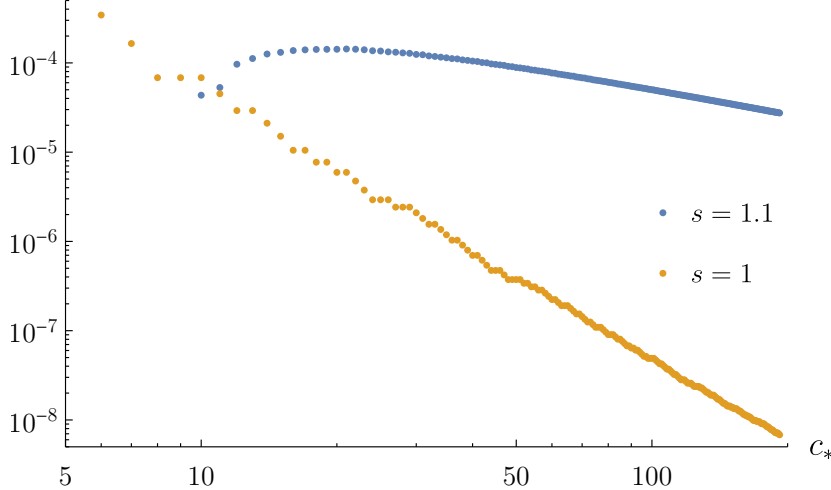

Figure 18: Two examples of the difference between $c_*$-truncated and exact values of $\operatorname{Im} A^{\mathrm{p}}(s, 0)$ as a function of $c_*$ for $s = 1$ and $s = 1.1$, illustrating a drastic change in convergence rates visible in Figure 17.

all energies, the results are consistent with $\beta > 0$.

In order to illustrate why the jumps in convergence happens across integers, in Figure 18 we plot $-\alpha c_*^{-\beta}$ (the difference between truncated and exact values) for two values: $s = 1$ and $s = 1.1$. The former reaches the asymptotic behavior fairly soon. For example, keeping the data points $c_{\max} = 40$ leads to the fit $\beta = 3.07 \pm 0.09$, while taking the extended set $c_{\max} = 190$ gives $\beta = 3.013 \pm 0.012$. On the other hand, for $s = 1.1$, the data set $c_{\max} = 40$ gives rise to the fit $\beta = 0.61 \pm 0.03$, while the set $c_{\max} = 190$ gives $\beta = 0.911 \pm 0.002$. The two values disagree, indicating a presence of the systematic error: the fact that for a given $c_{\max}$ the asymptotic regime might not have been reached. The qualitative difference between $s = 1$ and $s = 1.1$ can be observed in Figure 18. It is the fact that $s = 1.1$ develops a hump around $c_* \approx 20$ and settles down to its power-law behavior only for much larger values of $c_*$. Due to this feature, the error bars on the data points right above each $\mathbb{Z}_{>0}$ in Figure 17 are underestimated.

Finally, we can analyze the convergence of the full amplitude, which is intrinsically more difficult, because we do not a priori know the exact value. We perform the same analysis as above, except using a 3-parameter non-linear fit:

$$\gamma - \alpha c_*^{-\beta}. \tag{3.45}$$

Here, $\gamma$ is the $c_* \to \infty$ extrapolated value of the amplitude. This quantity, together with its error bars is plotted in Figure 4.

In Figure 19 we plot the exponents $\beta$ obtained by fitting the real part of the amplitude. Overall, uncertainties become much larger due to a more complicated fitting function used. We employ the same procedure as before, except in addition to filtering out by $\bar{R}^2$, we also discard data points for which the relative error on the central value $\gamma$ is bigger than 10%. Once again, we observe that $\beta \approx 0$ as $s \to 0$. For larger $s$, the convergence rate $\beta$ stabilizes to more or less a constant. For a range of values, it is consistent with the "random phase" model that would correspond to $\beta = 2$. Note that systematic errors, just as in the case of Figure 17, are not taken into account.

In the case of Figures 5 and 6, no extrapolation to $c_* \to \infty$ is used. All the points are plotted with $c_* = 10$ with a subset with slightly higher $c_* = 16$ and integer $s$ with $c_* = 1000$. In practice, we found that the result is convergent to a percent level around $c_* \approx 10$ and,

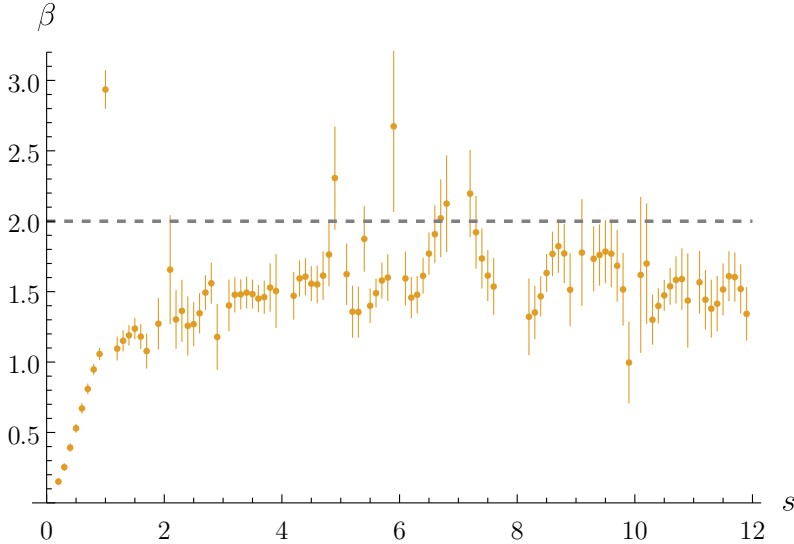

Figure 19: Estimated power-law exponent $\beta$ after fitting the data for the real part to (3.45) for every value of $s$. The value of $\beta = 2$ indicated by the dashed line would correspond to the "random phase" model explained in Section 3.3.1.

especially plotted on a logarithmic scale, using higher cutoffs does not lead to a noticeable difference. The spacing between values of $s$ sampled is $\delta s = 0.01$. The vertical spikes on the plots are caused by the change of the sign of the amplitude.

## 4  Warm-up: Two-point amplitude

Before diving directly into the derivation of the four-point function, we demonstrate some further features of the method at a simpler example, namely a two-point function. Let

$$I(s) = -i \int_\Gamma d\tau \int_0^1 dz \left( \frac{\vartheta_1(z,\tau)}{\eta(\tau)^3} \right)^{2s} , \tag{4.1}$$

for $s > 0$.[5] This is basically the two-point function in bosonic open-string theory. The contour $\Gamma$ runs as before from 0 to $\frac{1}{2}$. Of course, unless $s = 0$, this two-point function is off-shell. This does not prevent us from computing this integral, however. This calculation contains all the the main ideas that go into the computation of the four-point function below, but is still much lower in complexity and thus serves as a good toy example.

Compared to the partition function analyzed using the Rademacher method in Section 3.2, the new aspect we want to learn about from (4.1) is how to deal with branch cuts of the integrand.

### 4.1  Modular transformation

The general logic of the Rademacher contour posits that

$$I(s) = \sum_{c=1}^\infty \sum_{\substack{1 \leqslant a \leqslant \frac{c}{2} \\ (a,c)=1}} \int_{C_{a/c}} d\tau \int_0^1 dz \left( \frac{\vartheta_1(z,\tau)}{\eta(\tau)^3} \right)^{2s} . \tag{4.2}$$

---

[5]For $s \leqslant 0$, the Rademacher procedure does not converge in this case.

The $z$-contour is unaffected by this contour deformation. To compute the integral over the circle $C_{a/c}$, we use the modular properties of the integrand. Notice that

$$\Phi(z,\tau) = \frac{\vartheta_1(z,\tau)^2}{\eta(\tau)^6} \tag{4.3}$$

is a weak Jacobi form of index 1 and weight $-2$, which means that transforms as follows under modular transformations and shifts in $z$:

$$\Phi\left(\frac{z}{c\tau+d}, \frac{a\tau+b}{c\tau+d}\right) = (c\tau+d)^{-2} e^{\frac{2\pi i c z^2}{c\tau+d}} \Phi(z,\tau), \tag{4.4a}$$

$$\Phi(z+m\tau+n,\tau) = e^{-2\pi i(m^2\tau+2mz)} \Phi(z,\tau). \tag{4.4b}$$

Conceptually, these transformation behaviour means that $\Phi(z,\tau)$ is a holomorphic section of a certain line bundle over the moduli space of two-punctured tori $\overline{\mathcal{M}}_{1,2}$. Proceeding as in (3.11), we set

$$\tau = \frac{a\tau'+b}{c\tau'+d}, \tag{4.5}$$

for a new modular parameter $\tau'$. This modular transformation has the property that the line $\tau' \in i + \mathbb{R}$ gets mapped to the circle $C_{a/c}$. We have

$$\Phi(z,\tau) = \Phi\left(\frac{z(c\tau'+d)}{c\tau'+d}, \frac{a\tau'+b}{c\tau'+d}\right) \tag{4.6a}$$

$$= (c\tau'+d)^{-2} e^{2\pi i c(c\tau'+d)z^2} \Phi(z(c\tau'+d), \tau'). \tag{4.6b}$$

Thus we obtain

$$\int_{C_{a/c}} d\tau \int_0^1 dz \left(\frac{\vartheta_1(z,\tau)}{\eta(\tau)^3}\right)^{2s} = -\int_{\longrightarrow} \frac{d\tau'}{(c\tau'+d)^{2+2s}} \int_0^1 dz \, e^{2\pi i c(c\tau'+d)z^2} \left(\frac{\vartheta_1(z(c\tau'+d),\tau')}{\eta(\tau')^3}\right)^{2s}. \tag{4.7}$$

As before, $\longrightarrow$ denotes the contour parallel to the real axis. The minus sign appears because we turned around the orientation of the contour. Since we raised (4.6) to a fractional power, we need to be careful about branch cuts. In the integrand, by $(c\tau'+d)^{2+2s}$ we mean its principal branch.

The correct branch on the right-hand side can be determined by considering the integrand for $z \to 0$. Using that $\vartheta_1'(0,\tau) = 2\pi\eta(\tau)^3$, we have for the integrand before modular transformation:

$$\left(\frac{\vartheta_1(z,\tau)}{\eta(\tau)^3}\right)^{2s} \to (2\pi z)^{2s}, \tag{4.8}$$

and after:

$$\frac{e^{2\pi i c(c\tau'+d)z^2}}{(c\tau'+d)^{2s}} \left(\frac{\vartheta_1(z(c\tau'+d),\tau')}{\eta(\tau')^3}\right)^{2s} \to \frac{(2\pi z(c\tau'+d))^{2s}}{(c\tau'+d)^{2s}} = (2\pi z)^{2s}, \tag{4.9}$$

where we use the principal branch throughout. Since these two expressions agree, we conclude that the branch on the right hand side of (4.7) is specified by taking the principal branch in the region $z \to 0$ and then following the branch smoothly for the other values of $z$. A similar argument could have been applied also for $z \to 1$ and the branch again becomes the principal branch in that region.

Finally, we shift $\tau' \to \tau' - \frac{d}{c}$ since this will be more convenient in the following. We also rename $\tau' \to \tau$ for better readability. Thus, in the end we have

$$\int_{C_{a/c}} d\tau \int_0^1 dz \left(\frac{\vartheta_1(z,\tau)}{\eta(\tau)^3}\right)^{2s} = -\int_{\longrightarrow} \frac{d\tau}{(c\tau)^{2+2s}} q^{sc^2z^2} \left(\frac{\vartheta_1(zc\tau, \tau-\frac{d}{c})}{\eta(\tau-\frac{d}{c})^3}\right)^{2s}, \tag{4.10}$$

where $q = e^{2\pi i\tau}$.

## 4.2 Tropical behaviour

As in the example we explained in Section 3.2, the main trick to evaluate the integral on the right hand side of (4.10) explicitly is to push the horizontal contour up to very high values of $\text{Im}\,\tau$. The result is then only sensitive to the singular behaviour of the integrand.

Let us work out the leading singular behaviour of the integrand, which is controlled by the tropicalization of the integrand. To leading order, the integrand goes like $q^{\text{Trop}}$ as $\text{Im}\,\tau \to \infty$, where the function Trop still depends on the other moduli and kinematics of the problem ($z$ and $s$ in this case). We are interested in the limit $q \to 0$, which is dominated by most negative Trop.

We can work out Trop from the definition of the Jacobi theta function,

$$\vartheta_1(zc\tau, \tau - \tfrac{d}{c}) = -i \sum_{n=-\infty}^{\infty} (-1)^n\, e^{-\frac{\pi i d(2n-1)^2}{4c}}\, q^{\frac{1}{2}(n-\frac{1}{2})^2 - (n-\frac{1}{2})zc}. \tag{4.11}$$

The exponents $\frac{1}{2}(n-\frac{1}{2})^2 - (n-\frac{1}{2})zc$ grow when $n \to \pm\infty$ and thus there is a minimal exponent which controls the behaviour of the theta function near $q \to 0$. The minimum exponent appears for $n = \lfloor cz \rfloor + 1$. Combining this fact with the leading behaviour of the other factors in the integrand, we get the behavior

$$q^{sc^2z^2} q^{2s\left[\frac{1}{2}(\frac{1}{2}+\lfloor cz \rfloor)^2 - (\frac{1}{2}+\lfloor cz \rfloor)cz\right]} q^{-\frac{s}{4}} = q^{\text{Trop}}, \tag{4.12}$$

and hence we conclude

$$\text{Trop} = -s\,\{cz\}(1 - \{cz\}), \tag{4.13}$$

where $\{x\}$ denotes the fractional part, i.e., $\{x\} = x - \lfloor x \rfloor$. This function is periodic with period $\frac{1}{c}$. Moreover, we notice that it vanishes on the boundary of the segments $z \in [\frac{n}{c}, \frac{n+1}{c}]$. It means that the boundaries of these segments do not contribute to the integral, since when $z = \frac{n}{c}$, we can take $\text{Im}\,\tau \to \infty$, which makes the integrand arbitrarily small. This makes it natural to split up the integral into disjoint contributions. Let us set

$$z = \frac{n+\xi}{c}, \tag{4.14}$$

with $n \in \{0, 1, \ldots, c-1\}$ and $\xi \in [0, 1]$.

## 4.3 Branches of $\log\vartheta_1$

To continue, we should carefully discuss the branch of the integrand. It will be sufficient to study

$$\log\vartheta_1\left(cz\tau, \tau - \tfrac{d}{c}\right), \tag{4.15}$$

with $z = \frac{n+\xi}{c}$. Recall that we specified the branch of the integrand by taking the principal branch near $z \to 0$. We want to determine the branch of the logarithm that is obtained by continuously following the branch as we vary $z$ from 0 to our desired value. We claim that for this branch,

$$\log\vartheta_1((n+\xi)\tau, \tau - \tfrac{c}{d}) = -\pi i\tau(n+\xi)^2 + \pi i\tau(\xi - \tfrac{1}{2})^2 + \tfrac{\pi i}{2} - \tfrac{\pi i d}{4c} - 2\pi i\sum_{m=1}^{n}((\tfrac{md}{c}))$$

$$+ \log\prod_{\ell=1}^{\infty}\left(1 - e^{-\frac{2\pi i d\ell}{c}} q^\ell\right)\left(1 - e^{-\frac{2\pi i d(\ell+n)}{c}} q^{\ell-\xi}\right)\left(1 - e^{-\frac{2\pi i d(\ell-n-1)}{c}} q^{\ell+\xi-1}\right). \tag{4.16}$$

Recall the definition of the sawtooth function $((x))$ given in eq. (3.22), which will play a very important role in the following. We use the product representation of the Jacobi theta function

for the argument since it is more convenient. One can easily prove this claim by induction over $n$, as will be done below.

Notice that the function $\log \vartheta_1(cz\tau, \tau - \frac{d}{c})$ has branch points for

$$cz\tau \in \mathbb{Z} + \mathbb{Z}\left(\tau - \frac{d}{c}\right), \tag{4.17}$$

which never lie on the interval $z \in (0,1)$. Thus the choice of branch is independent of $\tau$ and it will be convenient to choose $\text{Im}\,\tau$ very large, i.e., $q = e^{2\pi i\tau}$ small. For $n = 0$, we use the product representation of $\vartheta_1$ to write

$$\log \vartheta_1(\xi\tau, \tau - \frac{c}{d}) = \log\left[ i\, q^{\frac{1}{8} - \frac{\xi}{2}} e^{-\frac{\pi id}{4c}} \prod_{\ell=1}^{\infty} \left(1 - e^{-\frac{2\pi id\ell}{c}} q^{\ell}\right) \right.$$
$$\left. \times \left(1 - e^{-\frac{2\pi id\ell}{c}} q^{\ell - \xi}\right)\left(1 - e^{-\frac{2\pi id(\ell-1)}{c}} q^{\ell + \xi - 1}\right) \right] \tag{4.18}$$

$$= \frac{\pi i}{2} + \frac{\pi i\tau}{4} - \pi i\tau\xi - \frac{\pi id}{4c} + \log\left[ \prod_{\ell=1}^{\infty} \left(1 - e^{-\frac{2\pi id\ell}{c}} q^{\ell}\right) \right.$$
$$\left. \times \left(1 - e^{-\frac{2\pi id\ell}{c}} q^{\ell - \xi}\right)\left(1 - e^{-\frac{2\pi id(\ell-1)}{c}} q^{\ell + \xi - 1}\right) \right]. \tag{4.19}$$

In the last line, we chose the principal branch for $\text{Re}\,\tau = \frac{d}{c}$ and large $\text{Im}\,\tau$, since this was our prescription for to determine the branch.[6]

We next discuss the induction step $n \to n+1$. We can start with the formula (4.16) for $n$ and then smoothly take $\xi$ from the range $[0,1]$ to the range $[1,2]$. We can discuss every factor in the infinite product separately. Since we are taking $q$ very small, the only dangerous factors are those where the exponent of $q$ can become less than zero, which only happens for the second term in the infinite product for $\ell = 1$. Thus it is enough to discuss the branch of

$$\log\left(1 - e^{2\pi i\varphi} q^{1-\xi}\right), \tag{4.20}$$

for a phase $\varphi$. For $\xi \in [0,1]$, the choice of branch in the function is clear. For $\xi > 1$, the second term dominates. The correct branch is obtained by the following consideration. First note that for $\varphi = \frac{1}{2}$, the branch is trivial to choose and we have

$$\log(1 + q^{1-\xi}) = 2\pi i\tau(1-\xi) + \log(1 + q^{\xi-1}). \tag{4.21}$$

For arbitrary $\varphi$, we have

$$\log(1 - e^{2\pi i\varphi} q^{1-\xi}) = 2\pi i\tau(1-\xi) + \pi i + 2\pi i\varphi + 2\pi ik + \log(1 + e^{-2\pi i\varphi} q^{\xi-1}), \tag{4.22}$$

for some integer $k$. Finally, we use that the phase can only jump for integer $\varphi$, since then the contour in $\xi$ hits the branch point. Thus the correct branch is

$$\log(1 - e^{2\pi i\varphi} q^{1-\xi}) = 2\pi i\tau(1-\xi) + 2\pi i((\varphi)) + \log(1 + e^{-2\pi i\varphi} q^{\xi-1}), \tag{4.23}$$

where we defined the sawtooth function $((\varphi))$ in (3.22). Applying this identity to (4.16) with $\varphi = -\frac{d(\ell+1)}{c}$ gives

$$\log \vartheta_1((n+\xi)\tau, \tau - \frac{c}{d})$$
$$= -\pi i\tau(n+\xi)^2 + \pi i\tau(\xi - \frac{1}{2})^2 + 2\pi i\tau(1-\xi) - \frac{\pi i}{2} - \frac{\pi id}{4c} - 2\pi i \sum_{m=1}^{n+1} ((\tfrac{md}{c}))$$

---

[6]The choice $\text{Re}\,\tau = \frac{d}{c}$ is not necessary if we look at the ratio $\frac{\vartheta_1(z,\tau)}{\eta(\tau)^3}$ since the leading exponent $q^{\frac{1}{8}}$ cancels out.

$$+\log\Big[\prod_{\ell=1}^{\infty}\big(1-e^{-\frac{2\pi id\ell}{c}}q^{\ell}\big)\prod_{\ell=2}^{\infty}\big(1-e^{-\frac{2\pi id(\ell+n)}{c}}q^{\ell-\xi}\big)$$

$$\times\big(1-e^{\frac{2\pi id(1+n)}{c}}q^{\xi-1}\big)\prod_{\ell=1}^{\infty}\big(1-e^{-\frac{2\pi id(\ell-n-1)}{c}}q^{\ell+\xi-1}\big)\Big],\qquad(4.24)$$

and after further massaging:

$$\log\vartheta_1((n+\xi)\tau,\tau-\tfrac{c}{d})=-\pi i\tau(n+\xi)^2+\pi i\tau(\xi-\tfrac{3}{2})^2-\tfrac{\pi i}{2}-\tfrac{\pi id}{4c}-2\pi i\sum_{m=1}^{n+1}(\!(\tfrac{md}{c})\!)$$

$$+\log\Big[\prod_{\ell=1}^{\infty}\big(1-e^{-\frac{2\pi id\ell}{c}}q^{\ell}\big)\big(1-e^{-\frac{2\pi id(\ell+n+1)}{c}}q^{\ell+1-\xi}\big)\big(1-e^{-\frac{2\pi id(\ell-n-2)}{c}}q^{\ell+\xi-2}\big)\Big].\quad(4.25)$$

This is the claimed expression expression for $\log\vartheta_1((n+1+\xi)\tau,\tau-\tfrac{c}{d})$, but with $\xi$ replaced with $\xi-1$, showing that (4.16) is the correct branch for the logarithm.

For future reference, let us note that

$$\sum_{m=1}^{c-1}\Big(\!\Big(\frac{md}{c}\Big)\!\Big)=0.\qquad(4.26)$$

This is because we are summing over all non-zero elements of the ring $\mathbb{Z}_c$. They are paired up as $md$ and $-md$ and hence cancel out pairwise thanks to $(\!(\tfrac{md}{c})\!)+(\!(-\tfrac{md}{c})\!)=0$. It continues to hold if $md=-md$ mod $c$, since then $\tfrac{md}{c}\equiv\tfrac{1}{2}$ and hence $(\!(\tfrac{md}{c})\!)=0$. This means that the phase on the last segment $\tfrac{c-1}{c}<z<1$ is again trivial. This had to happen because we can either use $z\to0$ or $z\to1$ to fix the branch as described above.

## 4.4 Thresholds from $q$-expansion

Let us now insert the branch (4.16) into (4.10). We get

$$\int_{C_{a/c}}d\tau\int_0^1 dz\left(\frac{\vartheta_1(z,\tau)}{\eta(\tau)^3}\right)^{2s}=\sum_{n=0}^{c-1}\int_{\longrightarrow}\frac{d\tau}{c(-ic\tau)^{2+2s}}\int_0^1 d\xi\; q^{s\xi(\xi-1)}e^{-4\pi is\sum_{m=1}^{n}(\!(\frac{md}{c})\!)}$$

$$\times\prod_{\ell=1}^{\infty}\frac{\big(1-e^{-\frac{2\pi id(\ell+n)}{c}}q^{\ell-\xi}\big)^{2s}\big(1-e^{-\frac{2\pi id(\ell-n-1)}{c}}q^{\ell+\xi-1}\big)^{2s}}{\big(1-e^{-\frac{2\pi id\ell}{c}}q^{\ell}\big)^{4s}}.\quad(4.27)$$

We absorbed the phase $e^{\pi is}$ and the overall minus sign into the denominator. We also get another factor $\tfrac{1}{c}$ from the Jacobian of the change of variables (4.14).

Let us $q$-expand the integrand. There are only finitely many terms in the $q$-expansion that can potentially contribute to the integral. Indeed, every term in the $q$-expansion is of the form $q^{\mathrm{Trop}_{m_D,m_U}}$ with

$$\mathrm{Trop}_{m_D,m_U}=s\xi(\xi-1)+m_D\xi+m_U(1-\xi),\qquad(4.28)$$

and $m_D,m_U$ to non-negative integers. A term of this form can only contribute when $\mathrm{Trop}_{m_D,m_U}<0$ for some choice of $\xi\in[0,1]$. $\mathrm{Trop}_{m_D,m_U}$ attains its minimum for

$$\xi_{\min}=\frac{m_U-m_D+s}{2s},\qquad(4.29)$$

which lies inside the unit interval for $|m_U-m_D|\leqslant s$. In the other case where $|m_D-m_U|\geqslant s$, the minimum of $\mathrm{Trop}_{m_D,m_U}$ is attained on the boundary of the interval, where $\mathrm{Trop}_{m_D,m_U}$ is always

non-negative. Thus only $(m_D, m_U)$ with $|m_D - m_U| \leqslant s$ can potentially be negative somewhere on the unit interval. In this case we have

$$\min_{\xi \in [0,1]} \text{Trop}_{m_D, m_U} = -\frac{[s - (\sqrt{m_D} + \sqrt{m_U})^2][s - (\sqrt{m_D} - \sqrt{m_U})^2]}{4s}. \tag{4.30}$$

This expression is negative and hence contributes to the integral when either

$$s \geqslant (\sqrt{m_D} + \sqrt{m_U})^2, \quad \text{or} \quad s \leqslant (\sqrt{m_D} - \sqrt{m_U})^2. \tag{4.31}$$

Since

$$s \geqslant |m_D - m_U| = |\sqrt{m_D} - \sqrt{m_U}|(\sqrt{m_D} + \sqrt{m_U}) \geqslant |\sqrt{m_D} - \sqrt{m_U}|^2, \tag{4.32}$$

the latter is incompatible with the assumption $s \geqslant |m_D - m_U|$. Thus only terms with $s \geqslant (\sqrt{m_D} + \sqrt{m_U})^2$ contribute to the integral. This is precisely how thresholds manifest themselves.

Assuming now that $s \geqslant (\sqrt{m_D} + \sqrt{m_U})^2$, we may extend the integration region $\xi \in [0, 1]$ to $\xi \in \mathbb{R}$, since the exponent $\text{Trop}_{m_D, m_U}$ is positive outside of the interval $[0, 1]$ and hence the region outside of the unit interval also leads to a vanishing contribution for the integral. This extension of the integration contour will simplify the analysis later on.

Next, let us note that the phase of a term of the form $q^{m_D \xi + m_U(1-\xi)}$ appearing in the $q$-expansion of the infinite product in (4.27) is given by $e^{\frac{2\pi i d}{c}(n m_D - (n+1)m_U)}$. This becomes obvious if we set $q_D = q^\xi$ and $q_U = q^{1-\xi}$, so that $q_D q_U = q$. We can then write terms appearing in (4.27) as

$$e^{-\frac{2\pi i d(\ell+n)}{c}} q^{\ell - \xi} = e^{\frac{2\pi i d}{c}(n(\ell-1)-(n+1)\ell)} q_D^{\ell-1} q_U^\ell, \tag{4.33a}$$

$$e^{-\frac{2\pi i d(\ell-n-1)}{c}} q^{\ell + \xi - 1} = e^{\frac{2\pi i d}{c}(n\ell - (n+1)(\ell-1))} q_D^\ell q_U^{\ell-1}, \tag{4.33b}$$

$$e^{-\frac{2\pi i d \ell}{c}} q^\ell = e^{\frac{2\pi i d}{c}(n\ell - (n+1)\ell)} q_D^\ell q_U^\ell. \tag{4.33c}$$

We can thus write

$$\prod_{\ell=1}^\infty \frac{\left(1 - e^{-\frac{2\pi i d(\ell+n)}{c}} q^{\ell-\xi}\right)^{2s} \left(1 - e^{-\frac{2\pi i d(\ell-n-1)}{c}} q^{\ell+\xi-1}\right)^{2s}}{\left(1 - e^{-\frac{2\pi i d \ell}{c}} q^\ell\right)^{4s}}$$

$$= \sum_{m_D, m_U = 0}^\infty Q_{m_D, m_U}^{(2)}(s) \, e^{\frac{2\pi i d}{c}(n m_D - (n+1)m_U)} q^{m_D \xi + m_U(1-\xi)}, \tag{4.34}$$

with

$$Q_{m_D, m_U}^{(2)}(s) = [q_D^{m_D} q_U^{m_U}] \prod_{\ell=1}^\infty \frac{(1 - q_D^{\ell-1} q_U^\ell)^{2s}(1 - q_D^\ell q_U^{\ell-1})^{2s}}{(1 - q_D^\ell q_U^\ell)^{4s}}. \tag{4.35}$$

The superscript (2) is intended to distinguish these coefficients from similar coefficients appearing in the four-point function case. We can insert this expansion in (4.27) to get

$$\int_{C_{a/c}} d\tau \int_0^1 dz \left(\frac{\vartheta_1(z,\tau)}{\eta(\tau)^3}\right)^{2s} = \sum_{n=0}^{c-1} \sum_{\substack{m_D, m_U \geqslant 0 \\ (\sqrt{m_D} + \sqrt{m_U})^2 \leqslant s}} e^{-4\pi i s \sum_{m=1}^n ((\frac{md}{c})) + \frac{2\pi i(n m_D - (n+1)m_U)d}{c}}$$

$$\times Q_{m_D, m_U}^{(2)}(s) \int_{\longrightarrow} \frac{d\tau}{c(-ic\tau)^{2+2s}} \int_{-\infty}^\infty d\xi \, q^{s\xi(\xi-1)+m_D\xi+m_U(1-\xi)}. \tag{4.36}$$

## 4.5 Assembling the result

The integral over $\xi$ is Gaussian and simple to perform. Let us denote

$$\Delta_{m_{\mathrm{D}},m_{\mathrm{U}}}(s) = [s - (\sqrt{m_{\mathrm{D}}} + \sqrt{m_{\mathrm{U}}})^2][s - (\sqrt{m_{\mathrm{D}}} - \sqrt{m_{\mathrm{U}}})^2]. \tag{4.37}$$

Evaluating the Gaussian integral leads to

$$\int_{C_{a/c}} \mathrm{d}\tau \int_0^1 \mathrm{d}z \left( \frac{\vartheta_1(z,\tau)}{\eta(\tau)^3} \right)^{2s} = \sum_{\substack{m_{\mathrm{D}},m_{\mathrm{U}} \geqslant 0 \\ (\sqrt{m_{\mathrm{D}}}+\sqrt{m_{\mathrm{U}}})^2 \leqslant s}} \sum_{n=0}^{c-1} \mathrm{e}^{-4\pi i s \sum_{m=1}^{n}(\!(\frac{md}{c})\!) + \frac{2\pi i (nm_{\mathrm{D}}-(n+1)m_{\mathrm{U}})d}{c}}$$

$$\times Q^{(2)}_{m_{\mathrm{D}},m_{\mathrm{U}}}(s) \int_{\longrightarrow} \frac{\mathrm{d}\tau}{(-i\tau)^{\frac{5}{2}+2s} c^{3+2s} \sqrt{2s}} \, q^{-\frac{\Delta_{m_{\mathrm{D}},m_{\mathrm{U}}}}{4s}}. \tag{4.38}$$

The integral over $\tau$ can be computed exactly. Let us consider

$$\int_{\longrightarrow} \frac{\mathrm{d}\tau}{(-i\tau)^z} \mathrm{e}^{-2\pi i \tau a}, \tag{4.39}$$

for an arbitrary positive parameter $a$ and arbitrary exponent $z$. Changing the variables to $x = 2\pi i \tau a$ gives

$$\int_{\longrightarrow} \frac{\mathrm{d}\tau}{(-i\tau)^z} \mathrm{e}^{-2\pi i \tau a} = -i(2\pi a)^{z-1} \int_{\uparrow} \mathrm{d}x \, (-x)^{-z} \mathrm{e}^{-x}. \tag{4.40}$$

Upon performing the change of variables from $\tau$ to $x$, the contour gets rotated by 90 degrees and now runs upwards, which we signified by the arrow $\uparrow$. We can deform this contour into the Hankel contour $\mathcal{H}$ which runs from $\infty + i\varepsilon$ to $\infty - i\varepsilon$ by surrounding whole branch cut going from $x = 0$ to $x = \infty$ along the real axis. We then use the Hankel representation of the Gamma function,

$$\int_{\mathcal{H}} \mathrm{d}x \, (-x)^{-z} \mathrm{e}^{-x} = -\frac{2\pi i}{\Gamma(z)}. \tag{4.41}$$

Therefore, we find

$$\int_{\longrightarrow} \frac{\mathrm{d}\tau}{(-i\tau)^z} \mathrm{e}^{-2\pi i \tau a} = i(2\pi a)^{z-1} \int_{\mathcal{H}} \mathrm{d}x \, (-x)^{-z} \mathrm{e}^{-x} = \frac{2\pi(2\pi a)^{z-1}}{\Gamma(z)}. \tag{4.42}$$

We can thus finish the computation as follows,

$$\int_{C_{a/c}} \mathrm{d}\tau \int_0^1 \mathrm{d}z \left( \frac{\vartheta_1(z,\tau)}{\eta(\tau)^3} \right)^{2s} = \frac{\pi^{\frac{5}{2}+2s}}{2^{1+2s} s^{2+2s} c^{3+2s} \Gamma(\frac{5}{2}+2s)}$$

$$\times \sum_{\substack{m_{\mathrm{D}},m_{\mathrm{U}} \geqslant 0 \\ (\sqrt{m_{\mathrm{D}}}+\sqrt{m_{\mathrm{U}}})^2 \leqslant s}} Q^{(2)}_{m_{\mathrm{D}},m_{\mathrm{U}}}(s) \Delta^{\frac{3}{2}+2s}_{m_{\mathrm{D}},m_{\mathrm{U}}}(s) \sum_{n=0}^{c-1} \mathrm{e}^{-4\pi i s \sum_{m=1}^{n}(\!(\frac{md}{c})\!) + \frac{2\pi i (nm_{\mathrm{D}}-(n+1)m_{\mathrm{U}})d}{c}}. \tag{4.43}$$

We can finally assemble all the circles of the Rademacher contour to obtain the final result for the integral (4.1),

$$I(s) = -i \sum_{c=1}^{\infty} \frac{\pi^{\frac{5}{2}+2s}}{2^{1+2s} s^{2+2s} c^{3+2s} \Gamma(\frac{5}{2}+2s)} \sum_{\substack{1 \leqslant a \leqslant \frac{c}{2} \\ (a,c)=1}} \sum_{\substack{m_{\mathrm{D}},m_{\mathrm{U}} \geqslant 0 \\ (\sqrt{m_{\mathrm{D}}}+\sqrt{m_{\mathrm{U}}})^2 \leqslant s}} Q^{(2)}_{m_{\mathrm{D}},m_{\mathrm{U}}}(s) \Delta^{\frac{3}{2}+2s}_{m_{\mathrm{D}},m_{\mathrm{U}}}(s)$$

$$\times \sum_{n=0}^{c-1} \mathrm{e}^{-4\pi i s \sum_{m=1}^{n}(\!(\frac{md}{c})\!) + \frac{2\pi i (nm_{\mathrm{D}}-(n+1)m_{\mathrm{U}})d}{c}}. \tag{4.44}$$

As usual, $d$ denotes the inverse of $a$ mod $c$. This is our final result for the two-point function.

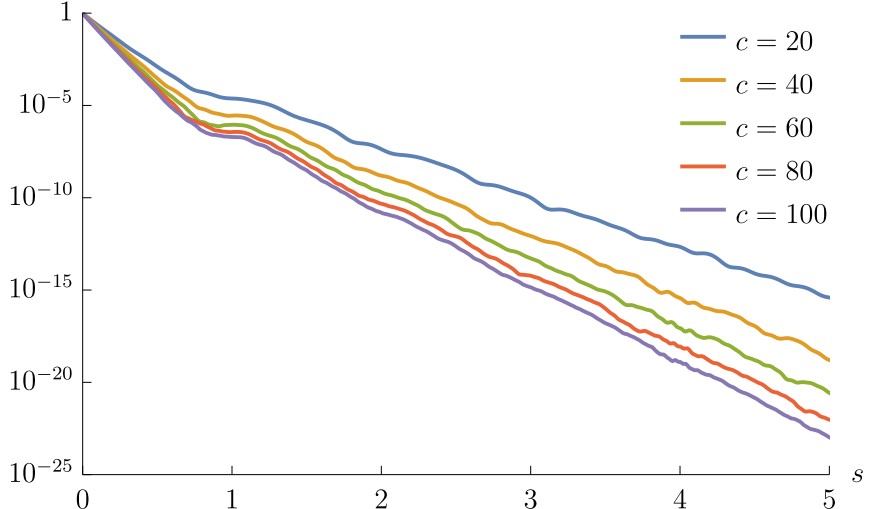

Figure 20: Convergence of the Rademacher method for the two-point function. We plot the relative error (4.47) on the $y$-axis, which decreases with larger cutoffs $c$ for any $s$.

## 4.6 Cross-checks

Given the amount of non-trivial manipulations that went into this computation, it would be reassuring to perform a stress test. We explain now such a check. Let us change the computation slightly and compute instead the integral over the contour in (4.1) that runs from 0 to 1 (instead of 0 to $\frac{1}{2}$). Let us denote the corresponding result by $\tilde{I}(s)$. The Rademacher logic still applies to this integral and (4.44) still holds, except that the summation range over $a$ gets extended to $1 \leqslant a \leqslant c$, reflecting the fact that we now need to use all the circles up to 1 in the contour.

The integral $\tilde{I}(s)$ over the new contour is very simple to evaluate analytically. Indeed, since the integrand is periodic in $\tau \to \tau + 1$, we are simply extracting the leading Fourier coefficient in $\tau$. Since

$$\frac{\vartheta_1(z, \tau)}{\eta(\tau)^3} \xrightarrow{\text{Im}\,\tau \to \infty} 2\sin(\pi z), \tag{4.45}$$

the constant Fourier coefficient of the integrand is $(2\sin(\pi z))^{2s}$. We thus get

$$\tilde{I}(s) = -i \int_0^1 \mathrm{d}z\, (2\sin(\pi z))^{2s} = -i\, \frac{4^s\, \Gamma(s + \frac{1}{2})}{\sqrt{\pi}\, \Gamma(s + 1)}. \tag{4.46}$$

We can easily check numerically whether this equals (4.44) with extended range over $a$. We plot in Figure 20 the quantity

$$\left| \frac{(4.44) \text{ with extended range } 1 \leqslant a \leqslant c}{(4.46)} - 1 \right|, \tag{4.47}$$

in the interval $0 \leqslant s \leqslant 5$ for larger and larger cutoffs $c$. Clearly, the error goes to zero as $c$ grows. Moreover, thanks to the presence of the factor $\frac{1}{c^{3+2s}}$ in (4.44), convergence is much faster for large values of $s$. This check nicely tests the whole formula.

As another remark, we note that the limit $s \to 0$ is rather subtle. The exact answer (4.46) clearly goes to $-i$ as $s \to 0$. On the other hand, (4.44) for fixed $c$ in goes to zero even if we extend the range to $1 \leqslant a \leqslant c$. This means that we are not allowed to commute the limit with the infinite sum over $c$ in (4.44). We can see this explicitly by plotting (4.44) (with extended

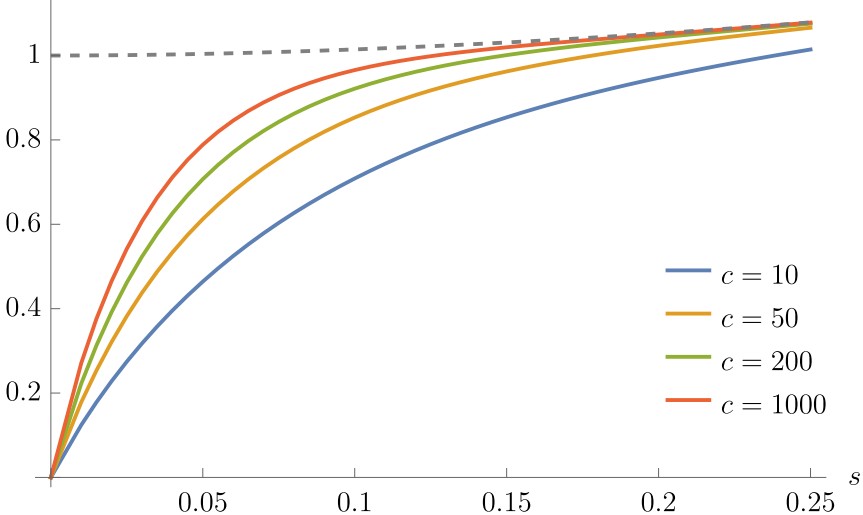

Figure 21: The behaviour of the Rademacher formula near $s = 0$. The gray dashed line is the exact answer (4.46) (multiplied by $i$ to make it real), while the different contours are the answer obtained form (4.44) when truncating the sum at different maximal values of $c$.

range $1 \leqslant a \leqslant c$) for different cutoffs $c$ near $s = 0$. The results are plotted in Figure 21. The curve obtained from the Rademacher method converges everywhere to the exact answer, except at $s = 0$.

## 5 Four-point planar amplitude $A^{\mathrm{p}}(s, t)$

We now derive the equation (3.25) for the planar amplitude in the $s$-channel. Many of the steps performed here are analogous to the steps for the two-point function and thus we keep the discussion of these steps more brief. The formula for the amplitude in the $u$-channel will turn out to be similar as well.

### 5.1 Integrand on the Ford circle $C_{a/c}$

We focus on the contribution of the Ford circle $C_{a/c}$ to the planar amplitude given by the integral (3.18). Let us call it $A^{\mathrm{p}}_{a/c}$. The full planar amplitude $A^{\mathrm{p}}$ is the sum of $A^{\mathrm{p}}_{a/c}$ for all irreducible fractions $\frac{a}{c}$ plus the cusp contribution $\Delta A^{\mathrm{p}}$. As in our toy example above, we want to perform the change of variables

$$\tau = \frac{a\tau' + b}{c\tau' + d}. \tag{5.1}$$

Using the modular behaviour of the theta-functions under, we have

$$A^{\mathrm{p}}_{a/c} = i \int_{\longrightarrow} \frac{\mathrm{d}\tau}{c^2\tau^2} q^{c^2 s z_{41} z_{32} - c^2 t z_{21} z_{43}} \left( \frac{\vartheta_1(z_{21} c\tau, \tau - \frac{d}{c}) \vartheta_1(z_{43} c\tau, \tau - \frac{d}{c})}{\vartheta_1(z_{31} c\tau, \tau - \frac{d}{c}) \vartheta_1(z_{42} c\tau, \tau - \frac{d}{c})} \right)^{-s}$$

$$\times \left( \frac{\vartheta_1(z_{32} c\tau, \tau - \frac{d}{c}) \vartheta_1(z_{41} c\tau, \tau - \frac{d}{c})}{\vartheta_1(z_{31} c\tau, \tau - \frac{d}{c}) \vartheta_1(z_{42} c\tau, \tau - \frac{d}{c})} \right)^{-t}. \tag{5.2}$$

Here we renamed $\tau' \to \tau - \frac{d}{c}$.[7] We also set $q = e^{2\pi i \tau}$. The overall sign changes again due to the choice of orientation for the contour. The branch of the right-hand side is determined as follows.

We first note that the integrand of the original integral (3.6) simplifies for small $z_{ij}$ to

$$\left(\frac{\vartheta_1(z_{21},\tau)\vartheta_1(z_{43},\tau)}{\vartheta_1(z_{31},\tau)\vartheta_1(z_{42},\tau)}\right)^{-s}\left(\frac{\vartheta_1(z_{32},\tau)\vartheta_1(z_{41},\tau)}{\vartheta_1(z_{31},\tau)\vartheta_1(z_{42},\tau)}\right)^{-t} \to \left(\frac{z_{21}z_{43}}{z_{31}z_{42}}\right)^{-s}\left(\frac{z_{32}z_{41}}{z_{31}z_{42}}\right)^{-t}, \qquad (5.3)$$

independently of $\tau$. This is compatible with the leading behaviour of the integrand (5.2) as $z_{ij} \to 0$. Thus we take the principal branch in (5.2) for small $z_{ij}$ and then follow the branch smoothly when varying $z_i$.

## 5.2 Tropicalization

We now want to again want to push the $\tau$ contour to large values of $\text{Im}\,\tau$. The leading behaviour is controlled by the function Trop, which appears as the leading exponent $q^{\text{Trop}}$ as $q \to 0$. It is given by

$$\text{Trop} = \frac{1}{2}\sum_{i>j}s_{ij}\{cz_{ij}\}(1-\{cz_{ij}\}), \qquad (5.4)$$

where we remind the reader that $\{x\}$ denotes the fractional part of $x$. We are again interested in the region with Trop $< 0$, since the regions with Trop $> 0$ give a vanishing contribution to the integral when we take the limit $\text{Im}\,\tau \to \infty$.

Clearly, Trop is a periodic function with period $\frac{1}{c}$ in all $z_i$'s. As a consequence, the regions where Trop $< 0$ will come in families, since we can always translate a region with Trop $< 0$ by a multiple of $\frac{1}{c}$ in the $z_i$'s to obtain a new region with Trop $< 0$. For example, the subregion of the parameter space $(z_1, z_2, z_3)$ with Trop $< 0$ in the $s$-channel is depicted in Figure 22. Recall that we fix $z_4 = 1$.

There are in total $\frac{1}{6}m(m+1)(m+2)$ regions in the $z_i$-parameter space, where Trop $< 0$. We will label them as $\Gamma_{n_1,n_2,n_3}$ with $1 \leqslant n_1 \leqslant n_2 \leqslant n_3 \leqslant c$. Each such $\Gamma_{n_1,n_2,n_3}$ is fully contained in the following region

$$\frac{n_{ij}-1}{c} \leqslant z_{ij} \leqslant \frac{n_{ij}+1}{c}, \qquad ij \in \{21, 43\}, \qquad (5.5a)$$

$$\frac{n_{ij}}{c} \leqslant z_{ij} \leqslant \frac{n_{ij}+1}{c}, \qquad ij \in \{31, 41, 32, 42\}. \qquad (5.5b)$$

Here, $n_{ij} \equiv n_i - n_j$ and $n_4 \equiv c$. We denote the contribution from the region $\Gamma_{n_1,n_2,n_3}$ by $A_{a/c}^{n_1,n_2,n_3}$, so that

$$A_{a/c}^{\text{p}} = \sum_{1\leqslant n_1\leqslant n_2\leqslant n_3\leqslant c} A_{a/c}^{n_1,n_2,n_3}. \qquad (5.6)$$

We then set

$$z_i = \frac{n_i + \xi_i}{c}, \qquad (5.7)$$

on each of the individual regions so that the integration range of $\xi_i$ is always the same in each region.

---

[7]The original $\tau$ will not appear again and hopefully this does not lead to confusions.

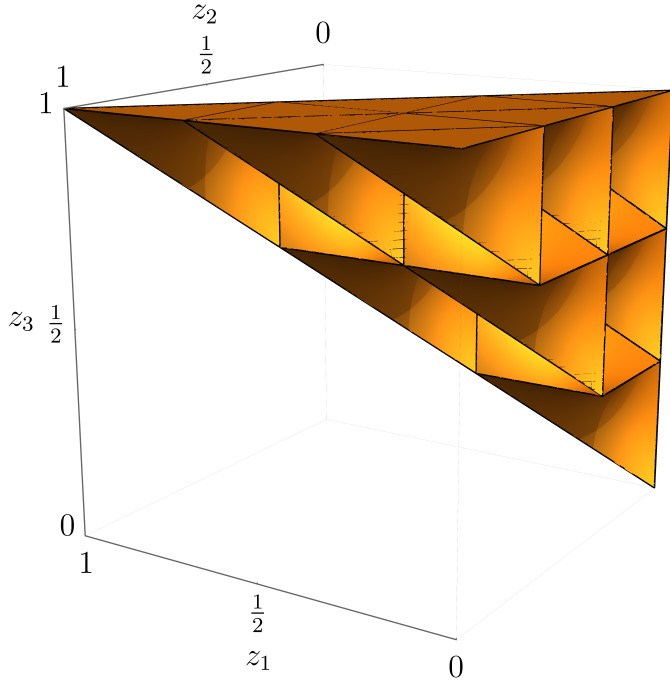

Figure 22: The regions $\Gamma_{n_1,n_2,n_3}$ in parameter space where Trop $< 0$ for $c = 3$.

## 5.3 Contributions with Trop $< 0$

We can determine the correct branch of the Jacobi theta functions raised to the powers of $s$ or $t$ by the same logic as for the two-point function. Inserting eq. (4.16) for the correct branch gives immediately

$$
A_{a/c}^{n_1,n_2,n_3} = i \int_{\longrightarrow} \frac{\mathrm{d}\tau}{c^5\tau^2} \int \mathrm{d}\xi_1 \, \mathrm{d}\xi_2 \, \mathrm{d}\xi_3 \prod_{i>j} q^{-\frac{1}{2}s_{ij}\xi_{ij}(\xi_{ij}-1)} e^{2\pi i s_{ij} \sum_{m=1}^{n_{ij}} (\!(\frac{md}{c})\!)}
$$
$$
\times \prod_{\ell=1}^{\infty} (1 - e^{-\frac{2\pi i d(\ell+n_{ij})}{c}} q^{\ell-\xi_{ij}})^{-s_{ij}} (1 - e^{-\frac{2\pi i d(\ell-n_{ij}-1)}{c}} q^{\ell+\xi_{ij}-1})^{-s_{ij}} . \tag{5.8}
$$

The integration region over the $\xi_i$'s is such that both the inequalities (5.5a) and $0 \leqslant z_1 \leqslant z_2 \leqslant z_3 \leqslant 1$ are satisfied. This means that for a generic region $\Gamma_{n_1,n_2,n_3}$ we have

$$
-1 \leqslant \xi_{21}, \xi_{43} \leqslant 1, \qquad 0 \leqslant \xi_{31}, \xi_{32}, \xi_{41}, \xi_{42} \leqslant 1 . \tag{5.9}
$$

For the regions with $n_{21} = 0$, we have the smaller integration region where $\xi_{21} \geqslant 0$ should be imposed. Similarly when $n_{43} = 0$, the integration region is restricted by $\xi_{43} \geqslant 0$. The branch in this formula is defined for $\xi_{ij} > 0$, where the constant factor in the infinite product dominates for small $q$.

## 5.4 Thresholds from the $q$-expansion

As a next step, we $q$-expand the integrand in (5.8). As for the two-point function, it will turn out that for a given $s$, there are only finitely many terms that contribute to the integral. We have to be careful with the two factors $(1 - e^{\frac{2\pi i d n_{21}}{c}} q^{\xi_{21}})^{-s}$ and $(1 - e^{\frac{2\pi i d n_{43}}{c}} q^{\xi_{43}})^{-s}$ that are present in the infinite product in (5.8). Since $\xi_{21}$ and $\xi_{43}$ are allowed to go to zero in the integration region, we are not allowed to $q$-expand these factors, but have to leave them unexpanded.

For the purpose of analyzing which term can dominate where, we notice that any term appearing in the $q$-expansion is of the form

$$q^{-\frac{1}{2}\sum_{i>j}s_{ij}\xi_{ij}(\xi_{ij}-1)+m_{\mathrm{L}}\xi_{21}+m_{\mathrm{D}}\xi_{32}+m_{\mathrm{R}}\xi_{43}+m_{\mathrm{U}}(1-\xi_{41})}(1-e^{\frac{2\pi i d n_{21}}{c}}q^{\xi_{21}})^{-s}(1-e^{\frac{2\pi i d n_{43}}{c}}q^{\xi_{43}})^{-s}, \quad (5.10)$$

for four non-negative integers that we denote by $m_{\mathrm{L}}$, $m_{\mathrm{D}}$, $m_{\mathrm{R}}$, and $m_{\mathrm{U}}$. The names indicate that they play the role of the (square of the) internal masses on the left, bottom, right and top part of a box Feynman diagram that approximates the worldsheet. It is easy to see that these integers satisfy the condition

$$0 \leqslant m_{\mathrm{L}}, m_{\mathrm{R}} \leqslant m_{\mathrm{D}} + m_{\mathrm{U}}. \quad (5.11)$$

Let us work out the contribution from such a term to $A_{a/c}^{n_1,n_2,n_3}$. We first consider the leading exponent as $q \to 0$

$$\mathrm{Trop}_{m_{\mathrm{L}},m_{\mathrm{D}},m_{\mathrm{R}},m_{\mathrm{U}}} = -\frac{1}{2}\sum_{i>j}s_{ij}\xi_{ij}(\xi_{ij}-1) + \left(\begin{cases} m_{\mathrm{L}}\xi_{21} & \text{if} \quad \xi_{21}>0 \\ (m_{\mathrm{L}}-s)\xi_{21} & \text{if} \quad \xi_{21}<0 \end{cases}\right) + m_{\mathrm{D}}\xi_{32}$$
$$+ \left(\begin{cases} m_{\mathrm{R}}\xi_{43} & \text{if} \quad \xi_{43}>0 \\ (m_{\mathrm{R}}-s)\xi_{43} & \text{if} \quad \xi_{43}<0 \end{cases}\right) + m_{\mathrm{U}}(1-\xi_{41}). \quad (5.12)$$

The term contributes to the integral if $\mathrm{Trop}_{m_{\mathrm{L}},m_{\mathrm{D}},m_{\mathrm{R}},m_{\mathrm{U}}}$ becomes negative somewhere on the integration region. A straightforward analysis shows that $\mathrm{Trop}_{m_{\mathrm{L}},m_{\mathrm{D}},m_{\mathrm{R}},m_{\mathrm{U}}}$ attains its minimum at $\xi_{21} = 0$ and $\xi_{43} = 0$ (but is not differentiable there). Thus it suffices to restrict $\mathrm{Trop}_{m_{\mathrm{L}},m_{\mathrm{D}},m_{\mathrm{R}},m_{\mathrm{U}}}$ to this special case and analyze where it is negative. We have, setting $\xi_3 \equiv \xi$ and $\xi_1 = 0$,

$$\mathrm{Trop}_{m_{\mathrm{L}},m_{\mathrm{D}},m_{\mathrm{R}},m_{\mathrm{U}}}\Big|_{\substack{\xi_{21}=0\\\xi_{43}=0}} = s\xi(\xi-1) + m_{\mathrm{D}}\xi + m_{\mathrm{U}}(1-\xi), \quad (5.13)$$

which coincides with the Trop function in the two-point function case, see eq. (4.28).

Let us remark that this is not surprising from a field theory point of view. The $\xi_i$'s play the role of Schwinger parameters and taking $\xi_{21} \to 0$, $\xi_{43} \to 0$ essentially reduces the diagram to a bubble diagram with masses squared $m_{\mathrm{D}}$ and $m_{\mathrm{U}}$, which is what we analyzed in Section 4. Thus the same conclusion as there holds and a term of the form (5.10) only contributes to the amplitude for $s \geqslant (\sqrt{m_{\mathrm{D}}} + \sqrt{m_{\mathrm{U}}})^2$.

## 5.5 Evaluating a single term in the $q$-expansion

We now focus on a single term in the $q$-expansion of the form (5.10). Evaluating such a term is the only essentially new ingredient not present in the two-point function analysis of Section 4. Let us change variables as follows

$$\xi_{21} = \alpha_{\mathrm{L}}, \qquad \xi_{43} = \alpha_{\mathrm{R}}, \qquad \xi_{31} = \frac{1}{s}(-m_{\mathrm{D}} + s + t_{\mathrm{L}} + u\alpha_{\mathrm{R}}). \quad (5.14)$$

We also integrate in the unity

$$1 = \sqrt{\frac{-is\tau}{2tu}}\int dt_{\mathrm{R}}\, q^{\frac{1}{4stu}(st_{\mathrm{R}}-(s+2t)t_{\mathrm{L}}-2tu\alpha_{\mathrm{R}}+(m_{\mathrm{D}}+m_{\mathrm{U}})t-st)^2}. \quad (5.15)$$

This yields the following contribution from a single term (5.10),

$$i\int_{\longrightarrow}\frac{d\tau}{c^5\tau^2}\int d\xi_1\, d\xi_2\, d\xi_3\, q^{-\frac{1}{2}\sum_{i>j}s_{ij}\xi_{ij}(\xi_{ij}-1)+m_{\mathrm{L}}\xi_{21}+m_{\mathrm{D}}\xi_{32}+m_{\mathrm{R}}\xi_{43}+m_{\mathrm{U}}(1-\xi_{41})}$$

$$\times (1 - e^{\frac{2\pi i d n_{21}}{c}} q^{\xi_{21}})^{-s} (1 - e^{\frac{2\pi i d n_{43}}{c}} q^{\xi_{43}})^{-s}$$

$$= i \int_{\to} \frac{d\tau}{c^5 \tau^2} \sqrt{\frac{-i\tau}{2stu}} \int dt_L \, dt_R \, d\alpha_L \, d\alpha_R \, q^{-\alpha_L(t_L - m_L) - \alpha_R(t_R - m_R) - P_{m_D, m_U}(s, t, t_L, t_R)}$$

$$\times (1 - e^{\frac{2\pi i d n_{21}}{c}} q^{\alpha_L})^{-s} (1 - e^{\frac{2\pi i d n_{43}}{c}} q^{\alpha_R})^{-s} . \tag{5.16}$$

Here, the polynomial $P_{m_D, m_U}(s, t, t_L, t_R)$ is given by

$$P_{m_D, m_U} = -\frac{\det \mathcal{G}_{p_1 p_2 p_3 \ell}}{\det \mathcal{G}_{p_1 p_2 p_3}} \tag{5.17a}$$

$$= -\frac{1}{4stu} \Big[ s^2(t_L - t_R)^2 + 2st(m_D + m_U - s)(t_L + t_R) - 4st\,t_L t_R$$
$$- 4st\,m_D m_U + t^2(m_D - m_U)^2 - st^2(2m_D + 2m_U - s) \Big]. \tag{5.17b}$$

Here, $\mathcal{G}_{p_1 p_2 p_3}$ and $\mathcal{G}_{p_1 p_2 p_3 \ell}$ denote the Gram determinants of the respective momenta (where $\ell$ is the field-theoretic loop momentum). As explained in [14], this polynomial is expected from field-theory considerations where it plays the role of the kernel in the Baikov representation [36] of the imaginary part of the amplitude.

The image of the integration region for the $\xi_i$'s is

$$\mathcal{R} = \left\{ (\alpha_L, \alpha_R, t_L, t_R) \,\middle|\, \begin{matrix} -1\,(0) \leqslant \alpha_L, \alpha_R \leqslant 1 \\ t_L - m_D \leqslant -u\alpha_R, \, t\alpha_R, \, s\alpha_L - u\alpha_R, \, s\alpha_L + t\alpha_R \leqslant s + t_L - m_D \end{matrix} \right\}, \tag{5.18}$$

with $t_R$ unrestricted. The lower limit on $\alpha_L$ is 0 instead of $-1$ when $n_{21} = 0$ and the lower limit of $\alpha_R$ is 0 when $n_{43} = 0$ and we indicated these special cases with values in the parenthesis. We claim that we can change the integration region to the following:

$$\tilde{\mathcal{R}} = \{ (\alpha_L, \alpha_R, t_L, t_R) \,|\, \alpha_L, \alpha_R \geqslant \infty\,(0), \, P_{m_D, m_U}(s, t, t_L, t_R) \geqslant 0 \}, \tag{5.19}$$

without changing the value of the integral. To see that this is possible, we need to check that the leading exponent

$$\text{Trop} = -\alpha_L(t_L - m_L) - \alpha_R(t_R - m_R) - P_{m_D, m_U}(s, t, t_L, t_R) - \min(\alpha_L, 0)s - \min(\alpha_R, 0)s \tag{5.20}$$

is everywhere positive on the difference $(\mathcal{R}^c \cap \tilde{\mathcal{R}}) \cup (\mathcal{R} \cap \tilde{\mathcal{R}}^c)$. For this statement to be true, one needs to use the fact that the range of $m_L$ and $m_R$ is bounded as in eq. (5.11). The statement is also only true if $s \geqslant (\sqrt{m_D} + \sqrt{m_U})^2$, which is the range where the term contributes. In practice, we checked the correctness of this statement numerically. It would be of course nice to show this analytically, but the involved algebra unfortunately gets very quickly very complicated.

After changing the integration region in (5.16), we can integrate out $\alpha_L$ and $\alpha_R$ analytically. In both cases, we need to compute an integral of the form

$$\int_{-\infty\,(0)}^{\infty} d\alpha \, q^{-\alpha t}(1 - e^{2\pi i \varphi} q^\alpha)^{-s} = \frac{i}{2\pi\tau} \int_0^{\infty\,(1)} dx \, x^{-t-1} \left(1 - e^{2\pi i \varphi} x\right)^{-s}, \tag{5.21}$$

for some phase $e^{2\pi i \varphi}$. The integration boundaries in parentheses apply when $\varphi \in \mathbb{Z}$. Let us first assume that $\tau \in i\mathbb{R}$ so that $q$ is real. Since varying $\tau$ does not change the branch cut structure of the integrand, the result depends analytically on $\tau$ and we can obtain the general result by analytic continuation. On the right-hand side, we changed variables to $x = q^\alpha$. The boundary $\alpha \to \infty$ gets mapped to $x = 0$, while the lower boundary gets mapped to $\infty$ and 1 in the two cases.

In the case $\varphi \in \mathbb{Z}$, we end up with the integral

$$(5.21) = \frac{i}{2\pi\tau} \int_0^1 dx \, x^{-t-1} (1-x)^{-s} = \frac{i}{2\pi\tau} \frac{\Gamma(1-s)\Gamma(-t)}{\Gamma(1-s-t)} . \tag{5.22}$$

Now assume that $\varphi \notin \mathbb{Z}$. We rotate the contour by defining

$$y = -e^{2\pi i \varphi} x = e^{2\pi i (\!(\varphi)\!)} x . \tag{5.23}$$

When rotating the contour, the arc at infinity gives a vanishing contribution to the integral in $s$-channel kinematics and can be discarded. Thus we have

$$(5.21) = \frac{i e^{2\pi i t (\!(\varphi)\!)}}{2\pi\tau} \int_0^\infty dy \, y^{-t-1}(1+y)^{-s} = \frac{i e^{2\pi i t (\!(\varphi)\!)}}{2\pi\tau} \frac{\Gamma(-t)\Gamma(s+t)}{\Gamma(s)} . \tag{5.24}$$

The choice of branch of $e^{2\pi i t (\!(\varphi)\!)}$ is correct, as can be easily seen from the following two facts: (i) the branch can only jump when $\varphi \in \mathbb{Z}$, since then the branch point crosses the integration contour and (ii) this is the correct branch for $\varphi = \frac{1}{2}$, where we do not have to rotate the contour at all. As mentioned above, these results still hold for $\tau \notin i\mathbb{R}$ since by varying $\tau$ continuously, no branch points cross the integration contour.

Coming back to (5.16), we can now fully evaluate the contribution of a single term in the $q$-expansion:

$$(5.16) = i \int_{\longrightarrow} \frac{d\tau}{c^5 \tau^2} \sqrt{\frac{-i\tau}{2stu}} \int_{P_{m_D,m_U}>0} dt_L \, dt_R \left(\frac{i}{2\pi\tau}\right)^2 q^{-P_{m_D,m_U}(s,t,t_L,t_R)}$$

$$\times \left( \begin{cases} e^{2\pi i (t_L - m_L)(\!(\frac{dn_{21}}{c})\!)}, & \text{if } n_{21} > 0 \\ \frac{\sin(\pi(s+t_L))}{\sin(\pi s)}, & \text{if } n_{21} = 0 \end{cases} \right) \left( \begin{cases} e^{2\pi i (t_R - m_R)(\!(\frac{dn_{43}}{c})\!)}, & \text{if } n_{43} > 0 \\ \frac{\sin(\pi(s+t_R))}{\sin(\pi s)}, & \text{if } n_{43} = 0 \end{cases} \right)$$

$$\times \frac{\Gamma(-t_L + m_L)\Gamma(-t_R + m_R)\Gamma(s + t_L - m_L)\Gamma(s + t_R - m_R)}{\Gamma(s)^2} . \tag{5.25}$$

In order to integrate out the $\tau$ variable, we can use the Hankel contour representation of the Gamma function,

$$\int_{\longrightarrow} \frac{d\tau}{(-i\tau)^{\frac{7}{2}}} e^{-2\pi i \tau a} = -i (2\pi a)^{\frac{5}{2}} \int_\uparrow dx \, (-x)^{-\frac{7}{2}} e^{-x} = \frac{2\pi (2\pi a)^{\frac{5}{2}}}{\Gamma(\frac{7}{2})} , \tag{5.26}$$

that we already explained in Section 4.5. The result is

$$(5.16) = -\frac{16\pi i}{15 c^5 \sqrt{stu}} \int_{P_{m_D,m_U}>0} dt_L \, dt_R \, P_{m_D,m_U}(s,t,t_L,t_R)^{\frac{5}{2}}$$

$$\times \left( \begin{cases} e^{2\pi i (t_L - m_L)(\!(\frac{dn_{21}}{c})\!)}, & \text{if } n_{21} > 0 \\ \frac{\sin(\pi(s+t_L))}{\sin(\pi s)}, & \text{if } n_{21} = 0 \end{cases} \right) \left( \begin{cases} e^{2\pi i (t_R - m_R)(\!(\frac{dn_{43}}{c})\!)}, & \text{if } n_{43} > 0 \\ \frac{\sin(\pi(s+t_R))}{\sin(\pi s)}, & \text{if } n_{43} = 0 \end{cases} \right)$$

$$\times \frac{\Gamma(-t_L + m_L)\Gamma(-t_R + m_R)\Gamma(s + t_L - m_L)\Gamma(s + t_R - m_R)}{\Gamma(s)^2} . \tag{5.27}$$

## 5.6 Assembling the result

We can now combine eq. (5.8) for the contribution of $A_{a/c}^{n_1,n_2,n_3}$ to the amplitude with our evaluation of a contribution of a single term in the $q$-expansion (5.27) to obtain

$$A_{a/c}^{n_1,n_2,n_3} = -\frac{16\pi i \, e^{2\pi i \sum_{i>j} s_{ij} \sum_{m=1}^{n_{ij}} (\!(\frac{md}{c})\!)}}{15 c^5 \sqrt{stu}} \sum_{\substack{m_L,m_D,m_R,m_U \geqslant 0 \\ (\sqrt{m_D}+\sqrt{m_U})^2 \leqslant s}} \left[ q^{m_L \xi_{21} + m_D \xi_{32} + m_R \xi_{43} + m_U (1-\xi_{41})} \right]$$

$$\times \prod_{i>j} \prod_{\ell=1}^{\infty} \left(1 - e^{-\frac{2\pi i d(\ell+n_{ij})}{c}} q^{\ell-\xi_{ij}}\right)^{-s_{ij}} \prod_{\ell=1+\delta_{ij,21}+\delta_{ij,41}}^{\infty} \left(1 - e^{-\frac{2\pi i d(\ell-n_{ij}-1)}{c}} q^{\ell+\xi_{ij}-1}\right)^{-s_{ij}}$$

$$\times \int_{P_{m_{\mathrm{D}},m_{\mathrm{U}}}>0} dt_{\mathrm{L}} \, dt_{\mathrm{R}} \, P_{m_{\mathrm{D}},m_{\mathrm{U}}}(s,t,t_{\mathrm{L}},t_{\mathrm{R}})^{\frac{5}{2}}$$

$$\times \left(\begin{cases} e^{2\pi i(t_{\mathrm{L}}-m_{\mathrm{L}})((\frac{dn_{21}}{c}))}, & \text{if } n_{21} > 0 \\ \frac{\sin(\pi(s+t_{\mathrm{L}}))}{\sin(\pi s)}, & \text{if } n_{21} = 0 \end{cases}\right) \left(\begin{cases} e^{2\pi i(t_{\mathrm{R}}-m_{\mathrm{R}})((\frac{dn_{43}}{c}))}, & \text{if } n_{43} > 0 \\ \frac{\sin(\pi(s+t_{\mathrm{R}}))}{\sin(\pi s)}, & \text{if } n_{43} = 0 \end{cases}\right)$$

$$\times \frac{\Gamma(-t_{\mathrm{L}}+m_{\mathrm{L}})\Gamma(-t_{\mathrm{R}}+m_{\mathrm{R}})\Gamma(s+t_{\mathrm{L}}-m_{\mathrm{L}})\Gamma(s+t_{\mathrm{R}}-m_{\mathrm{R}})}{\Gamma(s)^2} . \tag{5.28}$$

We can simplify this formula further. Let us look at the coefficients of the infinite product in more detail. We notice that the phase of a given term in the infinite product is entirely determined by the exponent and we can write

$$\left[q^{m_{\mathrm{L}}\xi_{21}+m_{\mathrm{D}}\xi_{32}+m_{\mathrm{R}}\xi_{43}+m_{\mathrm{U}}(1-\xi_{41})}\right] \prod_{i>j} \prod_{\ell=1}^{\infty} \left(1 - e^{-\frac{2\pi i d(\ell+n_{ij})}{c}} q^{\ell-\xi_{ij}}\right)^{-s_{ij}}$$

$$\times \prod_{\ell=1+\delta_{ij,21}+\delta_{ij,43}}^{\infty} \left(1 - e^{-\frac{2\pi i d(\ell-n_{ij}-1)}{c}} q^{\ell+\xi_{ij}-1}\right)^{-s_{ij}}$$

$$= e^{\frac{2\pi i d}{c}(m_{\mathrm{L}}n_{21}+m_{\mathrm{D}}n_{32}+m_{\mathrm{R}}n_{43}-m_{\mathrm{U}}(n_{41}+1))} \left[q^{m_{\mathrm{L}}\xi_{21}+m_{\mathrm{D}}\xi_{32}+m_{\mathrm{R}}\xi_{43}+m_{\mathrm{U}}(1-\xi_{41})}\right]$$

$$\times \prod_{i>j} \prod_{\ell=1}^{\infty} \left(1 - q^{\ell-\xi_{ij}}\right)^{-s_{ij}} \prod_{\ell=1+\delta_{ij,21}+\delta_{ij,43}}^{\infty} \left(1 - q^{\ell+\xi_{ij}-1}\right)^{-s_{ij}} . \tag{5.29}$$

Part of this phase combines with the phase that we obtained from evaluating the integrals over $\alpha_{\mathrm{L}}$ and $\alpha_{\mathrm{R}}$. We note that $e^{\frac{2\pi i m_{\mathrm{L}} n_{21} d}{c}} e^{-2\pi i m_{\mathrm{L}}((\frac{dn_{21}}{c}))} = (-1)^{m_{\mathrm{L}}}$. Setting

$$q_{\mathrm{L}} = q^{\xi_{21}}, \qquad q_{\mathrm{D}} = q^{\xi_{32}}, \qquad q_{\mathrm{R}} = q^{\xi_{43}}, \qquad q_{\mathrm{U}} = q^{1-\xi_{41}}, \tag{5.30}$$

recovers the definition (3.27) for the polynomials coefficients $Q_{m_{\mathrm{L}},m_{\mathrm{D}},m_{\mathrm{R}},m_{\mathrm{U}}}(s,t)$. We thus have at this stage

$$A_{a/c}^{n_1,n_2,n_3} = -\frac{16\pi \, e^{2\pi i \sum_{i>j} s_{ij} \sum_{m=1}^{n_{ij}} ((\frac{md}{c}))}}{15 c^5 \sqrt{stu}} \sum_{\substack{m_{\mathrm{L}},m_{\mathrm{D}},m_{\mathrm{R}},m_{\mathrm{U}} \geqslant 0 \\ (\sqrt{m_{\mathrm{D}}}+\sqrt{m_{\mathrm{U}}})^2 \leqslant s}} (-1)^{m_{\mathrm{L}}+m_{\mathrm{R}}} Q_{m_{\mathrm{L}},m_{\mathrm{D}},m_{\mathrm{R}},m_{\mathrm{U}}}(s,t)$$

$$\times e^{\frac{2\pi i d}{c}(m_{\mathrm{D}}n_{32}-m_{\mathrm{U}}(n_{41}+1))} \int_{P_{m_{\mathrm{D}},m_{\mathrm{U}}}>0} dt_{\mathrm{L}} \, dt_{\mathrm{R}} \, P_{m_{\mathrm{D}},m_{\mathrm{U}}}(s,t,t_{\mathrm{L}},t_{\mathrm{R}})^{\frac{5}{2}}$$

$$\times \left(\begin{cases} e^{2\pi i t_{\mathrm{L}}((\frac{dn_{21}}{c}))}, & \text{if } n_{21} > 0 \\ \frac{\sin(\pi(s+t_{\mathrm{L}}))}{\sin(\pi s)}, & \text{if } n_{21} = 0 \end{cases}\right) \left(\begin{cases} e^{2\pi i t_{\mathrm{R}}((\frac{dn_{43}}{c}))}, & \text{if } n_{43} > 0 \\ \frac{\sin(\pi(s+t_{\mathrm{R}}))}{\sin(\pi s)}, & \text{if } n_{43} = 0 \end{cases}\right)$$

$$\times \frac{\Gamma(-t_{\mathrm{L}}+m_{\mathrm{L}})\Gamma(-t_{\mathrm{R}}+m_{\mathrm{R}})\Gamma(s+t_{\mathrm{L}}-m_{\mathrm{L}})\Gamma(s+t_{\mathrm{R}}-m_{\mathrm{R}})}{\Gamma(s)^2} . \tag{5.31}$$

We can now carry out the sum over $m_{\mathrm{L}}$ and $m_{\mathrm{R}}$. Recall the definition for the polynomials $Q_{m_{\mathrm{D}},m_{\mathrm{U}}}(s,t)$ given in (3.28), which we reproduce here:

$$Q_{m_{\mathrm{D}},m_{\mathrm{U}}}(s,t,t_{\mathrm{L}},t_{\mathrm{R}}) \equiv \sum_{m_{\mathrm{L}},m_{\mathrm{R}}=0}^{m_{\mathrm{D}}+m_{\mathrm{U}}} Q_{m_{\mathrm{L}},m_{\mathrm{D}},m_{\mathrm{R}},m_{\mathrm{U}}}(s,t)(-t_{\mathrm{L}})_{m_{\mathrm{L}}}(-s-t_{\mathrm{L}}+m_{\mathrm{L}}+1)_{m_{\mathrm{D}}+m_{\mathrm{U}}-m_{\mathrm{L}}}$$

$$\times (-t_{\mathrm{R}})_{m_{\mathrm{R}}}(-s-t_{\mathrm{R}}+m_{\mathrm{R}}+1)_{m_{\mathrm{D}}+m_{\mathrm{U}}-m_{\mathrm{R}}}. \tag{5.32}$$

We used the fact that the range of $m_L$ and $m_R$ is given by (5.11).[8] We can hence write

$$A_{a/c}^{n_1,n_2,n_3} = -\frac{16\pi i \, e^{2\pi i \sum_{i>j} s_{ij} \sum_{m=1}^{n_{ij}} (\!(\frac{md}{c})\!)}}{15c^5\sqrt{stu}} \sum_{\substack{m_D,m_U \geqslant 0 \\ (\sqrt{m_D}+\sqrt{m_U})^2 \leqslant s}} e^{\frac{2\pi i d}{c}(m_D n_{32}-m_U(n_{41}+1))}$$

$$\times \int_{P_{m_D,m_U}>0} dt_L\, dt_R\, P_{m_D,m_U}(s,t,t_L,t_R)^{\frac{5}{2}}\, Q_{m_D,m_U}(s,t,t_L,t_R)$$

$$\times \left(\begin{cases} e^{2\pi i t_L (\!(\frac{dn_{21}}{c})\!)}, & \text{if } n_{21} > 0 \\ \frac{\sin(\pi(s+t_L))}{\sin(\pi s)}, & \text{if } n_{21} = 0 \end{cases}\right)\left(\begin{cases} e^{2\pi i t_R (\!(\frac{dn_{43}}{c})\!)}, & \text{if } n_{43} > 0 \\ \frac{\sin(\pi(s+t_R))}{\sin(\pi s)}, & \text{if } n_{43} = 0 \end{cases}\right)$$

$$\times \frac{\Gamma(-t_L)\Gamma(-t_R)\Gamma(s+t_L-m_D-m_U)\Gamma(s+t_R-m_D-m_U)}{\Gamma(s)^2}. \tag{5.33}$$

### 5.7 Renaming $(n_1, n_2, n_3)$

The final step is an aesthetic one. We change our labelling of the contributions $A_{a/c}^{n_1,n_2,n_3}$ to make the final formula more symmetric. Let us introduce

$$n_L = n_{21}, \qquad n_D = n_{32}, \qquad n_R = n_{43}, \qquad n_U = n_1 - 1, \tag{5.34}$$

so that $n_L, n_D, n_R, n_U \geqslant 0$ and their sum is constrained to $c-1$. In terms of these variables, the prefactor phase can be written as

$$\sum_{i>j} s_{ij} \sum_{m=1}^{n_{ij}} \left(\!\left(\frac{md}{c}\right)\!\right) = s\left[\sum_{m=1}^{n_L} + \sum_{m=1}^{n_R} - \sum_{m=1}^{n_D+n_L} - \sum_{m=1}^{n_D+n_R}\right]\left(\!\left(\frac{md}{c}\right)\!\right)$$

$$+ t\left[\sum_{m=1}^{n_D} + \sum_{m=1}^{c-1-n_U} - \sum_{m=1}^{n_D+n_L} - \sum_{m=1}^{n_D+n_R}\right]\left(\!\left(\frac{md}{c}\right)\!\right) \tag{5.35}$$

$$= -s\left[\sum_{m=n_L+1}^{n_L+n_D} + \sum_{m=n_R+1}^{n_R+n_D}\right]\left(\!\left(\frac{md}{c}\right)\!\right)$$

$$- t\left[\sum_{m=n_D+1}^{n_L+n_D} - \sum_{m=n_D+n_R+1}^{c-1-n_U}\right]\left(\!\left(\frac{md}{c}\right)\!\right). \tag{5.36}$$

By using periodicity in $m$ mod $c$, we can rewrite the last term as follows:

$$\sum_{m=n_D+n_R+1}^{c-1-n_U}\left(\!\left(\frac{md}{c}\right)\!\right) = \sum_{m=n_D+n_R+1-c}^{-1-n_U}\left(\!\left(\frac{md}{c}\right)\!\right) = -\sum_{m=n_U+1}^{n_U+n_L}\left(\!\left(\frac{md}{c}\right)\!\right), \tag{5.37}$$

where we shifted the summation domain by $c$ steps and then renamed $m \to -m$. Thus,

$$\sum_{i>j} s_{ij} \sum_{m=1}^{n_{ij}} \left(\!\left(\frac{md}{c}\right)\!\right) = -s \sum_{a=L,R} \sum_{m=n_a+1}^{n_a+n_D}\left(\!\left(\frac{md}{c}\right)\!\right) - t \sum_{a=D,U} \sum_{m=n_a+1}^{n_a+n_L}\left(\!\left(\frac{md}{c}\right)\!\right). \tag{5.38}$$

This looks still slightly asymmetric. However, we claim that

$$\sum_{a=L,R} \sum_{m=n_a+1}^{n_a+n_D}\left(\!\left(\frac{md}{c}\right)\!\right) = \sum_{a=L,R} \sum_{m=n_a+1}^{n_a+n_U}\left(\!\left(\frac{md}{c}\right)\!\right), \tag{5.39}$$

---

[8]This is almost what we called $Q_{m_D,m_U}$ in our earlier paper [14]. For $m_D = m_U$, the two are literally the same, while it differs by a factor of 2 from what we called $Q_{m_D,m_U}$ before, since we include both $m_D < m_U$ and $m_D > m_U$ in the present sum.

and similarly the second term is invariant under $n_\mathrm{L} \to n_\mathrm{R}$, which makes it obvious that the expression has in fact the required symmetries. To prove this, we only need to use that $((x))$ is an asymmetric function. We have

$$
\sum_{a=\mathrm{L,R}} \sum_{m=n_a+1}^{n_a+n_\mathrm{D}} \left(\!\left(\frac{md}{c}\right)\!\right) - \sum_{a=\mathrm{L,R}} \sum_{m=n_a+1}^{n_a+n_\mathrm{U}} \left(\!\left(\frac{md}{c}\right)\!\right)
$$

$$
= \left[ \sum_{m=n_\mathrm{L}+1}^{n_\mathrm{L}+n_\mathrm{D}} - \sum_{m=-n_\mathrm{R}-n_\mathrm{D}}^{-n_\mathrm{R}-1} - \sum_{m=n_\mathrm{L}+1}^{n_\mathrm{L}+n_\mathrm{U}} + \sum_{m=-n_\mathrm{R}-n_\mathrm{U}}^{-n_\mathrm{R}-1} \right] \left(\!\left(\frac{md}{c}\right)\!\right) \tag{5.40}
$$

$$
= \left[ \sum_{m=n_\mathrm{L}+1}^{n_\mathrm{L}+n_\mathrm{D}} - \sum_{m=n_\mathrm{L}+n_\mathrm{U}+1}^{n_\mathrm{L}+n_\mathrm{D}+n_\mathrm{U}} - \sum_{m=n_\mathrm{L}+1}^{n_\mathrm{L}+n_\mathrm{U}} + \sum_{m=n_\mathrm{L}+n_\mathrm{D}+1}^{n_\mathrm{L}+n_\mathrm{D}+n_\mathrm{U}} \right] \left(\!\left(\frac{md}{c}\right)\!\right) = 0 . \tag{5.41}
$$

Here we sent $m \to -m$ in the second and fourth term and then shifted summation variables by $c$. The first and last term, as well as the second and third term then join up and the terms cancel.

Thus we can write the phases fully symmetrically as follows,

$$
\sum_{i>j} s_{ij} \sum_{m=1}^{n_{ij}} \left(\!\left(\frac{md}{c}\right)\!\right) = -\frac{1}{2} \sum_{\substack{a=\mathrm{L,R} \\ b=\mathrm{D,U}}} \left[ s \sum_{m=n_a+1}^{n_a+n_b} + t \sum_{m=n_b+1}^{n_a+n_b} \right] \left(\!\left(\frac{md}{c}\right)\!\right) . \tag{5.42}
$$

Inserting this into (5.33) then finally yields (3.25). This is our final formula for the planar open-string amplitude in the $s$-channel.

## 5.8 Results in the $u$-channel

We now consider the $u$-channel contribution. As we shall see, we almost get it for free from the $s$-channel contribution. There are $\frac{1}{6}(m-1)m(m+1)$ regions in $(z_1, z_2, z_3)$-parameter space for which $\mathrm{Trop} < 0$. We will denote them by $\Gamma_{n_1, n_2, n_3}$ where $1 \leqslant n_1 \leqslant n_2 < n_3 \leqslant c$. They are separated by the inequalities

$$
\frac{n_{ij}}{c} \leqslant z_{ij} \leqslant \frac{n_{ij}+1}{c} , \qquad ij \in \{21, 41, 43\} , \tag{5.43a}
$$

$$
\frac{n_{32}-1}{c} \leqslant z_{32} \leqslant \frac{n_{32}}{c} , \tag{5.43b}
$$

$$
\frac{n_{ij}-1}{c} \leqslant z_{ij} \leqslant \frac{n_{ij}+1}{c} , \qquad ij \in \{31, 42\} . \tag{5.43c}
$$

Here we defined again $n_4 \equiv c$. Let us set $z_i = \frac{n_i + \xi_i}{c}$ as before.

It follows that the contribution $A_{a/c}^{n_1, n_2, n_3}$ to the $u$-channel amplitude equals

$$
A_{a/c}^{n_1, n_2, n_3} = i \int_{\longrightarrow} \frac{\mathrm{d}\tau}{c^5 \tau^2} \int \mathrm{d}\xi_1 \mathrm{d}\xi_2 \mathrm{d}\xi_3 \prod_{i>j} q^{-\frac{1}{2}s_{ij}\xi_{ij}(\xi_{ij}-1)} e^{2\pi i s_{ij} \sum_{m=1}^{n_{ij}-\delta_{ij,32}} \left(\!\left(\frac{md}{c}\right)\!\right)}
$$

$$
\times \prod_{\substack{i>j \\ ij \neq 32}} \prod_{\ell=1}^{\infty} \left( 1 - e^{-\frac{2\pi i d(\ell+n_{ij})}{c}} q^{\ell-\xi_{ij}} \right)^{-s_{ij}} \left( 1 - e^{-\frac{2\pi i d(\ell-n_{ij}-1)}{c}} q^{\ell+\xi_{ij}-1} \right)^{-s_{ij}}
$$

$$
\times \prod_{\ell=1}^{\infty} \left( 1 - e^{-\frac{2\pi i d(\ell+n_{32}-1)}{c}} q^{\ell-\xi_{32}-1} \right)^{-s_{32}} \left( 1 - e^{-\frac{2\pi i d(\ell-n_{32})}{c}} q^{\ell+\xi_{32}} \right)^{-s_{32}} . \tag{5.44}
$$

This formula is straightforward to derive. To deal with the factor of the form $\vartheta_1(c\tau z_{32}, \tau - \frac{d}{c}) = \vartheta_1(\tau(n_{32}-1+\xi_{32}+1), \tau - \frac{d}{c})$, we use eq. (4.16) with $n = n_{32}-1$ and then insert $\xi = \xi_{32}+1$, which is positive. This is almost identical to (5.8), except for a different integration region over the $\xi_i$'s and a slightly different phase. We also treated the factor with $ij = 32$ separately in order to ensure that we land on the correct branch.

We can relate this to the $s$-channel contributions as follows. Consider swapping $\xi_2$ with $\xi_3$, $n_2$ with $n_3$ and $s$ with $u$ (i.e. we swap all the labels 2 with the labels 3). This turns $\xi_{32} \to -\xi_{32}$, $n_{32} \to -n_{32}$, while the terms in the second line get permuted. We can then combine them with the terms in the third line to obtain

$$A_{a/c}^{n_1,n_3,n_2}\Big|_{2\leftrightarrow 3} = i \int_{\longrightarrow} \frac{\mathrm{d}\tau}{c^5\tau^2} \int \mathrm{d}\xi_1\,\mathrm{d}\xi_2\,\mathrm{d}\xi_3 \prod_{i>j} q^{-\frac{1}{2}s_{ij}\xi_{ij}(\xi_{ij}-1)}$$

$$\times e^{2\pi i \sum_{i>j, ij\neq 32} s_{ij} \sum_{m=1}^{n_{ij}} ((\frac{md}{c})) + 2\pi i t \sum_{m=1}^{-n_{32}-1}((\frac{md}{c}))}$$

$$\times \prod_{i>j} \prod_{\ell=1}^{\infty} \left(1 - e^{-\frac{2\pi id(\ell+n_{ij})}{c}} q^{\ell-\xi_{ij}}\right)^{-s_{ij}} \left(1 - e^{-\frac{2\pi id(\ell-n_{ij}-1)}{c}} q^{\ell+\xi_{ij}-1}\right)^{-s_{ij}}. \quad (5.45)$$

In terms of the new $\xi_i$'s, the integration region is bounded by

$$-1 \leqslant \xi_{21}, \xi_{43} \leqslant 1, \qquad 0 \leqslant \xi_{31}, \xi_{41}, \xi_{32}, \xi_{42} \leqslant 1, \quad (5.46)$$

which coincides with the integration region in the $s$-channel. Up to the slightly different phase, this directly coincides with the $s$-channel $A_{a,c}^{n_1,n_2,n_3}$. Note that $n_{32} < 0$, so a modification of this particular factor in the phase is expected. Note also that $n_{21} > 0$ and $n_{43} > 0$ on the right-hand side and thus only one of the cases in the $s$-channel formula appears in this case.

Let us also express this result in terms of $n_L = n_{31}$, $n_D = -n_{32}$, $n_R = n_{42}$ and $n_U = c-1-n_{41}$, so that $n_L + n_D + n_R + n_U = c - 1$. We have of course also the following inequalities:

$$n_L > 0, \quad n_D < 0, \quad n_R > 0, \quad n_U \geqslant 0, \quad n_L + n_D \geqslant 0, \quad n_R + n_D \geqslant 0. \quad (5.47)$$

Taking (3.25) and exchanging all labels finally gives

$$A_{a/c}^{n_L,n_D,n_R,n_U} = -\frac{16\pi i\, e^{-2\pi is \sum_{a=L,R} \sum_{m=n_a+n_D+1}^{n_a} ((\frac{md}{c})) + 2\pi it\left[\sum_{m=n_L+1}^{n_L+n_R+n_D} - \sum_{m=-n_D}^{n_R}\right]((\frac{md}{c}))}}{15c^5\sqrt{stu}}$$

$$\times \sum_{\substack{m_D,m_U \geqslant 0 \\ (\sqrt{m_D}+\sqrt{m_U})^2 \leqslant u}} e^{\frac{2\pi id}{c}(m_D n_D + m_U n_U)} \int_{P_{m_D,m_U}(s,u,t_L,t_R)\geqslant 0} \mathrm{d}t_L\,\mathrm{d}t_R\, P_{m_D,m_U}(s,u,t_L,t_R)^{\frac{5}{2}}$$

$$\times Q_{m_D,m_U}(s,u,t_L,t_R) \frac{\Gamma(-t_L)\Gamma(u+t_L-m_D-m_U)}{\Gamma(u)} e^{2\pi it_L((\frac{dn_L}{c}))}(L\leftrightarrow R). \quad (5.48)$$

# 6 Four-point non-planar amplitude $A^{\text{n-p}}(s,t)$

Let us derive the analogous formula for the non-planar annulus amplitude $A^{\text{n-p}}(s,t)$. Recall from Section 2.1 that the amplitude of the non-planar annulus was given by

$$A^{\text{n-p}} = \frac{-i}{32(1-e^{-\pi is})} \int_\Gamma \mathrm{d}\tau\,\mathrm{d}z_1\,\mathrm{d}z_2\,\mathrm{d}z_3 \prod_{j=1}^{2}\prod_{i=3}^{4} \vartheta_4(z_{ij},\tau)^{-s_{ij}}\big(\vartheta_1(z_{21},\tau)\vartheta_1(z_{43},\tau)\big)^{-s}. \quad (6.1)$$

The integration contour $\Gamma$ is the $\tau$-plane is the one described in Figure 12 with the endpoints at $\tau = 0$ and $\tau = 2$. The prefactor $(1-e^{-\pi is})^{-1}$ came from the choice of the contour, but we will

suppress it from now on for readability by introducing $\tilde{A}^{\text{n-p}} = (1 - e^{-\pi i s})A^{\text{n-p}}$. The integration region in the $z_i$'s can be described by the inequalities

$$0 \leqslant z_{21}, z_{43} \leqslant 1, \qquad 0 \leqslant z_{42} \leqslant 1. \tag{6.2}$$

Note also that the integrand is periodic in $(z_1, z_2) \to (z_1 + 1, z_2 + 1)$, which corresponds to taking the punctures around the inner boundary. We again want to compute the contribution from the integral around the circles $C_{a/c}$, where now $0 < \frac{a}{c} \leqslant 2$.

The treatment of the correct branches is more complicated in this case compared to the planar case and the generalization of the computation is not entirely straightforward.

## 6.1 Shifting $z_i$

Our strategy will be to recycle the computation of the planar annulus as much as possible. It is thus advantageous to relate $\vartheta_4$ to $\vartheta_1$. Let us set

$$z_1 = y_1, \qquad z_2 = y_2, \qquad z_3 = y_3 + \frac{\tau}{2}, \qquad z_4 = y_4 + \frac{\tau}{2}. \tag{6.3}$$

We do not fix $z_4 = 1$ for the moment. Then some of the arguments of the theta functions in (6.1) become $\vartheta_4(y_{31} + \frac{\tau}{2}, \tau)$ etc. Let us compute

$$\vartheta_4(z, \tau) = \vartheta_4(y + \tfrac{\tau}{2}, \tau) = \sum_{n \in \mathbb{Z}} (-1)^n e^{2\pi i n(y + \frac{\tau}{2}) + \pi i n^2 \tau} \tag{6.4a}$$

$$= e^{-\frac{\pi i \tau}{4} - \pi i y} \sum_{n \in \mathbb{Z}} (-1)^n e^{2\pi i (n + \frac{1}{2})y + \pi i (n + \frac{1}{2})^2 \tau} \tag{6.4b}$$

$$= i e^{-\frac{\pi i \tau}{4} - \pi i y} \vartheta_1(y, \tau). \tag{6.4c}$$

We can thus write

$$\tilde{A}^{\text{n-p}} = \frac{-i}{32} \int_\Gamma d\tau \, dy_1 \, dy_2 \, dy_3 \, e^{\pi i s - \frac{\pi i s \tau}{2} - \pi i s(y_3 + y_4 - y_1 - y_2)} \prod_{i > j} \vartheta_1(y_{ij}, \tau)^{-s_{ij}}, \tag{6.5}$$

where the integration contour in $y_i$ follows from the one in $z_i$ by shifting. The natural way to choose the branch in this expression is to take $y_i$ close to zero with $y_{ij} > 0$, then integrand simplifies to

$$e^{\pi i s - \frac{\pi i s \tau}{2}} \prod_{i > j} y_{ij}^{-s_{ij}}. \tag{6.6}$$

Since all $y_{ij}$'s are positive, the branch is the canonical one. We have to make sure that this choice follows from the original choice of branch in (6.1) by analytic continuation, which was specified by taking all $z_i$'s close to zero by similar reasoning.

It suffices to do this for one of the $\vartheta_4$'s. We can also assume that $\tau$ is purely imaginary and very large since the overall phase depends continuously on $\tau$. Consider

$$\log \vartheta_1(z + \tfrac{\sigma \tau}{2}, \tau), \tag{6.7}$$

for $\sigma \in [0, 1]$ and take $z \in \mathbb{R}$ small, but bigger than 0. For $\sigma = 0$, the choice of branch is clear, since $\vartheta_1(z, \tau) \to z \vartheta_1'(0, \tau) \sim 2\pi z e^{\frac{\pi i \tau}{4}}$ as $\text{Im}\,\tau \to \infty$. It corresponds to the choice of branch in (6.6). We want to follow the branch smoothly from $\sigma = 0$ to $\sigma = 1$. For large $\text{Im}\,\tau$ we have approximately

$$\log \vartheta_1(z + \tfrac{\sigma \tau}{2}, \tau) \sim \frac{\pi i \tau}{4} + \log\left(-i\, e^{\pi i(z + \frac{\sigma \tau}{2})} + i\, e^{-\pi i(z + \frac{\sigma \tau}{2})}\right). \tag{6.8}$$

We took out $\frac{\pi i \tau}{4}$, since this term is real. For $\sigma = 0$, both terms are equally relevant and as discussed above, we choose the principal branch. For $\sigma = 1$, the second term dominates and is essentially purely imaginary. However, we never cross the negative axis (since the first term is always less than the second term in magnitude) and thus the principal branch of the logarithm gives the correct answer everywhere. In particular for $\sigma = 1$, we get

$$\log \vartheta_1(z + \tfrac{\tau}{2}, \tau) \sim -\frac{\pi i \tau}{4} + \frac{\pi i}{2} - \pi i z \sim \frac{\pi i \tau}{4} + \frac{\pi i}{2} - \pi i z + \log \vartheta_4(z, \tau). \tag{6.9}$$

This shows that the branch in (6.5) is the one that we get from the original expression (6.1) by following the straight line from $z_{3,4} \sim 0$ to $z_{3,4} \sim \frac{\tau}{2}$.

## 6.2 Modular transformation

The next step is as before to set

$$\tau = \frac{a\tau' + b}{c\tau' + d}. \tag{6.10}$$

Then $\tau' \in i + \mathbb{R}$ gets mapped to the circle $C_{a/c}$. Hence, for the contribution from a single circle we get

$$\tilde{A}_{a/c} = \frac{i}{32} \int_{\longrightarrow} \frac{d\tau'}{(c\tau' + d)^2} \int dz_1 \, dz_2 \, dz_3 \, e^{\pi i s - \frac{\pi i s(a\tau' + b)}{2(c\tau' + d)} - \pi i s(y_3 + y_4 - y_1 - y_2)}$$
$$\times \prod_{i > j} e^{-\pi i c(c\tau' + d)s_{ij}y_{ij}^2} \vartheta_1((c\tau' + d)y_{ij}, \tau')^{-s_{ij}}. \tag{6.11}$$

We are guaranteed that this choice of branch is correct for $y_{ij} \to 0$, where the theta functions drastically simplify. Everywhere else, the branch is determined by analytic continuation.

## 6.3 Further shifts of $z_i$

To proceed further, we now want to re-express the result in terms of the original $z_i$. For this it is convenient to shift the $z_i$'s further. Set

$$z_{1,2} = \zeta_{1,2}, \qquad z_{3,4} = \zeta_{3,4} + \frac{a}{2c}. \tag{6.12}$$

Contrary to the shift that led to $y_i$, this is a real shift. This does not change the integration region and we can still integrate over

$$0 \leqslant \zeta_{21}, \zeta_{43} \leqslant 1, \qquad 0 \leqslant \zeta_{42} \leqslant 1. \tag{6.13}$$

We now have

$$y_{31} = z_{31} - \frac{a\tau' + b}{2(c\tau' + d)} = \zeta_{31} - \frac{a\tau' + b}{2(c\tau' + d)} + \frac{a}{2c} = \zeta_{31} + \frac{1}{2c(c\tau' + d)}. \tag{6.14}$$

This is advantageous because $y_{31}$ and $\zeta_{31}$ are much closer together now. Consider now one of the factors for $ij \in \{31, 32, 41, 42\}$. For the purpose of discussing the branch, we look at

$$f(\zeta) = \log \vartheta_1\left((c\tau' + d)\zeta + \tfrac{1}{2c}, \tau'\right). \tag{6.15}$$

We recall that the branch of the expression is determined from the behaviour near $y = 0$ with $y > 0$ small, i.e. $\zeta = -\frac{1}{2c(c\tau' + d)} + y$ with $y > 0$ small. On the other hand, we can naturally determine a branch by taking $\zeta = 0$, $\tau'$ large and purely imaginary. Since $0 < \frac{1}{2c} \leqslant \frac{1}{2}$, we have

$$\log \vartheta_1\left(\tfrac{1}{2c}, \tau'\right) = \tfrac{\pi i \tau}{4} + \log\left[2\sin\left(\tfrac{\pi}{2c}\right)\right], \tag{6.16}$$

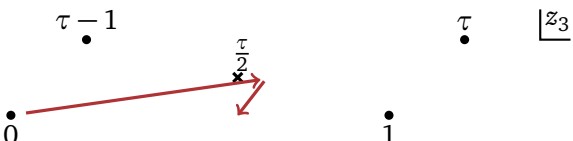

Figure 23: The two analytic continuations in $z_3$.

which also has a natural branch since $\sin\left(\frac{\pi}{2c}\right) > 0$. These two choices of branches are easily seen to be equivalent. Indeed, set $\zeta = -\frac{\sigma}{2c(c\tau'+d)} + y$ and follow the branch from $\sigma = 0$ to $\sigma = 1$. We can again take $\tau$ purely imaginary and very large. Then

$$\log\vartheta_1\left((c\tau'+d)y + \tfrac{1-\sigma}{2c}, \tau'\right) \sim \frac{\pi i\tau}{4} + \log\left[2\sin\pi\left((c\tau'+d)y + \tfrac{1-\sigma}{2c}\right)\right]. \tag{6.17}$$

Since the path $2\sin\pi\left((c\tau'+d)y + \frac{1-\sigma}{2c}\right)$ never crosses the negative real axis, we can just take the principal branch of the logarithm everywhere. We thus conclude that our alternative determination of the branch in terms of $\zeta$ is equally valid and will be more convenient in the following.

To summarize our analytic continuations so far, consider Figure 23. It shows the paths of the two analytic continuations we have performed. First, we analytically continued the integrand from $z_3 \sim 0$ to $z_3 \sim \frac{\tau}{2}$, then we analytically continued back to the real axis. In the picture, $\tau$ is close to $\frac{a}{c}$, so that $\mathrm{Im}\,\tau'$ is large. Clearly, this path of analytic continuation is equivalent to the horizontal path, since we do not surround any branch points of the integrand (represented by crosses in the picture). This confirms that we are still on the correct branch. The same comment applies for $z_4$.

We can plug this change of variables back into (6.5) to obtain the following expression:

$$\tilde{A}_{a/c} = \frac{i}{32}\int_{\longrightarrow}\frac{\mathrm{d}\tau}{(c\tau'+d)^2}\int_0^1 \mathrm{d}\zeta_1\,\mathrm{d}\zeta_2\,\mathrm{d}\zeta_3\,\mathrm{e}^{-\frac{\pi isa}{2c}+\pi is}$$
$$\times \prod_{i>j}\mathrm{e}^{-\pi ic(c\tau'+d)s_{ij}\zeta_{ij}^2}\vartheta_1\left((c\tau'+d)\zeta_{ij} + \tfrac{\delta(i,j)}{2c}, \tau'\right)^{-s_{ij}}. \tag{6.18}$$

Here and in the following, we used the short-hand notation

$$\delta(i,j) = \begin{cases} 1, & ij = 31,\ 32,\ 41\text{ or }42, \\ 0, & ij = 21,\ 43. \end{cases} \tag{6.19}$$

Finally, we shift $\tau' \to \tau - \frac{d}{c}$ and set $q = \mathrm{e}^{2\pi i\tau}$, yielding

$$\tilde{A}_{a/c} = \frac{i}{32}\int_{\longrightarrow}\frac{\mathrm{d}\tau}{c^2\tau^2}\int_0^1 \mathrm{d}\zeta_1\,\mathrm{d}\zeta_2\,\mathrm{d}\zeta_3\,\mathrm{e}^{-\frac{\pi isa}{2c}+\pi is}\prod_{i>j}q^{-\frac{1}{2}s_{ij}c^2\zeta_{ij}^2}\vartheta_1\left(c\tau\zeta_{ij} + \tfrac{\delta(i,j)}{2c}, \tau-\tfrac{d}{c}\right)^{-s_{ij}}. \tag{6.20}$$

We recall that the branch in this expression is chosen by setting all $\zeta_{ij} \sim 0$ with $\zeta_{21} > 0$, $\zeta_{43} > 0$, $\mathrm{Re}\,\tau = \frac{d}{c}$ and $\mathrm{Im}\,\tau$ large and evaluating the powers on the principal branch. Everywhere else, the branch is defined by analytic continuation. Notice that the expression is quite similar to the planar expression at this point, it only differs by an overall phase depending on $a$ and $c$ as well as the additional shifts in the theta function. These will also only lead to phases. Finally the integration region is different, see eq. (6.13).

At this point, it is convenient to shift the integration contour in $\tau$ to large imaginary values of $\tau$ so that we can again take advantage of the tropicalization of the integration.

## 6.4 Subdividing the $\zeta_i$-integration

We now continue the analysis for the $s$-channel. Since the integrand is the same as for the planar annulus up to phases, the contributions where Trop < 0 happen in the same region. In particular (5.5a) still holds when we replace $z_i \to \zeta_i$. The integration region is however larger and thus the integers $(n_1, n_2, n_3)$ can take more values. From the last inequality in (5.5a), we see that $0 \leqslant n_3 \leqslant c$. From the first inequality in (5.5a), we see that $0 \leqslant n_{21} \leqslant c$. To cover the whole integration region, we can let $0 \leqslant n_{32} \leqslant c - 1$. The upper and lower bounds here could be shifted, since the integrand is periodic. It is only important that we cover $c$ different values. We hence have overall

$$0 \leqslant n_{21} \leqslant c, \qquad 0 \leqslant n_{32} \leqslant c - 1, \qquad 0 \leqslant n_{43} \leqslant c. \tag{6.21}$$

This defines the regions $\Gamma_{n_1, n_2, n_3}$. There are $c(c+1)^2$ such regions. For each of the regions, we set

$$\zeta_i = \frac{n_i + \xi_i}{c}. \tag{6.22}$$

The integration region for $n_{21} > 0$, $n_{21} < c$, $n_{43} > 0$ and $n_{43} < c$ are given by

$$-1 \leqslant \xi_{21}, \xi_{43} \leqslant 1, \qquad 0 \leqslant \xi_{31}, \xi_{32}, \xi_{41}, \xi_{42} \leqslant 1. \tag{6.23}$$

The cases $n_{21} = 0$, $n_{21} = c$, $n_{43} = 0$ or $n_{43} = c$ are special since in this case the region for $\xi_{21}$, $\xi_{43}$ are modified. For $n_{21} = 0$, we have $0 \leqslant \xi_{21} \leqslant 1$, while for $n_{21} = c$, we have $-1 \leqslant \xi_{21} \leqslant 0$ and similarly for $n_{43}$.

## 6.5 More branches of $\log \vartheta_1$

We next analyze the correct branch of the integrand in these regions. We have as before, see eq. (4.16)

$$\log \vartheta_1((n+\xi)\tau, \tau - \tfrac{c}{d}) = -\pi i \tau (n+\xi)^2 + \pi i \tau (\xi - \tfrac{1}{2})^2 + \tfrac{\pi i}{2} - \tfrac{\pi i d}{4c} - 2\pi i \sum_{m=1}^{n} (\!(\tfrac{md}{c})\!)$$

$$+ \log\left[\prod_{\ell=1}^{\infty} \left(1 - e^{-\frac{2\pi i d \ell}{c}} q^{\ell}\right)\left(1 - e^{-\frac{2\pi i d(\ell+n)}{c}} q^{\ell-\xi}\right)\left(1 - e^{-\frac{2\pi i d(\ell-n-1)}{c}} q^{\ell+\xi-1}\right)\right], \quad (6.24)$$

where the branch on the right hand side is the correct one that comes from smoothly following the branch between $\xi = -n$ and $0 < \xi < 1$. We similarly want to relate the branches of $\log \vartheta_1((n+\xi)\tau + \tfrac{1}{2c}, \tau - \tfrac{d}{c})$ for different values of $n$. We claim

$$\log \vartheta_1((n+\xi)\tau + \tfrac{1}{2c}, \tau - \tfrac{c}{d}) = -\pi i \tau (n+\xi)^2 + \pi i \tau (\xi - \tfrac{1}{2})^2 + \tfrac{\pi i}{2} - \tfrac{\pi i(d+2)}{4c} - 2\pi i \sum_{m=1}^{n} (\!(\tfrac{2md+1}{2c})\!)$$

$$+ \log\left[\prod_{\ell=1}^{\infty} \left(1 - e^{-\frac{2\pi i d \ell}{c}} q^{\ell}\right)\left(1 - e^{-\frac{\pi i(2d(\ell+n)+1)}{c}} q^{\ell-\xi}\right)\left(1 - e^{-\frac{\pi i(2d(\ell-n-1)-1)}{c}} q^{\ell+\xi-1}\right)\right], \quad (6.25)$$

The argument is by induction over $n$ and identical to the one given in Section 4.3. Thus we will not repeat the argument here.

Taking (4.16) and (6.25) now gives the following formula for the contribution of $\tilde{A}^{n_1,n_2,n_3}_{a/c}$ to the non-planar amplitude:

$$\tilde{A}^{n_1,n_2,n_3}_{a/c} = \frac{i}{32} \int_{\longrightarrow} \frac{d\tau}{c^5 \tau^2} \int d\xi_1 \, d\xi_2 \, d\xi_3 \, e^{-\frac{\pi i s(a+2)}{2c} + \pi i s + \sum_{i>j} s_{ij} \sum_{m=1}^{n_{ij}} (\!(\frac{2md+\delta(i,j)}{2c})\!)}$$

$$\times \prod_{i>j} q^{-\frac{1}{2}s_{ij}\xi_{ij}(\xi_{ij}-1)} \prod_{\ell=1}^{\infty} (1-e^{-\frac{\pi i(2d(\ell+n_{ij})+\delta(i,j))}{c}} q^{\ell-\xi_{ij}})^{-s_{ij}}$$

$$\times (1-e^{-\frac{\pi i(2d(\ell-n_{ij}-1)-\delta(i,j))}{c}} q^{\ell+\xi_{ij}-1})^{-s_{ij}}.\tag{6.26}$$

## 6.6 Evaluating the contribution from fixed $(n_1, n_2, n_3)$ in the $s$-channel

Next, we evaluate the contribution from a given $(n_1, n_2, n_3)$. This is very similar to the situation for the planar annulus. As discussed repeatedly, we can just do the $q$-expansion of the integrand, except for the factors involving $q^{\xi_{21}}$ or $q^{\xi_{43}}$, since those may go to zero. As before, when extracting the coefficient of the term $q^{m_L\xi_{21}+m_D\xi_{32}+m_R\xi_{43}+m_U(1-\xi_{41})}$, we get

$$[q^{m_L\xi_{21}+m_D\xi_{32}+m_R\xi_{43}+m_U(1-\xi_{41})}] \prod_{i>j} \prod_{\ell=1}^{\infty} \left(1-e^{-\frac{\pi i(2d(\ell+n_{ij})+\delta(i,j))}{c}} q^{\ell-\xi_{ij}}\right)^{-s_{ij}}$$

$$\times \prod_{\ell=2-\delta(i,j)}^{\infty} \left(1-e^{-\frac{\pi i(2d(\ell-n_{ij}-1)-\delta(i,j))}{c}} q^{\ell+\xi_{ij}-1}\right)^{-s_{ij}}$$

$$= e^{\frac{\pi i}{c}(2m_L dn_{21}+m_D(2dn_{32}+1)+2m_R dn_{43}-m_U(2d(n_{41}+1)+1))} Q_{m_L,m_D,m_R,m_U}(s,t),\tag{6.27}$$

where we used the definition of $Q_{m_L,m_D,m_R,m_U}$ given in (3.27). We thus have

$$\tilde{A}_{a/c}^{n_1,n_2,n_3} = \frac{i}{32c^5} e^{-\frac{\pi is(a+2)}{2c}+\pi is+2\pi i\sum_{i>j} s_{ij}\sum_{m=1}^{n_{ij}}((\frac{2md+\delta(i,j)}{2c}))} \sum_{m_L,m_D,m_R,m_U \geqslant 0} Q_{m_L,m_D,m_R,m_U}(s,t)$$

$$\times e^{\frac{\pi i}{c}(2m_L dn_{21}+m_D(2dn_{32}+1)+2m_R dn_{43}-m_U(2d(n_{41}+1)+1))}$$

$$\times \int_{\longrightarrow} \frac{d\tau}{\tau^2} \int d\xi_1 d\xi_2 d\xi_3 \, q^{-\sum_{i>j}\frac{1}{2}s_{ij}\xi_{ij}(\xi_{ij}-1)+m_L\xi_{21}+m_D\xi_{32}+m_R\xi_{43}+m_U(1-\xi_{41})}$$

$$\left(1-e^{\frac{2\pi i dn_{21}}{c}} q^{\xi_{21}}\right)^{-s} \left(1-e^{\frac{2\pi i dn_{43}}{c}} q^{\xi_{43}}\right)^{-s}.\tag{6.28}$$

This expression is actually imprecise for $n_{21} = c$ or $n_{43} = c$, a case that we did not encounter in the planar amplitude. In this case, we have $\xi_{21} < 0$ and thus we cannot specify the branch using the region $\xi_{21} > 0$. In this case, we can follow the argument above how to relate these two branches and the correct prescription is

$$\left(1-e^{\frac{2\pi i dn_{21}}{c}} q^{\xi_{21}}\right)^{-s} = e^{-2\pi is((\frac{dn_{21}}{c}))} q^{-s\xi_{21}} \left(1-e^{-\frac{2\pi i dn_{21}}{c}} q^{-\xi_{21}}\right)^{-s}.\tag{6.29}$$

This cancels the corresponding phase factor in $e^{-2\pi is\sum_{m=1}^{n_{21}}((\frac{md}{c}))}$. For $n_{21} = c$, we have $((\frac{dn_{21}}{c})) = 0$ according to our definition (3.22) and thus the prefactor does not need modification. For $n_{21} = c$ we then use the right hand side of this equation for the correct branch. A similar comment applies to the case $n_{43} = c$.

We now continue as in the planar case in evaluating the integrals over the $\xi_i$'s. After introducing $\alpha_L$, $\alpha_R$, $t_L$ and $t_R$ as in Section 5.5, this is reduced to the computation of the integrals

$$\int_{-\infty(0)}^{\infty(0)} d\alpha \, q^{-\alpha t} \left(1-e^{2\pi i\varphi} q^{\alpha}\right)^{-s}.\tag{6.30}$$

There is now a new case appearing corresponding to $n_{21} = c$ or $n_{43} = c$ in which the upper limit of the integral is 0. In this case, we should change the specification of the branch as discussed above by first factoring out $e^{-2\pi is((\varphi))} q^{-\alpha s}$. We then have for $\varphi \in \mathbb{Z}$:

$$\int_{-\infty}^{0} d\alpha \, q^{-\alpha(s+t)}(1-q^{-\alpha})^{-s} = \frac{i}{2\pi\tau}\int_0^1 dx \, x^{s+t-1}(1-x)^{-s} \tag{6.31a}$$

$$= \frac{i}{2\pi\tau}\frac{\Gamma(1-s)\Gamma(s+t)}{\Gamma(t+1)} \tag{6.31b}$$

$$= -\frac{\sin(\pi t)}{\sin(\pi s)}\frac{i}{2\pi\tau}\frac{\Gamma(-t)\Gamma(s+t)}{\Gamma(s)}. \tag{6.31c}$$

Thus we get

$$\tilde{A}_{a/c}^{n_1,n_2,n_3} = -\frac{\pi i \, e^{-\frac{\pi i s(a+2)}{2c}+\pi i s+2\pi i \sum_{i>j}s_{ij}\sum_{m=1}^{n_{ij}}((\frac{2md+\delta(i,j)}{2c}))}}{30c^5\sqrt{stu}}\sum_{\substack{m_L,m_D,m_R,m_U\geqslant 0 \\ (\sqrt{m_D}+\sqrt{m_U})^2\leqslant s}}Q_{m_L,m_D,m_R,m_U}(s,t)$$

$$\times(-1)^{m_L+m_R}e^{\frac{\pi i}{c}(m_D(2dn_{32}+1)-m_U(2d(n_{41}+1)+1))}\int_{P_{m_D,m_U}>0}dt_L \, dt_R \, P_{m_D,m_U}(s,t,t_L,t_R)^{\frac{5}{2}}$$

$$\times\left(\begin{cases}e^{2\pi i t_L((\frac{dn_{21}}{c}))}, & \text{if } 0<n_{21}<c \\ \frac{\sin(\pi(s+t_L))}{\sin(\pi s)}, & \text{if } n_{21}=0 \\ -\frac{\sin(\pi t_L)}{\sin(\pi s)}, & \text{if } n_{21}=c\end{cases}\right)\left(\begin{cases}e^{2\pi i t_R((\frac{dn_{43}}{c}))}, & \text{if } 0<n_{43}<c \\ \frac{\sin(\pi(s+t_R))}{\sin(\pi s)}, & \text{if } n_{43}=0 \\ -\frac{\sin(\pi t_R)}{\sin(\pi s)}, & \text{if } n_{43}=c\end{cases}\right)$$

$$\times\frac{\Gamma(-t_L+m_L)\Gamma(-t_R+m_R)\Gamma(s+t_L-m_L)\Gamma(s+t_R-m_R)}{\Gamma(s)^2}. \tag{6.32}$$

We now recall the definition of (3.28) to perform the sum over $m_L$ and $m_R$. This gives

$$\tilde{A}_{a/c}^{n_1,n_2,n_3} = -\frac{\pi i \, e^{-\frac{\pi i s(a+2)}{2c}+\pi i s+2\pi i \sum_{i>j}s_{ij}\sum_{m=1}^{n_{ij}}((\frac{2md+\delta(i,j)}{2c}))}}{30c^5\sqrt{stu}}\sum_{\substack{m_D,m_U\geqslant 0 \\ (\sqrt{m_D}+\sqrt{m_U})^2\leqslant s}}e^{\frac{\pi i}{c}(m_D-m_U)}$$

$$\times e^{\frac{2\pi i d}{c}(m_D n_{32}-m_U(n_{41}+1))}\int_{P_{m_D,m_U}>0}dt_L \, dt_R \, P_{m_D,m_U}(s,t,t_L,t_R)^{\frac{5}{2}}Q_{m_D,m_U}(s,t,t_L,t_R)$$

$$\times\left(\begin{cases}e^{2\pi i t_L((\frac{dn_{21}}{c}))}, & \text{if } 0<n_{21}<c \\ \frac{\sin(\pi(s+t_L))}{\sin(\pi s)}, & \text{if } n_{21}=0 \\ -\frac{\sin(\pi t_L)}{\sin(\pi s)}, & \text{if } n_{21}=c\end{cases}\right)\left(\begin{cases}e^{2\pi i t_R((\frac{dn_{43}}{c}))}, & \text{if } 0<n_{43}<c \\ \frac{\sin(\pi(s+t_R))}{\sin(\pi s)}, & \text{if } n_{43}=0 \\ -\frac{\sin(\pi t_R)}{\sin(\pi s)}, & \text{if } n_{43}=c\end{cases}\right)$$

$$\times\frac{\Gamma(-t_L)\Gamma(-t_R)\Gamma(s+t_L-m_D-m_U)\Gamma(s+t_R-m_D-m_U)}{\Gamma(s)^2}. \tag{6.33}$$

It does not seem particularly fruitful to express things in terms of $n_L$, $n_D$, $n_R$ and $n_U$ in this case. Their geometric meaning is also much less clear, since there are two boundaries in this case. Let us merely note that

$$\sum_{m=0}^{c-1}\left(\left(\frac{2md+1}{2c}\right)\right) = \sum_{m=0}^{c-1}\left(\left(\frac{2m+1}{2c}\right)\right) \tag{6.34}$$

$$= \frac{1}{2}\sum_{m=0}^{c-1}\left[\left(\left(\frac{2m+1}{2c}\right)\right)+\left(\left(\frac{2(-m-1)+1}{2c}\right)\right)\right] = 0, \tag{6.35}$$

and thus the overall phase only depends on $n_{31}$, $n_{32}$, $n_{42}$ and $n_{41}$ mod $c$.

## 6.7 Results in the $u$-channel

We finally treat the $u$-channel of the non-planar annulus diagram. Start again with (6.20). The regions $\Gamma_{n_1,n_2,n_3}$ are analogous to the $u$-channel regions of the planar annulus, because the

two integrals only differ by phases. In particular, the region $\Gamma_{n_1,n_2,n_3}$ is also specified by (5.43), except that the $\zeta_i$'s now play the role of $z_i$'s. There are $c^3$ regions in total. We can label them by with

$$0 \leqslant n_{21} \leqslant c-1\,, \qquad 0 \leqslant n_{32} \leqslant c-1\,, \qquad 0 \leqslant n_{43} \leqslant c-1\,. \tag{6.36}$$

Now essentially the same formula as (5.44) holds for the non-planar case as well, except that various phases are modified accordingly:

$$
\begin{aligned}
\tilde{A}_{a/c}^{n_1,n_2,n_3} = {}& \frac{i}{32}\, e^{-\frac{\pi i s(a+2)}{2c}+\pi i s} \int_{\longrightarrow} \frac{d\tau}{c^5\tau^2} \int d\xi_1\, d\xi_2\, d\xi_3 \\
& \times \prod_{i>j} q^{-\frac{1}{2}s_{ij}\xi_{ij}(\xi_{ij}-1)} e^{2\pi i s_{ij}\sum_{m=1}^{n_{ij}-\delta_{ij,32}}\left(\!\left(\frac{2md+\delta(i,j)}{2c}\right)\!\right)} \\
& \times \prod_{\substack{i>j \\ ij\neq 32}} \prod_{\ell=1}^{\infty} \left(1-e^{-\frac{\pi i(2d(\ell+n_{ij})+\delta(i,j))}{c}} q^{\ell-\xi_{ij}}\right)^{-s_{ij}} \\
& \qquad\qquad \times \left(1-e^{-\frac{\pi i(2d(\ell-n_{ij}-1)-\delta(i,j))}{c}} q^{\ell+\xi_{ij}-1}\right)^{-s_{ij}} \\
& \times \prod_{\ell=1}^{\infty} \left(1-e^{-\frac{\pi i(2d(\ell+n_{32}-1)+1)}{c}} q^{\ell-\xi_{32}-1}\right)^{-s_{32}} \\
& \qquad\qquad \times \left(1-e^{-\frac{\pi i(2d(\ell-n_{32})-1)}{c}} q^{\ell+\xi_{32}}\right)^{-s_{32}}. \tag{6.37}
\end{aligned}
$$

One can then give the same argument as in the planar case to relate this to the non-planar $s$-channel amplitude. Let us exchange all quantities labeled with 2 with all the corresponding quantities labelled with 3. This maps $\delta(i,j)$ to

$$
\tilde{\delta}(i,j) = \begin{cases} 1\,, & ij \in \{21, 41, 43\}\,, \\ -1\,, & ij = 32\,, \\ 0\,, & ij \in \{31, 42\}\,. \end{cases} \tag{6.38}
$$

One obtains

$$
\begin{aligned}
\tilde{A}_{a/c}^{n_1,n_2,n_3}\bigg|_{2\leftrightarrow 3} = {}& \frac{i}{32}\, e^{-\frac{\pi i u(a+2)}{2c}+\pi i u} \int \frac{d\tau}{c^5\tau^2} \int d\xi_1\, d\xi_2\, d\xi_3 \prod_{i>j}\prod_{i>j} q^{-\frac{1}{2}s_{ij}\xi_{ij}(\xi_{ij}-1)} \\
& \times e^{2\pi i \sum_{i>j,\, ij\neq 32} s_{ij}\sum_{m=1}^{n_{ij}}\left(\!\left(\frac{2md+\tilde{\delta}(i,j)}{2c}\right)\!\right)+2\pi i u \sum_{m=1}^{-n_{32}-1}\left(\!\left(\frac{2md+1}{2c}\right)\!\right)} \\
& \times \prod_{i>j}\prod_{\ell=1}^{\infty} \left(1-e^{-\frac{\pi i(2d(\ell+n_{ij})+\tilde{\delta}(i,j))}{c}} q^{\ell-\xi_{ij}}\right)^{-s_{ij}} \\
& \qquad\qquad \times \left(1-e^{-\frac{\pi i d((\ell-n_{ij}-1)-\tilde{\delta}(i,j))}{c}} q^{\ell+\xi_{ij}-1}\right)^{-s_{ij}}. \tag{6.39}
\end{aligned}
$$

Up to slightly different phases, this coincides with the $s$-channel expression. In particular, we can proceed as before and get

$$
\begin{aligned}
\tilde{A}_{a/c}^{n_1,n_2,n_3}\bigg|_{2\leftrightarrow 3} = {}& -\frac{\pi i\, e^{-\frac{\pi i u(a+2)}{2c}+\pi i u+2\pi i \sum_{i>j,\, ij\neq 32} s_{ij}\sum_{m=1}^{n_{ij}}\left(\!\left(\frac{2md+\tilde{\delta}(i,j)}{2c}\right)\!\right)+2\pi i u \sum_{m=1}^{-n_{32}-1}\left(\!\left(\frac{2md+1}{2c}\right)\!\right)}}{30c^5\sqrt{stu}} \\
& \times \sum_{\substack{m_{\mathrm{L}},m_{\mathrm{D}},m_{\mathrm{R}},m_{\mathrm{U}}\geqslant 0 \\ (\sqrt{m_{\mathrm{D}}}+\sqrt{m_{\mathrm{U}}})^2\leqslant s}} e^{\frac{\pi i}{c}(m_{\mathrm{L}}(2dn_{21}+1)+m_{\mathrm{D}}(2dn_{32}-1)+m_{\mathrm{R}}(2dn_{43}+1)-m_{\mathrm{U}}(2d(n_{41}+1)+1))}
\end{aligned}
$$

$$\times Q_{m_{\rm L},m_{\rm D},m_{\rm R},m_{\rm U}}(s,t)\int_{P_{m_{\rm D},m_{\rm U}}>0} dt_{\rm L}\, dt_{\rm R}\, P_{m_{\rm D},m_{\rm U}}(s,t,t_{\rm L},t_{\rm R})^{\frac{5}{2}}$$
$$\times e^{2\pi i(t_{\rm L}-m_{\rm L})((\frac{2dn_{21}+1}{2c}))+2\pi i(t_{\rm R}-m_{\rm R})((\frac{2dn_{43}+1}{2c}))}$$
$$\times \frac{\Gamma(-t_{\rm L}+m_{\rm L})\Gamma(-t_{\rm R}+m_{\rm R})\Gamma(s+t_{\rm L}-m_{\rm L})\Gamma(s+t_{\rm R}-m_{\rm R})}{\Gamma(s)^2}. \tag{6.40}$$

The phases again partially cancel and we can perform the sum over $m_{\rm L}$ and $m_{\rm R}$. We obtain

$$\tilde{A}^{n_1,n_2,n_3}_{a/c}\Big|_{2\leftrightarrow3} = -\frac{\pi i\, e^{-\frac{\pi i u(a+2)}{2c}+\pi i u+2\pi i\sum_{i>j,ij\neq32}s_{ij}\sum_{m=1}^{n_{ij}}((\frac{2md+\tilde{\delta}(i,j)}{2c}))+2\pi i u\sum_{m=1}^{-n_{32}-1}((\frac{2md+1}{2c}))}}{30c^5\sqrt{stu}}$$
$$\times \sum_{\substack{m_{\rm D},m_{\rm U}\geqslant0\\(\sqrt{m_{\rm D}}+\sqrt{m_{\rm U}})^2\leqslant s}} e^{\frac{\pi i}{c}(-m_{\rm D}-m_{\rm U})+\frac{2\pi i d}{c}(m_{\rm D}n_{32}-m_{\rm U}(n_{41}+1))}$$
$$\times \int_{P_{m_{\rm D},m_{\rm U}}>0} dt_{\rm L}\, dt_{\rm R}\, P_{m_{\rm D},m_{\rm U}}(s,t,t_{\rm L},t_{\rm R})^{\frac{5}{2}}Q_{m_{\rm D},m_{\rm U}}(s,t,t_{\rm L},t_{\rm R})$$
$$\times e^{2\pi i t_{\rm L}((\frac{2dn_{21}+1}{2c}))+2\pi i t_{\rm R}((\frac{2dn_{43}+1}{2c}))}$$
$$\times \frac{\Gamma(-t_{\rm L})\Gamma(-t_{\rm R})\Gamma(s+t_{\rm L}-m_{\rm L}-m_{\rm R})\Gamma(s+t_{\rm R}-m_{\rm L}-m_{\rm R})}{\Gamma(s)^2}. \tag{6.41}$$

We can now swap the labels 2 and 3 back, which leads to

$$\tilde{A}^{n_1,n_2,n_3}_{a/c} = -\frac{\pi i\, e^{-\frac{\pi i s(a+2)}{2c}+\pi i s+2\pi i\sum_{i>j}s_{ij}\sum_{m=1}^{n_{ij}-\delta_{ij,32}}((\frac{2md+\delta(i,j)}{2c}))}}{30c^5\sqrt{stu}}$$
$$\times \sum_{\substack{m_{\rm D},m_{\rm U}\geqslant0\\(\sqrt{m_{\rm D}}+\sqrt{m_{\rm U}})^2\leqslant s}} e^{\frac{\pi i}{c}(-m_{\rm D}-m_{\rm U})+\frac{2\pi i d}{c}(-m_{\rm D}n_{32}-m_{\rm U}(n_{41}+1))}$$
$$\times \int_{P_{m_{\rm D},m_{\rm U}}(u,t,t_{\rm L},t_{\rm R})>0} dt_{\rm L}\, dt_{\rm R}\, P_{m_{\rm D},m_{\rm U}}(u,t,t_{\rm L},t_{\rm R})^{\frac{5}{2}}Q_{m_{\rm D},m_{\rm U}}(u,t,t_{\rm L},t_{\rm R})$$
$$\times e^{2\pi i t_{\rm L}((\frac{2dn_{31}+1}{2c}))+2\pi i t_{\rm R}((\frac{2dn_{42}+1}{2c}))}$$
$$\times \frac{\Gamma(-t_{\rm L})\Gamma(-t_{\rm R})\Gamma(u+t_{\rm L}-m_{\rm L}-m_{\rm R})\Gamma(u+t_{\rm R}-m_{\rm L}-m_{\rm R})}{\Gamma(u)^2}. \tag{6.42}$$

# 7 Mass shifts and decay widths

With the formulas we derived, we will cross check them first in a simpler case, namely in the computation of the mass shifts and decay widths. They appear as the double residue of the amplitude at integer $s$ in the $s$-channel. This leads to lots of classical number theory.

## 7.1 Double residues in terms of Gauss sums

To illustrate the procedure, we first discuss the mass-shift at $s=1$. Only terms with $n_{\rm L}=n_{\rm R}=0$ can contribute to the double residue in (3.25). It is straightforward to take the double residue and derive eq. (3.30) from it.

To obtain the mass-shift at $s=1$, we only need the term with $m_{\rm D}=m_{\rm U}=0$. It thus remains to compute the integral

$$\int_{P_{0,0}>0} dt_{\rm L}\, dt_{\rm R}\, P_{0,0}(s,t,t_{\rm L},t_{\rm R})^{\frac{5}{2}}. \tag{7.1}$$

Since the region $P_{0,0} > 0$ is bounded by an ellipse, we can change coordinates to map it to the unit circle, which gives immediately

$$\int_{P_{0,0}>0} dt_L \, dt_R \, P_{0,0}(s,t,t_L,t_R)^{\frac{5}{2}} = \frac{s^4\sqrt{stu}}{64} \int_{\mathbb{D}} dx \, dy \, (1-x^2-y^2)^{\frac{5}{2}} = \frac{\pi s^4\sqrt{stu}}{224}, \tag{7.2}$$

where $\mathbb{D} = \{x^2 + y^2 < 1\}$. We thus have simply

$$\mathrm{DRes}_{s=1} A^{0,n_D,0,n_U} = -\frac{\pi^2 i \, e^{\frac{2\pi i d}{c} n_D n_U}}{210 c^5}. \tag{7.3}$$

Note that we can omit the sawtooth function $((x))$ in the exponent since $s$ is integer. Given that $n_U = c - 1 - n_D$, it is more convenient to write this entirely in terms of $n \equiv n_D$. To obtain the double residue, we have to sum over $n \in \{0, \dots, c-1\}$, $d$ and $c$. The sum over $n$ is known as a Gauss sum:

$$G(-d,-d,c) \equiv \sum_{n=0}^{c-1} e^{-\frac{2\pi i n(n+1)d}{c}}. \tag{7.4}$$

The general notation will be explained below. These are very classical objects in number theory. We also have

$$\mathrm{DRes}_{s=1} \Delta A^p = -\frac{i}{(2\pi)^2} \mathrm{Res}_{s=1} \frac{\Gamma(1-s)\Gamma(-t)}{\Gamma(1-s-t)} = \frac{i}{(2\pi)^2}. \tag{7.5}$$

Hence we obtain

$$\mathrm{DRes}_{s=1} A^p = \frac{i}{(2\pi)^2} - \sum_{c=1}^{\infty} \sum_{a=1,(a,c)=1}^{\frac{c}{2}} \frac{\pi^2 i \, G(-a^*,-a^*,c)}{210 c^5}, \tag{7.6}$$

where $a^*$ is the modular image of $a$ mod $c$, i.e., $aa^* = 1$ mod $c$.

More generally, we can evaluate the double residues at higher integer levels $s_* \in \mathbb{Z}_{>0}$. Their real and imaginary parts correspond to mass shifts and decay widths respectively. The general formula is

$$\mathrm{DRes}_{\substack{s=s_* \\ a/c}} A^{0,n,0,c-1-n} = -\frac{\pi i e^{-4\pi i s_* \sum_{m=1}^{n}((\frac{md}{c}))}}{60 c^5 s_*^2 (s_*!)^2} \sum_{\substack{m_D,m_U \geqslant 0 \\ (\sqrt{m_D}+\sqrt{m_U})^2 \leqslant s_*}} \Delta_{m_D,m_U}(s_*)^{\frac{7}{2}} \, e^{\frac{2\pi i d}{c}(m_D n - m_U(n+1))}$$

$$\times \int_{\mathbb{D}} dx \, dy \, (1-x^2-y^2)^{\frac{5}{2}} Q_{m_D,m_U}(s_*,t,t_L,t_R)$$

$$\times (t_L+1)_{s_*-m_D-m_U-1}(t_R+1)_{s_*-m_D-m_U-1}, \tag{7.7}$$

where

$$t_{L,R} = \frac{\sqrt{\Delta_{m_D,m_U}(s_*)}}{2\sqrt{s_*}}\left(\sqrt{s_*+t}\, x \pm \sqrt{-t}\, y\right) + \frac{1}{2}(m_D + m_U - s_*), \tag{7.8}$$

and $\Delta_{m_D,m_U}$ was defined in eq. (4.37). Since $s_*$ is integer, we can simplify the sum of the sawtooth function:

$$e^{-4\pi i s_* \sum_{m=1}^{n}((\frac{md}{c}))} = e^{-\frac{2\pi i s d n(n+1)}{c}}. \tag{7.9}$$

At this point, we can perform the sum over $n$. They are classical Gauss sums:

$$G(-ds_*, d(m_D - m_U - s_*), c) = \sum_{n=0}^{c-1} e^{-\frac{2\pi i d n(s_* n + s_* - m_D + m_U)}{c}}. \tag{7.10}$$

Putting everything together, we obtain

$$
\begin{aligned}
\operatorname*{DRes}_{s=s_*} A^{\mathrm p} &= \frac{i}{(2\pi)^2}\frac{\Gamma(t+s_*)}{\Gamma(t+1)\Gamma(s)} - \sum_{c=1}^{\infty}\sum_{a=1,(a,c)=1}^{\frac{c}{2}}\frac{\pi i}{60c^5 s_*^2 (s_*!)^2}\sum_{\substack{m_{\mathrm D},m_{\mathrm U}\geqslant 0 \\ (\sqrt{m_{\mathrm D}}+\sqrt{m_{\mathrm U}})^2\leqslant s_*}}\Delta_{m_{\mathrm D},m_{\mathrm U}}(s_*)^{\frac{7}{2}} \\
&\quad\times e^{-\frac{2\pi i s_* m_{\mathrm U} a^*}{c}}G(-a^* s_*, a^*(m_{\mathrm D}-m_{\mathrm U}-s_*),c)\int_{\mathbb D}\mathrm{d}x\,\mathrm{d}y\,(1-x^2-y^2)^{\frac{5}{2}} \\
&\quad\times Q_{m_{\mathrm D},m_{\mathrm U}}(s_*,t,t_{\mathrm L},t_{\mathrm R})(t_{\mathrm L}+1)_{s_*-m_{\mathrm D}-m_{\mathrm U}-1}(t_{\mathrm R}+1)_{s_*-m_{\mathrm D}-m_{\mathrm U}-1}. \qquad (7.11)
\end{aligned}
$$

The integrals for integer $s_*$ can always be performed analytically and for a given mass level this expression can be further simplified. For example, for the first few mass levels, we get

$$
\begin{aligned}
\operatorname*{DRes}_{s=2} A &= \frac{i(1+t)}{(2\pi)^2} - \frac{i\pi^2(1+t)}{3780}\sum_{c=1}^{\infty}\frac{1}{c^5}\sum_{a=1,(a,c)=1}^{\frac{c}{2}}\Big[G(-2a^*,-a^*,c) \\
&\quad + 16G(-2a^*,-2a^*,c) + e^{-\frac{2\pi i a^*}{c}}G(-2a^*,-a^*,c)\Big], \qquad (7.12a)
\end{aligned}
$$

$$
\begin{aligned}
\operatorname*{DRes}_{s=3} A &= \frac{i(1+t)(2+t)}{8\pi^2} - \frac{i\pi^2(1+t)(2+t)}{4490640}\sum_{c=1}^{\infty}\frac{1}{c^5}\sum_{a=1,(a,c)=1}^{\frac{c}{2}} \\
&\quad\times\Big[113G(-3a^*,-a^*,c)+2048G(-3a^*,-2a^*,c)+6561G(-3a^*,-3a^*,c) \\
&\quad + 2048e^{-\frac{2i\pi a^*}{c}}G(-3a^*,-4d,c)+113e^{-\frac{4i\pi a^*}{c}}G(-3a^*,-5a^*,c)\Big], \qquad (7.12b)
\end{aligned}
$$

$$
\begin{aligned}
\operatorname*{DRes}_{s=4} A &= \frac{i(1+t)(2+t)(3+t)}{24\pi^2} - \frac{i\pi^2(2+t)}{39852933120}\sum_{c=1}^{\infty}\frac{1}{c^5}\sum_{a=1,(a,c)=1}^{\frac{c}{2}} \\
&\quad\times\Big[\big(103827t^2+415308t+309568\big)G(-4a^*,-a^*,c) \\
&\quad + 22528\big(87t^2+348t+272\big)G(-4a^*,-2a^*,c) \\
&\quad + 19683\big(405t^2+1620t+1216\big)G(-4a^*,-3a^*,c) \\
&\quad + 524288\big(24t^2+96t+71\big)G(-4a^*,-4a^*,c) \\
&\quad + 19683\big(405t^2+1620t+1216\big)e^{-\frac{2i\pi a^*}{c}}G(-4a^*,-5a^*,c) \\
&\quad + 22528\big(87t^2+348t+272\big)e^{-\frac{4i\pi a^*}{c}}G(-4a^*,-6a^*,c) \\
&\quad + \big(103827t^2+415308t+309568\big)e^{-\frac{6i\pi a^*}{c}}G(-4a^*,-7a^*,c)\Big]. \qquad (7.12c)
\end{aligned}
$$

Here,

$$
G(a,b,c) = \sum_{n=0}^{c}e^{\frac{2\pi i(an^2+bn)}{c}} \qquad (7.13)
$$

denotes the general quadratic Gauss sum. As we will explain below, it can be efficiently calculated. The first three mass-levels are special since the degeneracy is not lifted and the double residues have a factorized form. Starting from $s=5$, the expressions also contain square roots, but they become too unwieldy to display here. Expressions up to $s_*\leqslant 16$ are provided in the ancillary file DRes.txt. We also evaluate them numerically to high precision with the results shown in App. D.

## 7.2 Gauss sums

To continue, it is useful to recall some elementary number theory that allows us to evaluate Gauss sums. We refer to any book on number theory for more details. Let us define the

Legendre symbol for any odd prime $p$ as follows:

$$\left(\frac{a}{p}\right) = \begin{cases} 0, & a \equiv 0 \bmod p, \\ 1, & a \text{ is a quadratic residue mod } p, \\ -1, & a \text{ is not a quadratic residue mod } p. \end{cases} \tag{7.14}$$

A quadratic residue mod $p$ is by definition an element of the finite field $\mathbb{F}_p$ that has a square root. For example in $\mathbb{F}_5$,

$$1^2 = 1, \quad 2^2 = 4, \quad 3^2 = 4, \quad 4^2 = 1. \tag{7.15}$$

Hence 1 and 4 are quadratic residues mod 5 and correspondingly $\left(\frac{1}{5}\right) = \left(\frac{4}{5}\right) = 1$, whereas $\left(\frac{2}{5}\right) = \left(\frac{3}{5}\right) = -1$.

One then extends the definition to the Jacobi symbol as follows. For an odd integer $n$, let $n = p_1^{m_1} p_2^{m_2} \cdots p_k^{m_k}$ be its prime factorization. Then one defines

$$\left(\frac{a}{n}\right) = \prod_{i=1}^{k} \left(\frac{a}{p_1}\right)^{m_k}. \tag{7.16}$$

The Jacobi symbol is a multiplicative function in both the top and bottom argument,

$$\left(\frac{ab}{n}\right) = \left(\frac{a}{n}\right)\left(\frac{b}{n}\right), \qquad \left(\frac{a}{mn}\right) = \left(\frac{a}{m}\right)\left(\frac{b}{m}\right). \tag{7.17}$$

Famously, it satisfies the law of quadratic reciprocity. For $a$ and $b$ odd coprime integers, one has

$$\left(\frac{a}{b}\right)\left(\frac{b}{a}\right) = (-1)^{\frac{(a-1)(b-1)}{4}}. \tag{7.18}$$

The law of quadratic reciprocity can be exploited to give a fast algorithm to compute the Jacobi symbol (runtime $\mathcal{O}(\log a \log b)$).

Let us recall the definition of the Gauss sum:

$$G(a, b, c) = \sum_{n=0}^{c-1} e^{\frac{2\pi i (an^2 + bn)}{c}}. \tag{7.19}$$

They can be evaluated in closed form in terms of Jacobi symbols. First, we can reduce to the case where $(a, c) = 1$ as follows,

$$G(a, b, c) = \begin{cases} (a, c)\, G\!\left(\frac{a}{(a,c)}, \frac{b}{(a,c)}, \frac{c}{(a,c)}\right), & b \mid (a, c), \\ 0, & \text{otherwise.} \end{cases} \tag{7.20}$$

Assuming $(a, c) = 1$, we can next reduce to the case with $b = 0$ by 'completing the square' in the sum. For odd $c$ and even $b$, this is always possible. We first reduce to these cases by using

$$G(a, b, c) = \begin{cases} 0, & (a, c) = 1, c \equiv 0 \bmod 4 \text{ and } b \text{ odd}, \\ 2 G(2a, b, \frac{c}{2}), & (a, c) = 1, c \equiv 2 \bmod 4 \text{ and } b \text{ odd}. \end{cases} \tag{7.21}$$

We can now assume that $b$ is even or $c$ is odd. This always ensures the existence of a solution to the equation

$$2am + b \equiv 0 \bmod c. \tag{7.22}$$

Indeed, for $c$ odd, 2 is invertible and the solution is given by $m = -2^*a^*b$, where $^*$ denotes the inverse mod $c$. For $b$ even, we can divide the equation by 2 and solve instead $am + \frac{b}{2} \equiv 0 \bmod c$,

which always has a solution. We can then shift the summation variable by $m$ to eliminate the linear term and get

$$G(a, b, c) = e^{\frac{2\pi i(am^2 + bm)}{c}} G(a, 0, c). \tag{7.23}$$

Finally, $G(a, 0, c)$ with $(a, c) = 1$ is computed by the following classical formula,

$$G(a, 0, c) = \begin{cases} 0, & (a, c) = 1 \text{ and } c \equiv 2 \bmod 4, \\ \varepsilon(c)\sqrt{c}\left(\frac{a}{c}\right), & (a, c) = 1 \text{ and } c \text{ odd}, \\ (1 + i)\varepsilon(a)^{-1}\sqrt{c}\left(\frac{c}{a}\right), & (a, c) = 1 \text{ and } c \equiv 0 \bmod 4. \end{cases} \tag{7.24}$$

We used the abbreviation

$$\varepsilon(c) = \begin{cases} 1, & c \equiv 1 \bmod 4, \\ i, & c \equiv 3 \bmod 4. \end{cases} \tag{7.25}$$

This explains how to efficiently compute the Gauss sums appearing in the formulas for the mass-shifts (7.12). We have implemented these formulae to generate results in the Appendix D.

## 7.3 Recovering the decay width

As a consistency check of our analysis, we will now demonstrate that the imaginary part of (7.12) equals the expected value that is obtained by just computing the contribution from the annulus ($c = 1$) and from the Möbius strip ($c = 2$). Let us first note that

$$\operatorname{Re} e^{-\frac{2\pi i s m_{\mathrm{U}} a^*}{c}} G(-a^* s, a^*(m_{\mathrm{D}} - m_{\mathrm{U}} - s), c) = \operatorname{Re} \sum_{n=0}^{c-1} e^{-\frac{2\pi i a^*(sn(n+1) - m_{\mathrm{D}} n + m_{\mathrm{U}}(n+1))}{c}}, \tag{7.26}$$

and this is obviously unchanged when replacing $a^* \to -a^*$. Since the modular inverse of $a$ mod $c$ satisfies $(-a)^* = -a^*$, we could hence sum over $a = 1, \ldots, c$ with $(a, c) = 1$, as long as we compensate by a factor of 2. The exception to this occurs for $c = 1$ and $c = 2$, since extending the summation range to $c$ would count $a = \frac{c}{2}$ only once. For $c \geqslant 3$, $a$ can never be equal to $\frac{c}{2}$, since it would either be non-integer or not be coprime to $c$. Thus, the imaginary part of (7.6) can be written as

$$\operatorname{Im} \operatorname*{DRes}_{s=1} A^{\mathrm{p}} = \frac{\pi^2 G(-1, -1, 1)}{420} - \frac{\pi^2 G(-1, -1, 2)}{420 \cdot 2^5} + \frac{1}{4\pi^2} - \frac{1}{2} \sum_{c=1}^{\infty} \sum_{a=1, (a,c)=1}^{c} \frac{\pi^2 G(-a^*, -a^*, c)}{210 c^5}. \tag{7.27}$$

The first two terms are precisely the contribution from the annulus and the Möbius strip. As expected, the Möbius strip contributes with a negative sign. Thus, it remains to show that the last two terms cancel.

The same logic applies to the higher mass-shifts. Consistency with the previously computed imaginary part requires that when we extend the sum over $a$ up to $c$ and compensate by dividing by 2, then the infinite sum should precisely cancel the simple contribution that comes from $\Delta A^{\mathrm{p}}$. Taking the modular inverse is unnecessary at this point because when $a$ runs over $\mathbb{Z}_c^\times$, then so does $a^*$. Here $\mathbb{Z}_c^\times$ is the set of units in the ring $\mathbb{Z}_c$ (i.e. all elements with $(a, c) = 1$, since those have an inverse mod $c$). Let us hence denote $a^* = d$ in the following.

To summarize, we need to show that

$$\frac{1}{2\pi^2} \overset{!}{=} \sum_{c=1}^{\infty} \sum_{d \in \mathbb{Z}_c^{\times}} \frac{\pi^2 G(-d,-d,c)}{210 c^5}, \tag{7.28a}$$

$$\frac{1+t}{2\pi^2} \overset{!}{=} \frac{\pi^2(1+t)}{3780} \sum_{c=1}^{\infty} \frac{1}{c^5} \sum_{d \in \mathbb{Z}_c^{\times}} \left[ G(-2d,-d,c) + 16 G(-2d,-2d,c) + e^{-\frac{2\pi i d}{c}} G(-2d,-d,c) \right], \tag{7.28b}$$

and so on. We first demonstrate the equality explicitly for $s = 1$ and then explain how it generalizes for higher values of $s$.

### 7.3.1  Case $s = 1$

Let us set

$$F(c) = \frac{1}{c} \sum_{d \in \mathbb{Z}_c^{\times}} G(-d,-d,c) = \frac{1}{c} \sum_{d \in \mathbb{Z}_c^{\times}} \sum_{n=1}^{c} e^{-\frac{2\pi i n(n-1)d}{c}}. \tag{7.29}$$

This definition agrees with (3.32), except for $c = 1$ and $c = 2$. Our aim is to determine $F$ explicitly. The result will be $F(c) = |\mu(c)|$, where $\mu(c)$ is the Möbius function, defined by

$$\mu(n) = \begin{cases} 1, & n \text{ has an even number of prime factors}, \\ -1, & n \text{ has an odd number of prime factors}, \\ 0, & n \text{ has a repeated prime factor}. \end{cases} \tag{7.30}$$

We prove this in two steps. First, we show that $F(c)$ is multiplicative, i.e. for $c = c_1 c_2$ and $(c_1, c_2) = 1$ we have $F(c_1 c_2) = F(c_1) F(c_2)$. The Chinese remainder theorem says that $\mathbb{Z}_{c_1 c_2}^{\times} \cong \mathbb{Z}_{c_1}^{\times} \times \mathbb{Z}_{c_2}^{\times}$, i.e. $d \mapsto (d_1 = d \bmod c_1, d_2 = d \bmod c_2)$ is a group isomorphism. It is the restriction of the corresponding ring isomorphism $\mathbb{Z}_{c_1 c_2} \cong \mathbb{Z}_{c_1} \times \mathbb{Z}_{c_2}$ to the group of units. We also notice that

$$(c_1 + c_2, c_1 c_2) = (c_1 + c_2, c_1)(c_1 + c_2, c_2) = (c_2, c_1)(c_1, c_2) = 1, \tag{7.31}$$

and hence $c_1 + c_2$ is a unit. Thus we may replace $d \in \mathbb{Z}_c^{\times}$ in the sum with $(c_1 + c_2)d$, since both run over the units of $\mathbb{Z}_c$. We then get

$$F(c_1 c_2) = \frac{1}{c_1 c_2} \sum_{n \in \mathbb{Z}_{c_1 c_2}} \sum_{d \in \mathbb{Z}_{c_1 c_2}^{\times}} e^{-\frac{2\pi i n(n-1)(c_1+c_2)d}{c_1 c_2}} \tag{7.32}$$

$$= \frac{1}{c_1 c_2} \sum_{n \in \mathbb{Z}_{c_1 c_2}} \sum_{d \in \mathbb{Z}_{c_1 c_2}^{\times}} e^{-\frac{2\pi i n(n-1)d}{c_1}} e^{-\frac{2\pi i n(n-1)d}{c_2}}. \tag{7.33}$$

Now let $d_i = d \bmod c_i$ and $n_i = n \bmod c_i$. Then with the help of the Chinese remainder theorem we conclude

$$F(c_1 c_2) = \frac{1}{c_1 c_2} \sum_{n_1 \in \mathbb{Z}_{c_1}} \sum_{n_2 \in \mathbb{Z}_{c_2}} \sum_{d_1 \in \mathbb{Z}_{c_1}^{\times}} \sum_{d_2 \in \mathbb{Z}_{c_2}^{\times}} e^{-\frac{2\pi i n_1(n_1-1)d_1}{c_1}} e^{-\frac{2\pi i n_2(n_2-1)d_2}{c_2}} = F(c_1) F(c_2). \tag{7.34}$$

It then remains to evaluate $F(p^k)$ for $p$ prime and $k \geq 1$, since this will determine $F(c)$ completely.

$$F(p^k) = \frac{1}{p^k} \left( \sum_{n \in \mathbb{Z}_{p^k}} \sum_{d \in \mathbb{Z}_{p^k}} e^{-\frac{2\pi i n(n-1)d}{p^k}} - \sum_{n \in \mathbb{Z}_{p^k}} \sum_{p \,|\, d \in \mathbb{Z}_{p^k}} e^{-\frac{2\pi i n(n-1)d}{p^k}} \right) \tag{7.35}$$

$$= \frac{1}{p^k} \left( \sum_{n\in\mathbb{Z}_{p^k}} \sum_{d\in\mathbb{Z}_{p^k}} e^{-\frac{2\pi i n(n-1)d}{p^k}} - \sum_{n\in\mathbb{Z}_{p^k}} \sum_{d\in\mathbb{Z}_{p^{k-1}}} e^{-\frac{2\pi i n(n-1)d}{p^{k-1}}} \right) \tag{7.36}$$

$$= \frac{1}{p^k} \left( \sum_{n\in\mathbb{Z}_{p^k}} p^k \delta_{p^k|n(n-1)} - \sum_{n\in\mathbb{Z}_{p^k}} p^{k-1} \delta_{p^{k-1}|n(n-1)} \right). \tag{7.37}$$

Now $p^k \mid n(n-1)$ precisely for $n=0$ or $n=1 \in \mathbb{Z}_{p^k}$. The same reasoning applies in the second term where $p^{k-1} \mid n(n-1)$ when $n = rp^{k-1}$ or $n = rp^{k-1}+1$ with $r = 0,\dots,p-1$. For $k \geqslant 2$ these are $2p$ possibilities, whereas for $k=1$, these are only $p$ possibilities. Thus

$$F(p^k) = \frac{1}{p^k} \left( 2p^k - (2-\delta_{k,1})\, p \times p^{k-1} \right) = \delta_{k,1} = |\mu(p^k)|. \tag{7.38}$$

Thus by multiplicativity of $F$ and the Moebius function we conclude

$$F(c) = |\mu(c)|, \tag{7.39}$$

for any positive integer $c$. We can then evaluate

$$\sum_{c=1}^{\infty} \frac{|\mu(c)|}{c^4} = \prod_{p\in\mathbb{P}} \sum_{k=0}^{\infty} \mu(p^k) p^{-4k} \tag{7.40}$$

$$= \prod_{p\in\mathbb{P}} (1 + p^{-4}) \tag{7.41}$$

$$= \prod_{p\in\mathbb{P}} \frac{1-p^{-8}}{1-p^{-4}} = \frac{\zeta(4)}{\zeta(8)} = \frac{105}{\pi^4}, \tag{7.42}$$

where we used multiplicativity of $|\mu(c)|$ and the fact that every integer can be uniquely written as the product of its prime factors. We also used the Euler product of the Riemann zeta-function,

$$\zeta(\sigma) = \prod_{p\in\mathbb{P}} (1-p^{-\sigma})^{-1}. \tag{7.43}$$

We thus get

$$\sum_{c=1}^{\infty} \sum_{d\in\mathbb{Z}_c^\times} \frac{\pi^2 G(-d,-d,c)}{210 c^5} = \frac{\pi^2}{210} \sum_{c=1}^{\infty} \frac{|\mu(c)|}{c^4} = \frac{1}{2\pi^2}, \tag{7.44}$$

which demonstrates eq. (7.28a).

### 7.3.2 Higher values of $s$

For higher decay widths we can proceed similarly. We define more generally

$$F_s^{m_D,m_U}(c) = \frac{1}{c} \sum_{d\in\mathbb{Z}_c^\times} e^{-\frac{2\pi i m_U d}{c}} G(-ds, d(m_D - m_U - s), c) \tag{7.45}$$

$$= \frac{1}{c} \sum_{d\in\mathbb{Z}_c^\times} \sum_{n=0}^{c-1} e^{-\frac{2\pi i d(sn(n+1) - m_D n + m_U(n+1))}{c}}, \tag{7.46}$$

which again agrees with the definition (3.32) except for $c = 1$ and $c = 2$. The same argument as for $s = 1$, $m_\mathrm{D} = m_\mathrm{U} = 0$ shows that $F_s^{m_\mathrm{D}, m_\mathrm{U}}(c)$ is a multiplicative function. Thus it suffices again to compute $F_s^{m_\mathrm{D}, m_\mathrm{U}}(c)$ on prime powers. Proceeding as before, this gives

$$F_s^{m_\mathrm{D}, m_\mathrm{U}}(p^k) = \frac{1}{p^k} \sum_{n \in \mathbb{Z}_{p^k}} \left( \sum_{d \in \mathbb{Z}_{p^k}} - \sum_{d \in \mathbb{Z}_{p^k}, d \equiv 0 \bmod p} \right) e^{-\frac{2\pi i d (sn(n+1) - m_\mathrm{D} n + m_\mathrm{U}(n+1))}{p^k}} \tag{7.47}$$

$$= \sum_{n \in \mathbb{Z}_{p^k}} \left( \delta_{p^k | (sn(n+1) - m_\mathrm{D} n + m_\mathrm{U}(n+1))} - \frac{1}{p} \delta_{p^{k-1} | (sn(n+1) - m_\mathrm{D} n + m_\mathrm{U}(n+1))} \right) \tag{7.48}$$

$$= \sum_{n \in \mathbb{Z}_{p^k}} \delta_{p^k | (sn(n+1) - m_\mathrm{D} n + m_\mathrm{U}(n+1))} - \sum_{n \in \mathbb{Z}_{p^{k-1}}} \delta_{p^{k-1} | (sn(n+1) - m_\mathrm{D} n + m_\mathrm{U}(n+1))} . \tag{7.49}$$

We hence need to count the number of solutions to the equation

$$sn^2 + (s - m_\mathrm{D} + m_\mathrm{U})n + m_\mathrm{U} \equiv 0 \bmod p^k . \tag{7.50}$$

This is done in Appendix C. Let us note that the discriminant of this quadratic equation is given by

$$\Delta_{m_\mathrm{D}, m_\mathrm{U}} = \left[ s - (\sqrt{m_\mathrm{D}} + \sqrt{m_\mathrm{U}})^2 \right] \left[ s - (\sqrt{m_\mathrm{D}} - \sqrt{m_\mathrm{U}})^2 \right] . \tag{7.51}$$

Let us first consider the generic case, by which we mean that $\Delta_{m_\mathrm{D}, m_\mathrm{U}} \neq 0 \bmod p$, $p \neq 2$ and $s \neq 0 \bmod p$. Let us denote the set of all the special primes for which this is not the case by $\mathbb{P}_{s, m_\mathrm{D}, m_\mathrm{U}}$. In this case, the number of solutions is independent of $k \geqslant 1$ and is given by the Legendre symbol

$$\left( \frac{\Delta_{m_\mathrm{D}, m_\mathrm{U}}}{p} \right) + 1 . \tag{7.52}$$

This implies that for a generic prime

$$F_s^{m_\mathrm{D}, m_\mathrm{U}}(p^k) = \left( \frac{\Delta_{m_\mathrm{D}, m_\mathrm{U}}}{p} \right) \delta_{k,1} = \left( \frac{\Delta_{m_\mathrm{D}, m_\mathrm{U}}}{p} \right) |\mu(p^k)| . \tag{7.53}$$

For the exceptional primes $\mathbb{P}_{s, m_\mathrm{D}, m_\mathrm{U}}$, the formula for the number of integer solutions is more complicated and explained in Appendix C. It is sufficient here to know that since $\Delta_{s, m_\mathrm{D}, m_\mathrm{U}} \neq 0$ by construction, the number of solutions always stabilizes for $k \geqslant k_0$ and thus $F_s^{m_\mathrm{D}, m_\mathrm{U}}(p^k) = 0$ for sufficiently high $k$. By multiplicativity, we can write the sum involved in the mass-shift in terms of an infinite product over primes,

$$\sum_{c=1}^{\infty} \frac{F_s^{m_\mathrm{D}, m_\mathrm{U}}(c)}{c^4} = \prod_{p \in \mathbb{P}} \left( 1 + \left( \frac{\Delta_{m_\mathrm{D}, m_\mathrm{U}}}{p} \right) p^{-4} \right) \prod_{p \in \mathbb{P}_{s, m_\mathrm{D}, m_\mathrm{U}}} \frac{\sum_{k \geqslant 0} F_s^{m_\mathrm{D}, m_\mathrm{U}}(p^k) p^{-4k}}{1 + \left( \frac{\Delta_{m_\mathrm{D}, m_\mathrm{U}}}{p} \right) p^{-4}} . \tag{7.54}$$

Since there are finitely many exceptions, the second factor on the right hand side is easy to evaluate. It remains to evaluate the first factor. Here $\left( \frac{\Delta}{c} \right)$ appears with $c$ not necessarily odd. This constitutes a generalization of the Jacobi symbol known as the Kronecker symbol. Its definition involves in general several case distinctions, but we do not need all of them because $\Delta > 0$ and $\Delta \equiv 0, 1 \bmod 4$. In this special case, the definition reads

$$\left( \frac{\Delta}{c} \right) = \left( \frac{\Delta}{|c|} \right) = \prod_{p \in \mathbb{P}} \left( \frac{\Delta}{p} \right)^{k_p} , \tag{7.55}$$

where $|c| = \prod_p p^{k_p}$ is the prime factorization of $|c|$. Hence we only need to define

$$\left( \frac{\Delta}{2} \right) = \begin{cases} 0, & \Delta \equiv 0 \bmod 2, \\ 1, & \Delta \equiv \pm 1 \bmod 8, \\ -1, & \Delta \equiv \pm 3 \bmod 8. \end{cases} \tag{7.56}$$

The Kronecker symbol is periodic in $c$ with period $\Delta$ because of quadratic reciprocity (this requires $\Delta \equiv 0, 1 \bmod 4$). We evaluate for $\Delta \equiv 0, 1 \bmod 4$,

$$\prod_{p \in \mathbb{P}} \left(1 - \left(\frac{\Delta}{p}\right) p^{-4}\right)^{-1} = \sum_{c=1}^{\infty} \left(\frac{\Delta}{c}\right) \frac{1}{c^4} \tag{7.57}$$

$$= \frac{1}{2} \sum_{c \in \mathbb{Z} \setminus \{0\}} \operatorname*{Res}_{z=c} \frac{1}{z^4} \sum_{m=0}^{\Delta-1} \left(\frac{\Delta}{m}\right) \frac{\pi}{\Delta} \cot\left(\frac{\pi(z-m)}{\Delta}\right) \tag{7.58}$$

$$= -\frac{1}{2} \operatorname*{Res}_{z=0} \frac{1}{z^4} \sum_{m=1}^{\Delta-1} \left(\frac{\Delta}{m}\right) \frac{\pi}{\Delta} \cot\left(\frac{\pi(z-m)}{\Delta}\right) \tag{7.59}$$

$$= \sum_{(m,\Delta)=1} \frac{\pi^4 (2 + \cos(\frac{2m\pi}{\Delta}))}{6\Delta^4 \sin(\frac{m\pi}{\Delta})^4} \left(\frac{\Delta}{m}\right). \tag{7.60}$$

We then finish the calculation by noting that

$$\prod_{p \in \mathbb{P}} \left(1 + \left(\frac{\Delta}{p}\right) p^{-4}\right) = \prod_{p, \, \Delta \not\equiv 0 \bmod p} \frac{1 - p^{-8}}{1 - \left(\frac{\Delta}{p}\right) p^{-4}} \tag{7.61}$$

$$= \frac{1}{\zeta(8)} \prod_{p \mid \Delta} (1 - p^{-8})^{-1} \sum_{(m,\Delta)=1} \frac{\pi^4 (2 + \cos(\frac{2m\pi}{\Delta}))}{6\Delta^4 \sin(\frac{m\pi}{\Delta})^4} \left(\frac{\Delta}{m}\right). \tag{7.62}$$

Combining (7.54) and (7.62) allows us to compute the sums appearing in the imaginary part of the mass-shift. It is then simple to implement these formulas and check the required identities such as (7.28). We checked that the corresponding identities hold up to $s = 12$.

## 8 Conclusions

In this work, we revisited the formal expression (1.1) describing scattering amplitudes of strings as integrals over the moduli space of Riemann surfaces with punctures. We converted it into a practical formula (1.2) to compute one-loop four-point open-string amplitudes. It can be thought of as a sum over thin worldsheets with a given number of windings, where terms with more and more windings become more suppressed. This formula allowed us to compute the corresponding amplitudes at finite values of $\alpha'$ for the first time, as illustrated on examples in Figures 4, 5, and 6. There are a couple of open questions that we were not able to fully solve as well as a number of future research directions, which we outline below.

**Convergence of the Rademacher expansion.** While we have provided, in our view, strong evidence for the convergence of the Rademacher contour, we were unable to rigorously prove it. As we have also mentioned, the convergence properties deteriorate at low energies, since the phases in the Rademacher expansion tend to be close to unity and do not cancel out. At $s = t = 0$ the convergence completely breaks down, which is the manifestation of the massless branch cut in our formula.

In order to develop this formalism more systematically, it is of vital importance to understand the involved phases better. While the sawtooth function $((x))$ makes a frequent appearance in number theory, the sums in (3.25) at least naively are not easy to bound using standard number-theoretic techniques. One obviously would like to do better than simply argue for the randomness of these phases for high values of $c$. We should mention that there are some cases in the literature where convergence of the Rademacher series for positive weight is established [37].

**Low-energy expansion.** A related issue is the important cross check of making contact with the low-energy expansion of the amplitudes. There is a large body of literature studying the $\alpha'$ expansion of one-loop string amplitudes, see [38–42] for the open string in particular. It seems to be quite hard to extract the low-energy behaviour from the Rademacher formula, since this is the regime where convergence breaks down. It is also not possible to take $\alpha'$ derivatives of our formula and commute the derivatives with the infinite sum over $c$. Every such derivative makes the individual terms grow faster with $c$ and after a sufficient number of derivatives, the sum no longer converges. To better understand the involved subtlety, consider the infinite sum

$$\sum_{n\neq 0}\frac{1}{|n|}e^{inx} = -\log\left(4\sin(\tfrac{x}{2})^2\right), \tag{8.1}$$

which has similar convergence properties as the sums that we encountered in this paper and the breakdown of convergence at $x = 0$ also manifests itself as a logarithmic branch cut. Without knowing the right-hand side, it is equally challenging to extract the series expansion of the left-hand side around $x = 0$, since the sum is divergent as soon as one takes a derivative in $x$ and commutes it with the infinite sum.

**Analytic continuation in $s$ and $t$.** The Rademacher expansion of the planar $s$-channel string amplitude given in eq. (3.25) is only valid for physical kinematics. In fact, it does not even converge when we start to consider complex values of $s$ and $t$, since then the phases start to grow exponentially. However, as is well-known, it is often fruitful to study the extension of the amplitude to the complex Mandelstam invariants and study its analytic properties. In the context of string amplitudes, some analytic properties of the amplitude were studied in [14,32], but a full understanding is missing. The fact that (3.25) cannot be easily analytically continued does not mean that such an analytic continuation does not exist. In fact, one might come to a similar conclusion in the toy example (8.1), but of course the right hand side has a perfectly good analytic continuation with branch points at $x \in 2\pi\mathbb{Z}$, where the convergence of the sum breaks down.

It would be very desirable to have a formula for the string amplitude that holds for arbitrary complex values of $s$ and $t$; or short of that, a way to access the other branches of the amplitude for real values of $s$ and $t$. We expect that deforming the integration contour appropriately can achieve such a goal and plan to report on it elsewhere [13].

**High-energy limit.** Since we now have explicit control over the amplitude at intermediate energies, it is natural to try to extract the asymptotics for very high energies from our formula. The high-energy limit of string amplitudes was first analyzed by Gross and Mende [22] and Gross and Mañes [21] in the case of the open string, where the integral over moduli space was evaluated using saddle-point techniques. As already mentioned in Section 1, performing this computation rigorously is currently out of reach and the results of [21, 22] should be viewed as a heuristic. As we saw in this work, the asymptotic formula of Gross and Mañes seem to be true "on average", but there is a number of very complicated oscillations on top that seem hard to predict from a saddle-point of view.[9]

Our formula seems to open a different avenue to access the high-energy behaviour in detail as we have already demonstrated numerically, see Figure 5. A good understanding of the growth behaviour of the polynomials $Q_{m_{\mathrm{D}},m_{\mathrm{U}}}(s,t,t_{\mathrm{L}},t_{\mathrm{R}})$ for large values of the parameters would give a detailed analytic control over the high-energy limit of the amplitude and can

---

[9]Incidentally, there seems to be a nice analogy with the recent discussion of wormholes in quantum gravity, where the saddle point gives the "averaged" contribution to some quantity such as the spectral form factor, while the true behaviour of the quantity has many erratic oscillations on top of this averaged smooth behaviour, see e.g. [43] and numerous follow-up works.

hopefully make contact with the saddle-point evaluation. Additionally, the integration contour we proposed in this work can be taken as a starting point for a rigorous saddle-point analysis using methods of complex Morse theory.

**Other string topologies.** As an immediate question, one may ask how general the methods employed in this paper are. The logic generalizes straightforwardly to other open string one-loop amplitudes with an arbitrary number of external vertex operators. However, the modular weight of the integrand is at least naively positive for $n \geqslant 5$ external vertex operators and convergence might become even more delicate than in the four-point case. We should however note that this might be misleading. Contrary to the four-point function, the five-point function does not admit a canonical integrand that should be integrated over $\mathcal{M}_{1,5}$, but instead several different representations that differ by total derivatives. It might be more illuminating to think of the integrand as living on the moduli space of super Riemann surfaces as described by [4, 44], where the integrand has a canonical form. Since there is no non-trivial topology in the fermionic directions of moduli space, the contour deformation into the Rademacher contour straightforwardly extends to supermoduli space and its complexification.

The extension to higher loops and closed strings at one loop is much less clear at this stage. One would expect that the general logic that one can derive an infinite sum representation for the string amplitude where every term is controlled by a degeneration in complexified moduli space continuous to hold. To make such an expectation concrete, one needs a version of the Rademacher contour for other genera. For the open string at two loops, this seems to be possible. In [45, 46], the Rademacher expansion for the inverse of the Igusa cusp form $\Phi_{10}^{-1}$ at genus 2 was derived in the context of microstate counting of black holes. This is almost what we need for the partition function of the bosonic open string in analogy to what we discussed in Section 3.2 at one loop. Indeed, the inverse of the Igusa cusp form is the partition function of 24 free bosons.

However, for the closed string, even at genus 1, the mathematical technology needed for this computation is to our knowledge not available in the literature. In the simplest toy model for the partition function of the closed bosonic string, one wants to evaluate the integral

$$\int_{\mathcal{F}} \frac{\mathrm{d}^2\tau}{(\operatorname{Im}\tau)^{14} |\eta(\tau)^{24}|^2} \, , \tag{8.2}$$

over the fundamental domain. One again has to modify the contour near the cusp to implement the $i\varepsilon$ prescription. Let us set $\tau = x + y$ with $x \in \mathbb{R}$ and $y \in i\mathbb{R}$ for the real slice of moduli space. One then allows $x$ and $y$ to be both complex in order to pass to the complexification. The appropriate complexification of the moduli space is given by two copies of the upper half-plane modded out by a single diagonal modular group, $(\mathbb{H} \times \mathbb{H})/\mathrm{PSL}(2,\mathbb{Z})$. The proper contour for the integral (8.2) analogous to the one discussed in Section 3.1 is then

$$\int_{\Gamma} \frac{\mathrm{d}^2\tau}{(\operatorname{Im}\tau)^{14} |\eta(\tau)^{24}|^2} = \int_{\mathcal{F}_L} \frac{\mathrm{d}^2\tau}{(\operatorname{Im}\tau)^{14} |\eta(\tau)^{24}|^2} + \int_{-\frac{1}{2}}^{\frac{1}{2}} \mathrm{d}x \int_{iL-\mathbb{R}_{\geqslant 0}} \mathrm{d}y \, \frac{1}{y^{14} \eta(x+y)^{24} \eta(-x+y)^{24}} \, , \tag{8.3}$$

which is indeed convergent. Here, $\mathcal{F}_L$ is the usual fundamental domain cut off at $\operatorname{Im}\tau = L$ that also often features in other regularizations of the integral over moduli space. While the integral can be easily evaluated numerically (and equals roughly $29399.1 + 98310i$), we are not aware of any exact analytic evaluation of the integral. See however [47] for a term-by-term evaluation in the $q$-expansion, [48] for the evaluation of integrals of holomorphic modular functions using the transformation property of the Eisenstein series $E_2$ and [49, 50] for the application of the Rankin–Selberg method for similar integrals.

**Oscillations and relation to chaos.** We have seen numerically that the real part of the one-loop amplitude features many seemingly erratic oscillations, see Figures 4 and 5. The meaning of these oscillations from a scattering amplitudes point of view is not entirely clear, since there are very few consistency checks that can be performed on the real part of the one-loop amplitude without knowing further data. In particular, it is not directly constrained by unitary. Some constraints are imposed by the analytic structure, which allow one to compute dispersion relations. We will report on these elsewhere [13].

One perspective on these amplitudes is from the point of view of chaos. Going to stronger coupling will eventually make contact with black hole physics (although black holes are non-perturbative in the string coupling of the form $\mathcal{O}(e^{-1/g_s^2})$ and thus not necessarily visible in string perturbation theory). Such a view for tree-level scattering amplitudes with one or more heavy external states was advocated in [51]. We believe that the one-loop amplitude is a much better probe for such chaotic behaviour since it involves arbitrarily massive internal states. It would be interesting to make this link more precise.

# Acknowledgments

We thank Nima Arkani-Hamed, Pinaki Banerjee, Simon Caron-Huot, Eric D'Hoker, Aaron Hillman, Abhiram Kidambi, Juan Maldacena, Giulio Salvatori, Oliver Schlotterer, and Gabriele Veneziano for useful discussions.

**Funding information** L.E. and S.M. are supported by the grant DE-SC0009988 from the U.S. Department of Energy. S.M. gratefully acknowledges funding provided by the Sivian Fund.

# A   Cusp contribution $\Delta A^{\mathrm{p}}$

In this appendix we evaluate (3.4) explicitly. Its content is virtually identical to the Appendix A of [24], but we include the computation for completeness. It is more convenient to fix $z_4 = \frac{1}{2}$ in (3.4) instead of the choice $z_4 = 1$ that we used in the rest of the article. The integration region is then $-\frac{1}{2} \leqslant z_1 \leqslant z_2 \leqslant z_3 \leqslant \frac{1}{2}$. Next, we change variables according to

$$\sigma_i = \tan(\pi z_i). \tag{A.1}$$

We then use the trigonometric identity

$$\sin(\arctan(\sigma_i) - \arctan(\sigma_j)) = \frac{\sigma_i - \sigma_j}{\sqrt{(1 + \sigma_i^2)(1 + \sigma_j^2)}}, \tag{A.2}$$

to rewrite the integral as

$$\Delta A^{\mathrm{p}} = \frac{i}{2\pi^3} \int_{-\infty \leqslant \sigma_1 \leqslant \sigma_2 \leqslant \sigma_3 \leqslant \infty} \prod_{i=1}^{3} \frac{\mathrm{d}\sigma_i}{1 + \sigma_i^2} \left(\frac{\sigma_{21}}{\sigma_{31}}\right)^{-s} \left(\frac{\sigma_{32}}{\sigma_{31}}\right)^{-t}. \tag{A.3}$$

Let us change variables and trade $x = \frac{\sigma_{21}}{\sigma_{31}}$ for $\sigma_2$. We get

$$\Delta A^{\mathrm{p}} = \frac{i}{2\pi^3} \int_0^1 \mathrm{d}x \, x^{-s}(1-x)^{-t} \int_{\sigma_1 \leqslant \sigma_3} \frac{\mathrm{d}\sigma_1 \, \mathrm{d}\sigma_3 \, (\sigma_3 - \sigma_1)}{(1 + \sigma_1^2)(1 + \sigma_3^2)(1 + ((1-x)\sigma_1 + x\sigma_3)^2)}. \tag{A.4}$$

We now want to compute the integral over $\sigma_1$ and $\sigma_3$. We set $\sigma_3 = \sigma_1 + y$, so that the integral over $y$ runs from 0 to $\infty$. The integral over $\sigma_1$ can be then computed first by standard residue

techniques. We close the contour in the upper half-plane and pick up the three poles, located at

$$\sigma_1 = i, \qquad \sigma_1 = i - y, \qquad \sigma_1 = i - xy. \tag{A.5}$$

We thus get the following three contributions corresponding to these three poles,

$$\int_{\sigma_1 \leqslant \sigma_3} \frac{\mathrm{d}\sigma_1 \, \mathrm{d}\sigma_3 \, (\sigma_3 - \sigma_1)}{(1 + \sigma_1^2)(1 + \sigma_3^2)(1 + ((1-x)\sigma_1 + x\sigma_3)^2)}$$

$$= \pi \int_0^\infty \mathrm{d}y \left[ \frac{1}{xy(2i+y)(2i+xy)} + \frac{1}{(1-x)y(2i-y)(2i-y(1-x))} \right.$$

$$\left. - \frac{1}{xy(1-x)(2i-xy)(2i+(1-x)y)} \right] \tag{A.6}$$

$$= 2\pi \int_0^\infty \mathrm{d}y \left[ \frac{y(1+x)}{(y^2+4)(x^2y^2+4)} + \frac{y(2-x)}{(y^2+4)((1-x)^2y^2+4)} \right]. \tag{A.7}$$

The two terms are related by $x \to 1 - x$. For the first term, we have

$$\int_0^\infty \mathrm{d}y \, \frac{y(1+x)}{(y^2+4)(x^2y^2+4)} = \frac{1}{4(1-x)} \int_0^\infty \mathrm{d}y \left[ \frac{y}{y^2+4} - \frac{x^2y}{x^2y^2+4} \right] \tag{A.8}$$

$$= \lim_{y \to \infty} \frac{1}{8(1-x)} \left[ \log(4+y^2) - \log(4+x^2y^2) \right] \tag{A.9}$$

$$= -\frac{\log(x)}{4(1-x)}. \tag{A.10}$$

Thus we conclude

$$\Delta A^{\mathrm{p}} = -\frac{i}{4\pi^2} \int_0^1 \mathrm{d}x \, x^{-s}(1-x)^{-t} \left( \frac{\log(x)}{1-x} + \frac{\log(1-x)}{x} \right) \tag{A.11}$$

$$= \frac{i}{4\pi^2} \int_0^1 \mathrm{d}x \left( \frac{\mathrm{d}}{\mathrm{d}s} x^{-s}(1-x)^{-t-1} + \frac{\mathrm{d}}{\mathrm{d}t} x^{-s-1}(1-x)^{-t} \right) \tag{A.12}$$

$$= \frac{i}{4\pi^2} \left( \frac{\mathrm{d}}{\mathrm{d}s} \frac{\Gamma(1-s)\Gamma(-t)}{\Gamma(1-s-t)} + \frac{\mathrm{d}}{\mathrm{d}t} \frac{\Gamma(-s)\Gamma(1-t)}{\Gamma(1-s-t)} \right). \tag{A.13}$$

We can also write this as

$$\Delta A^{\mathrm{p}} = -\frac{i}{4\pi^2} \frac{\mathrm{d}}{\mathrm{d}\alpha'} \left( (\alpha')^2 \frac{\Gamma(-\alpha's)\Gamma(-\alpha't)}{\Gamma(1-\alpha's-\alpha't)} \right) \bigg|_{\alpha'=1}, \tag{A.14}$$

which shows that this is indeed related to the $\alpha'$-derivative of the tree-level four-point function. The factor $(\alpha')^2$ reflects the fact that the polarization tensor $t_8$ given in eq. (2.6) is quartic in the momenta.

# B Convergence of the Rademacher method

In this Appendix, we discuss convergence of the Rademacher method.

## B.1 Integrals over modular forms

We first look at the case, where we integrate

$$\int_\Gamma \mathrm{d}\tau \, f(\tau), \tag{B.1}$$

where $f$ is modular of weight $k < 0$. We discuss for simplicity the case, where the contour $\Gamma$ starts at $\tau = 0$ and ends at $\tau = 1$, so that the circle $C_{a/c}$ with $0 < \frac{a}{c} \leqslant 1$ appears in the Rademacher contour. The discussion is essentially unchanged if we modify the endpoint of the contour. We allow $f$ to have multiplier phases, i.e. it transforms as

$$f\left(\frac{a\tau + b}{c\tau + d}\right) = \rho\begin{pmatrix} a & b \\ c & d \end{pmatrix}(c\tau + d)^k f(\tau), \tag{B.2}$$

where $\rho : \mathrm{PSL}(2, \mathbb{Z}) \longrightarrow \mathrm{U}(1)$ is a character of $\mathrm{PSL}(2, \mathbb{Z})$ and we pick the principal branch of $(c\tau + d)^k$. Moreover, $f(\tau)$ could have a singularity at $\tau = i\infty$, which is known as a weakly holomorphic modular form. The canonical example is $f(\tau) = \eta(\tau)^{\frac{k}{2}}$, where $\rho$ is given in terms of certain Dedekind sums. Here $k$ can be any real number. This is the case that appears for the string partition function without additional punctures. We claim that the Rademacher method converges absolutely in this case for $k < 0$. For $0 \leqslant k < 2$, convergence is much more subtle and is not absolute, but depends on the multiplier phases. For $k \geqslant 2$, the Rademacher method diverges. We will restrict ourselves to proving convergence for $k < 0$. The arguments going into the proof are standard, see e.g. [35].

To prove convergence, we need to show that

$$\left| \int_{\Gamma_n} \mathrm{d}\tau \, f(\tau) - \int_{\Gamma_\infty} \mathrm{d}\tau \, f(\tau) \right| \longrightarrow 0, \tag{B.3}$$

where $\Gamma_n$ is the $n$-th Rademacher contour as described in Section 3.2. There are two statements here, namely that the integral over $\Gamma_\infty$ is finite and that it is close to the integral over $\Gamma_n$ for large $n$.

**Finiteness of the integral over $\Gamma_\infty$.** Let us consider a fixed fraction $\frac{a}{c}$ that appears in the $n$-th Farey sequence $F_n$. Let us bound the integral over $C_{a/c}$. To prepare for the argument below, it is useful to consider also the fraction $\frac{a'}{c'}$ that appears directly to the right of $\frac{a}{c}$ in the Farey sequence $F_n$.[10] The properties of the Farey sequence imply that

$$a'c - ac' = 1. \tag{B.4}$$

Thus we can choose as modular transformation

$$\gamma(\tau) = \frac{a\tau - a'}{c\tau - c'}, \tag{B.5}$$

for the circle $C_{a/c}$. We hence have

$$\left| \int_{C_{a/c}} \mathrm{d}\tau \, f(\tau) \right| = \left| \int_{\mathbb{R}+i} \frac{\mathrm{d}\tau}{(c\tau - c')^{2-k}} f(\tau) \right|. \tag{B.6}$$

Since $f(\tau)$ is a periodic function (up to phases), $f(\tau)$ is bounded for $\tau \in i + \mathbb{R}$ and we have $|f(\tau)| \leqslant C$ for some constant $C$ independent of $\frac{a}{c}$. Thus

$$\left| \int_{C_{a/c}} \mathrm{d}\tau \, f(\tau) \right| \leqslant C \int_{\mathbb{R}} \frac{\mathrm{d}x}{|c(i + x) - c'|^{2-k}} \tag{B.7}$$

$$= C \int_{\mathbb{R}} \frac{\mathrm{d}x}{\left((cx - c')^2 + c^2\right)^{\frac{2-k}{2}}} \tag{B.8}$$

---

[10]Obviously, $\frac{a_L}{c_L}$ does not exist when $\frac{a}{c}$ is the last term in $F_n$, but this does not matter for the argument.

$$= C\,\frac{\sqrt{\pi}\,\Gamma(\frac{1-k}{2})}{\Gamma(\frac{2-k}{2})}\,c^{k-2}\,. \tag{B.9}$$

Renaming the prefactor $C'$ we thus have

$$\left|\int_{\Gamma_\infty} d\tau\, f(\tau)\right| \leqslant C' \sum_{c=1}^{\infty} \sum_{\substack{1\leqslant a\leqslant c \\ (a,c)=1}} c^{k-2} < C' \sum_{c=1}^{\infty} c^{k-1} < \infty\,, \tag{B.10}$$

for $k < 0$. Thus the integral over $\Gamma_\infty$ is well-defined for $k < 0$.

**Convergence of $\Gamma_n$.** We next have to check that the integral over $\Gamma_n$ converges to the integral over $\Gamma_\infty$ for $n \to \infty$. Consider again two consecutive fractions $\frac{a}{c} < \frac{a'}{c'}$ in the Farey sequence. Consider the contact point of $C_{a/c}$ with the circle to the right $C_{a'/c'}$. It is given by

$$\tau_{a/c,a'/c'} = \frac{ac+a'c'+i}{c^2+(c')^2}\,. \tag{B.11}$$

In the contour $\Gamma_n$, the integral around $C_{a/c}$ stopt as this contact point. Let us again take the modular transformation

$$\gamma(\tau) = \frac{a\tau - a'}{c\tau - c'}\,. \tag{B.12}$$

After mapping to the horizontal contour, the contour is restricted to $\operatorname{Re}\tau \geqslant 0$. Hence the difference in the integral around the full circle and the integral only up to the contact point with the circle to the right is given by

$$\int_{i+\mathbb{R}_{\leqslant 0}} \frac{d\tau}{(c\tau - c')^{2-k}}\, f(\tau) \tag{B.13}$$

times a potential multiplier phase. We now bound this in absolute value. We have

$$\left|\int_{i+\mathbb{R}_{\leqslant 0}} \frac{d\tau}{(c\tau - c')^{2-k}}\, f(\tau)\right| \leqslant \int_0^\infty \frac{dx}{|c(i-x)-c'|^{2-k}}\,|f(i-x)| \tag{B.14}$$

$$\leqslant C \int_0^\infty \frac{dx}{\left[c^2+(cx+c')^2\right]^{1-\frac{k}{2}}} \tag{B.15}$$

$$\leqslant C \int_0^\infty \frac{dx}{\left[c^2+(c')^2+(cx)^2\right]^{1-\frac{k}{2}}} \tag{B.16}$$

$$= \frac{C\sqrt{\pi}\,\Gamma(\frac{1-k}{2})}{2\,\Gamma(\frac{2-k}{2})} \times \frac{1}{c}\,(c^2+(c')^2)^{\frac{k-1}{2}}\,. \tag{B.17}$$

As above, we will call the prefactor $C'$. $c$ and $c'$ are the denominators of two consecutive elements in $\mathcal{F}_n$. We thus have

$$\left|\sum_{\frac{a}{c}\in F_n} \int_{C_{a/c}} d\tau\, f(\tau) - \int_{\Gamma_n} d\tau\, f(\tau)\right| < C' \sum_{\substack{(c,c')\text{ neighboring} \\ \text{denominators in } F_n}} \frac{1}{c(c^2+c'^2)^{\frac{1-k}{2}}}\,. \tag{B.18}$$

We sum here over both orderings, i.e. for $\frac{a}{c}$, $\frac{a'}{c'}$ two consecutive terms in the Farey sequence, we include both $(c,c')$ and $(c',c)$ in the above sum over neighboring denominators. One of

them captures the missing right arc of the Ford circle, while the other one captures by a similar argument the missing left arc. We now claim that for neighboring denominators in $F_n$ we have $c + c' > n$. This simply follows because if $\frac{a}{c} < \frac{a'}{c'}$ are two fractions in the Farey sequence with $c + c' \leqslant n$, then $\frac{a+a'}{c+c'}$ is also a valid fraction with denominator at most $n$ in between $\frac{a}{c}$ and $\frac{a'}{c'}$. But this means that two fractions $\frac{a}{c} < \frac{a'}{c'}$ with $c + c' \leqslant n$ cannot be neighbors and hence $c + c' > n$. This means that

$$
(c^2 + c'^2)^{\frac{1-k}{2}} > \left( \frac{n^2}{2} \right)^{\frac{1-k}{2}}. \tag{B.19}
$$

Thus we have

$$
\left| \sum_{a/c \in F_n} \int_{C_{a/c}} d\tau \, f(\tau) - \int_{\Gamma_n} d\tau \, f(\tau) \right| < C'' n^{k-1} \sum_{c \text{ denominator in } F_n} \frac{1}{c}. \tag{B.20}
$$

For every choice of denominator $c$ in $F_n$, there are $\phi(c)$ many choices of numerators, where $\phi(c)$ is the Euler totient function. Since $\phi(c) < c$, we can trivially bound

$$
\sum_{c \text{ denominator in } F_n} \frac{1}{c} < \sum_{c=1}^{n} \frac{1}{c} \times c = n. \tag{B.21}
$$

We then conclude

$$
\left| \sum_{a/c \in F_n} \int_{C_{a/c}} d\tau \, f(\tau) - \int_{\Gamma_n} d\tau \, f(\tau) \right| < C'' n^k, \tag{B.22}
$$

for a constant $C''$. We finish by noting that

$$
\left| \int_{\Gamma_\infty} d\tau \, f(\tau) - \int_{\Gamma_n} d\tau \, f(\tau) \right| < C'' n^k + \left| \int_{\Gamma_\infty} d\tau \, f(\tau) - \sum_{a/c \in F_n} \int_{C_{a/c}} d\tau \, f(\tau) \right| \tag{B.23}
$$

$$
\leqslant C'' n^k + \sum_{c=n+1}^{\infty} \sum_{\substack{1 \leqslant a \leqslant c \\ (a,c)=1}} \left| \int_{C_{a/c}} d\tau \, f(\tau) \right| \tag{B.24}
$$

$$
\leqslant C'' n^k + C' \sum_{c=n+1}^{\infty} c^{k-1} \tag{B.25}
$$

$$
< \left( C'' + \frac{C'}{|k|} \right) n^k \longrightarrow 0, \tag{B.26}
$$

as $n \to \infty$. This shows convergence of the Rademacher contour.

## B.2 The case of the four-point function

We now come to the case of interest, namely the planar four-point function

$$
\int d\tau \int \prod_{i=1}^{3} dz_i \prod_{i>j} \vartheta_1(z_{ij}, \tau)^{-s_{ij}}. \tag{B.27}
$$

We will not be able to rigorously show convergence of the Rademacher method in this case. However, we will rigorously show convergence of the following regularized expression

$$
\int d\tau \int \prod_{i=1}^{3} dz_i \, \eta(\tau)^{-\varepsilon} \prod_{i>j} \vartheta_1(z_{ij}, \tau)^{-s_{ij}}, \tag{B.28}
$$

for any $\varepsilon > 0$. We will then end up with a problem of commuting the limit with the infinite sum appearing in the Rademacher expansion. We will explain good heuristics why the sum and limit can be exchanged, but we were not able to establish this rigorously. The discussion in the non-planar case is again essentially identical.

**Convergence of the regulated integral.** Let us again consider the integral of (B.28) over the Ford circle $C_{a/c}$. As in the case without $z_i$-variables, we let $\frac{a'}{c'}$ be the next fraction in the $n$-th Farey sequence $F_n$ and use the modular transformation (B.12). The integral over $C_{a/c}$ is then given by the regulated version of (5.2),

$$A_{a/c}(\varepsilon) = i\rho_\varepsilon \begin{pmatrix} a & a' \\ c & c' \end{pmatrix} \int_{\mathbb{R}+i} \frac{\mathrm{d}\tau}{(c\tau)^{2+\varepsilon}} \, \eta(\tau + \tfrac{c'}{c})^{-\varepsilon}$$
$$\times \int \mathrm{d}z_1 \, \mathrm{d}z_2 \, \mathrm{d}z_3 \prod_{1 \leq i < j \leq 4} q^{-\frac{1}{2}s_{ij}c^2 z_{ij}^2} \vartheta_1(cz_{ij}\tau, \tau + \tfrac{c'}{c})^{-s_{ij}}. \quad \text{(B.29)}$$

$\rho_\varepsilon$ is a particular multiplier phase that appears from the modular transformation behaviour of the Dedekind eta-function. It is given in terms of Dedekind sums, but we will not need its precise form. Let us bound the integral over the $z_i$'s. As in Section 5, we write $z_i = \frac{n_i + \xi_i}{c}$ and divide the integral into different contributions. We have

$$\left| \int \mathrm{d}z_1 \, \mathrm{d}z_2 \, \mathrm{d}z_3 \prod_{1 \leq i < j \leq 4} q^{-\frac{1}{2}s_{ij}c^2 z_{ij}^2} \vartheta_1(cz_{ij}\tau, \tau + \tfrac{c'}{c})^{-s_{ij}} \right|$$
$$\leq \frac{1}{c^3} \sum_{n_1, n_2, n_3} \int \mathrm{d}\xi_1 \, \mathrm{d}\xi_2 \, \mathrm{d}\xi_3 \prod_{1 \leq i < j \leq 4} \left| q^{-\frac{1}{2}s_{ij}\xi_{ij}^2} \vartheta_1(\xi\tau - \tfrac{nc'}{c}, \tau + \tfrac{c'}{c})^{-s_{ij}} \right|. \quad \text{(B.30)}$$

The appearing theta function $\vartheta_1(z, \tau)$ is evaluated for $-1 \leq \operatorname{Im} z \leq 1$ and $\operatorname{Im} \tau = 1$. Since the theta function is periodic in $z \to z + 1$ and $\tau \to \tau + 1$, we can reduce this to the compact region $0 \leq \operatorname{Re} z \leq 1$, $-1 \leq \operatorname{Im} z \leq 1$, $\operatorname{Im} \tau = 1$ and $0 \leq \operatorname{Re} \tau \leq 1$. By continuity, the integrand is hence $\leq C(s, t)$ independent of $c$ and $c'$. The sum over $n_1$, $n_2$ and $n_3$ gives a factor of order $c^3$. Since also $\eta(\tau)^{-\varepsilon}$ is bounded on $\mathbb{R} + i$, we conclude that

$$\left| \eta(\tau + \tfrac{c'}{c})^{-\varepsilon} \int \mathrm{d}z_1 \, \mathrm{d}z_2 \, \mathrm{d}z_3 \prod_{1 \leq i < j \leq 4} q^{-\frac{1}{2}s_{ij}c^2 z_{ij}^2} \vartheta_1(cz_{ij}\tau, \tau + \tfrac{c'}{c})^{-s_{ij}} \right| \leq C'(s, t), \quad \text{(B.31)}$$

for some constant $C'(s, t)$ independent of $c$, $c'$ and $\varepsilon$.

At this point the analysis reduces back to the analysis in Section B.1. In particular, the same argument explained there carries over and shows that the Rademacher series converges for all $\varepsilon > 0$. The regulator is necessary in the argument to make the weight negative.

**Comments about the limit $\varepsilon \to 0$.** We have rigorously shown that the four-point function in the planar case is computed by

$$A^{\mathrm{p}} - \Delta A^{\mathrm{p}} = \lim_{\varepsilon \to 0} \sum_{c=1}^{\infty} \sum_{\substack{1 \leq a \leq \frac{c}{2} \\ (a,c)=1}} \sum_{\substack{n_{\mathrm{L}}, n_{\mathrm{D}}, n_{\mathrm{R}}, n_{\mathrm{U}} \geq 0 \\ n_{\mathrm{L}} + n_{\mathrm{D}} + n_{\mathrm{R}} + n_{\mathrm{U}} = c-1}} A_{a/c}^{n_{\mathrm{L}}, n_{\mathrm{D}}, n_{\mathrm{R}}, n_{\mathrm{U}}}(\varepsilon), \quad \text{(B.32)}$$

where $A_{a/c}^{n_{\mathrm{L}}, n_{\mathrm{D}}, n_{\mathrm{R}}, n_{\mathrm{U}}}(\varepsilon)$ is the analogue of eq. (3.25) that one derives by regulating the integrals with an additional factor of $\eta(\tau)^{-\varepsilon}$. In particular we have

$$\lim_{\varepsilon \to 0} A_{a/c}^{n_{\mathrm{L}}, n_{\mathrm{D}}, n_{\mathrm{R}}, n_{\mathrm{U}}}(\varepsilon) = A_{a/c}^{n_{\mathrm{L}}, n_{\mathrm{D}}, n_{\mathrm{R}}, n_{\mathrm{U}}}. \quad \text{(B.33)}$$

We would be done if we could commute the limit $\varepsilon \to 0$ with the infinite sum over $c$. But the example of the two-point function and the convergence discussed in Section 4 shows that this can potentially fail.

A sufficient criterion that allows us to commute the limit and the sum over $c$ is *absolute convergence* of the sum for $\varepsilon = 0$. This is the content of the dominated convergence theorem. We are hence done if we can show that

$$\sum_{c=1}^{\infty} \left| \sum_{\substack{1 \leqslant a \leqslant \frac{c}{2} \\ (a,c)=1}} \sum_{\substack{n_L, n_D, n_R, n_U \geqslant 0 \\ n_L + n_D + n_R + n_U = c-1}} A_{a/c}^{n_L, n_D, n_R, n_U} \right| < \infty\,. \tag{B.34}$$

If we naively estimate the internal sum, by taking all absolute values inside, it is of order $\frac{1}{c}$, which leads to a logarithmically divergent sum. Thus, one needs to be more careful. In order to deal with less special cases, we can restrict to $n_L > 0$ and $n_R > 0$, since the special cases $n_L = 0$ or $n_R = 0$ only leads to $\mathcal{O}(c^3)$ terms each of order $\mathcal{O}(c^{-5})$ and thus contributes an absolutely convergent part to the sum over $c$.

We will discuss now the case for $s < 1$, where we do not have to sum over $m_D$ and $m_U$. The case for higher $s$ is very similar. Consider now eq. (3.21) for $A_{a/c}^{n_L, n_D, n_R, n_U}$. We can take the sum inside the integral over $t_L$ and $t_R$ and only need to estimate the sum over phases

$$\sum_{\substack{1 \leqslant a \leqslant \frac{c}{2} \\ (a,c)=1}} \sum_{\substack{n_L, n_D, n_R, n_U \geqslant 0 \\ n_L + n_D + n_R + n_U = c-1}} e^{-\pi i \sum_{a=L,R, b=D,U} \left[ s \sum_{m=n_a+1}^{n_a+n_b} + t \sum_{m=n_b+1}^{n_a+n_b} \right] ((\frac{md}{c})) + 2\pi i t_L ((\frac{dn_L}{c})) + 2\pi i t_R ((\frac{dn_R}{c}))}\,. \tag{B.35}$$

We do not know how to estimate this sum rigorously. Standard methods for estimate exponential sums, such as the Van der Corput lemma fail, because the exponent is not a smooth function in the summation variables. However, heuristically, the appearing phases should be random for large enough values of $c$. Thus we most optimistically expect that

$$\left| \sum_{\substack{1 \leqslant a \leqslant \frac{c}{2} \\ (a,c)=1}} \sum_{\substack{n_L, n_D, n_R, n_U \geqslant 0 \\ n_L + n_D + n_R + n_U = c-1}} e^{-\pi i \sum_{a=L,R, b=D,U} \left[ s \sum_{m=n_a+1}^{n_a+n_b} + t \sum_{m=n_b+1}^{n_a+n_b} \right] ((\frac{md}{c})) + 2\pi i t_L ((\frac{dn_L}{c})) + 2\pi i t_R ((\frac{dn_R}{c}))} \right| \leqslant C(s,t,t_L,t_R) c^2\,, \tag{B.36}$$

since every sum only contributes a factor $\sqrt{c}$. For the following argument, even the much weaker estimate $c^{5-\varepsilon}$ for any $\varepsilon > 0$ would be good enough. This then also implies that

$$\left| \sum_{\substack{1 \leqslant a \leqslant \frac{c}{2} \\ (a,c)=1}} \sum_{\substack{n_L, n_D, n_R, n_U \geqslant 0 \\ n_L + n_D + n_R + n_U = c-1}} A_{a/c}^{n_L, n_D, n_R, n_U} \right| \leqslant \frac{C(s,t)}{c^3}\,, \tag{B.37}$$

for some function $C(s,t)$. This is good enough to ensure absolute convergence in eq. (B.34). We have hence made it at least very plausible that the Rademacher series indeed converges. We also checked convergence numerically, see Section 3.5.

## C  Number of solutions to quadratic equations mod prime powers

In this appendix, we discuss the number of solutions to the quadratic equation

$$an^2 + bn + c \equiv 0 \bmod p^k\,, \tag{C.1}$$

where $p^k$ is a prime power. Let us denote the number of solutions by $N(a,b,c,k)$. We need this information in Section 7 to evaluate the decay widths from the Rademacher expression

explicitly. Of course, the following discussion is very standard elementary number theory and can be found in essentially any book on number theory. We give the arguments here both to make the paper self-contained, but also because we have not found a complete explicit equation for $N(a, b, c, k)$ in the literature. The result of the discussion is the combination of eqs. (C.7) and (C.18) that give the result for general $p \neq 2$ and $p = 2$, respectively for $a \not\equiv 0 \bmod p$. Eq. (C.26) reduces the equation to the case $a \not\equiv 0 \bmod p$.

## C.1 Generic case

For the generic case, we assume that $p \geqslant 3$ and $a \not\equiv 0 \bmod p$. We will treat the other cases later. One can proceed by completing the square. We can multiply by $4a$, since this is a unit in the ring $\mathbb{Z}_{p^k}$ by assumption. We can then write the equation as

$$(2an + b)^2 + 4ac - b^2 = 0 \bmod p^k. \tag{C.2}$$

Let us denote by $D = b^2 - 4ac$ the discriminant of the polynomial. We hence learn that the number of solutions to the original equation equals the number of solutions to the equation

$$x^2 \equiv D \bmod p^k. \tag{C.3}$$

This can be solved via the theory of quadratic residues. In the simplest case where $D \not\equiv 0 \bmod p$, a quadratic residue mod $p^k$ exists precisely when it exists mod $p$. Hence the number of solutions is given by

$$D \not\equiv 0 \bmod p: \quad N(a, b, c, k) = \left(\frac{D}{p}\right) + 1, \tag{C.4}$$

where $\left(\frac{D}{p}\right)$ is the Legendre symbol. We gave the definition in Section 7.2 Let now $D = dp^\ell$ such that $d \not\equiv 0 \bmod p$. If $\ell \geqslant k$, we simply need to solve $x^2 \equiv 0 \bmod p^k$ and there are precisely $p^{\lfloor \frac{k}{2} \rfloor}$ solutions to this, since any solution is of the form $p^{\lceil \frac{k}{2} \rceil} a$. If $\ell < k$, we want to solve

$$x^2 \equiv dp^\ell \bmod p^k. \tag{C.5}$$

This implies that $x$ needs to have $p$ precisely $\frac{\ell}{2}$ times as a prime factor. Thus a solution will only exist when $\ell$ is even. If $\ell$ is even, we can set $x = p^{\frac{\ell}{2}} y$, which gives the equation

$$p^\ell(y^2 - d) \equiv 0 \bmod p^k. \tag{C.6}$$

This determines $y \bmod p^{k-\ell}$ and there are $p^{\frac{\ell}{2}}$ ways to lift it mod $p^{k-\frac{\ell}{2}}$, which then determines $x \bmod p^k$. Thus, we obtain the number of solutions mod $p^k$:

$$D = dp^\ell: \quad N(a, b, c, k) = \begin{cases} p^{\lfloor \frac{k}{2} \rfloor}, & k \leqslant \ell, \\ \delta_{\ell \in 2\mathbb{Z}} p^{\frac{\ell}{2}} \left(\left(\frac{d}{p}\right) + 1\right), & k > \ell. \end{cases} \tag{C.7}$$

When $D = 0$, we formally have $\ell = \infty$ and the number of solutions keeps growing indefinitely.

## C.2 The case $p = 2$

Let us consider the special case with $p = 2$. We still assume that $a \not\equiv 0 \bmod 2$, i.e. $a$ is odd. Here our previous method of completing the square does not work since it required multiplying by 4. Even after obtaining a solution mod 2 of the quadratic equation, we still would need to solve a linear equation.

Instead, we follow a more basic strategy. The following is essentially the logic of Hensel's lemma. For the base case with $k = 1$, we can simply try the two potential solutions. For $b$

even, we get precisely one solution because either $n = 0$ or $n = 1$ solves the equation. For $b$ odd and $c$ odd, we get no solution, while for $b$ odd and $c$ even, we get two solutions. Hence for $b$ odd and $c$ odd, we get no solutions mod $2^k$ for all $k$.

Let us first investigate the simpler case where $b$ odd and $c$ even, where we got two basic solutions. Assume that we have found all solutions mod $2^k$ and we want to find the solutions mod $2^{k+1}$. Let $n$ be a solution mod $2^k$ and lift it arbitrarily to an integer. Then a potential solution mod $2^{k+1}$ that reduces to $n$ has the form $n' = n + 2^k t$ for $t = 0$ or $t = 1$. Plugging this into the equation gives

$$a(n + 2^k t)^2 + b(n + 2^k t) + c \equiv an^2 + bn + c + 2^k bt \bmod 2^{k+1}. \tag{C.8}$$

Since $an^2 + bn + c \equiv 0 \bmod 2^k$, we can divide this equation by $2^k$ and conclude

$$bt \equiv \frac{an^2 + bn + c}{2^k} \bmod 2. \tag{C.9}$$

Since $b$ is odd, we can remove it on the left hand side and conclude that $t$ is uniquely fixed. This shows that any solution mod $2^k$ lifts uniquely to a solution mod $2^{k+1}$ and hence the number of solutions is independent of $k$. Thus there are always two solutions in this case. Thus we have found so far

$$b \text{ odd}: \quad N(a, b, c, k) = 2\delta_{c \in 2\mathbb{Z}}. \tag{C.10}$$

Next, we consider the case where $b$ is even. We can assume that $c$ is even as well, since otherwise we replace $n \to n + 1$ to reduce to this case. $n = 0$ is the unique solution mod 2. Now consider the equation mod 4. Any solution needs to be even and thus we have $n = 2t$ for some $t$. We have

$$a(2t)^2 + 2bt + c \equiv 0 \bmod 4. \tag{C.11}$$

Thus for the solution to survive, we need that $c$ is divisible by 4, in which case we obtain 2 solutions. We next consider the equation mod 8,

$$4at^2 + 2bt + c \equiv 0 \bmod 8. \tag{C.12}$$

We can divide both sides by 4 and obtain

$$at^2 + \frac{b}{2}t + \frac{c}{4} \equiv 0 \bmod 2, \tag{C.13}$$

where the fractions are well-defined by assumption. Now we can repeat the same procedure. We thus found a recursion relation for $b$ even and $k \geqslant 2$

$$b \text{ even}: \quad N(a, b, c, k) = 2 \begin{cases} N(a, \frac{b}{2}, \frac{c}{4}, k-2), & c \equiv 0 \bmod 4, \\ N(a, a + \frac{b}{2}, \frac{a+b+c}{4}, k-2), & a + b + c \equiv 0 \bmod 4, \\ 0, & \text{otherwise}, \end{cases} \tag{C.14}$$

where the second line follows from the first under the substitution $n \to n + 1$.

We can reformulate (C.10) and (C.14) in a better way as follows. First of all, the case distinction between $b$ odd and $b$ even can be rephrased as a case distinction between $D = b^2 - 4ac$ odd or even. If $b$ is odd, then $b^2 \equiv 1 \bmod 8$. In this case we have

$$D = 1 + 4c \bmod 8, \tag{C.15}$$

and thus $c = \frac{D-1}{4} \bmod 2$. This allows us to express (C.10) through the discriminant. One can also check that the condition $b$ odd, and $c \equiv 0 \bmod 4$ or $a + b + c \equiv 0 \bmod 4$ is equivalent to the condition $D \equiv 0, 4 \bmod 16$ and the recursion relation divides the discriminant by 4. This shows

that the number of solutions does in fact only depend on the discriminant of the polynomial. Let us denote this number as $N(D, k)$. Let us write $D = 2^\ell d$ as for the other primes. For $k \leqslant \ell$, we apply the recursion relation. We apply it $\lfloor \frac{k}{2} \rfloor$ times and obtain for $k \leqslant \ell - 2$

$$n(2^\ell d, k) = 2^{\lfloor \frac{k}{2} \rfloor}, \tag{C.16}$$

just as in the case for a generic prime number. For $k = \ell - 1$ and $k = \ell$ one has to be more careful, since one has to ensure that one can apply the recursion relations in the last two steps. This leads to the condition that $\ell$ is even and for $k = \ell$ to the additional condition $d \equiv 1 \bmod 4$. For $k > \ell$, we need that $\ell$ is even in order to get a solution, since we need to end up with an odd discriminant after applying the recursion a number of times. We then have

$$n(2^\ell d, k) = 2^{\frac{\ell}{2}} n(d, k - \ell) = 2^{\frac{\ell}{2}} \delta_{d \equiv 1 \bmod 8}. \tag{C.17}$$

To summarize the whole discussion,

$$D = 2^\ell d: \qquad N(a, b, c, k) = \begin{cases} 2^{\lfloor \frac{k}{2} \rfloor}, & k \leqslant \ell - 2, \\ \delta_{\ell \in 2\mathbb{Z}} 2^{\frac{\ell-2}{2}}, & k = \ell - 1, \\ \delta_{\ell \in 2\mathbb{Z}} 2^{\frac{\ell}{2}} \delta_{d \equiv 1 \bmod 4}, & k = \ell, \\ \delta_{\ell \in 2\mathbb{Z}} 2^{\frac{\ell}{2}} 2 \delta_{d \equiv 1 \bmod 8}, & k > \ell. \end{cases} \tag{C.18}$$

In particular, the number of solutions stabilizes for large $k$, except when $D = 0$.

## C.3 The case for $a \equiv 0 \bmod p$

Finally, we have to deal with the case where $a$ is not a unit in the ring $\mathbb{Z}_{p^k}$. Let us write $a = p^m \alpha$ for $\alpha \not\equiv 0 \bmod p$.

Let us first dispense of the trivial case where $\alpha = 0$. In this case, we are actually solving a linear equation,

$$bn + c \equiv 0 \bmod p^k. \tag{C.19}$$

The number of solutions is easy to determine. Write $b = p^r \beta$, $c = p^s \gamma$. Then

$$N(0, p^r \beta, p^s \gamma, k) = \begin{cases} p^k, & k \leqslant r, s, \\ p^r, & r \leqslant k, s, \\ 0, & s < k, r. \end{cases} \tag{C.20}$$

Now let us suppose that $\alpha \neq 0$. In some cases, we can immediately reduce to known cases. If $m \leqslant r, s$, we divide the equation by $p^m$ and see that

$$N(p^m \alpha, p^r \beta, p^s \gamma, k) = p^m N(\alpha, p^{r-m} \beta, p^{s-m} \gamma, k - m), \tag{C.21}$$

which we know how to treat. If $s < m, r$, there are no solutions for $k > s$. If remains to analyze the case $r \leqslant m, s$ for which we have

$$N(p^m \alpha, p^r \beta, p^s \gamma, k) = p^r N(\alpha p^{m-r}, \beta, p^{s-r} \gamma, k - m). \tag{C.22}$$

Thus we may assume that $b \not\equiv 0 \bmod p$.

For $k \leqslant m$, the quadratic term is not visible and thus there is a unique solution since $b$ is assumed to be invertible mod $p$. Let $n$ be the solution mod $p^m$. Then we try again to lift it to a solution mod $p^{m+1}$. Hence set $n' = n + p^m t$. Plugging this into the equation gives

$$an^2 + bn + c + bp^m t \equiv 0 \bmod p^{m+1}. \tag{C.23}$$

We can divide both sides by $p^m$ and obtain the equation

$$bt \equiv -\frac{an^2 + bn + c}{p^m} \mod p \,. \tag{C.24}$$

Hence we can lift the solution to a unique solution since $b \not\equiv 0 \mod p$. Thus for all $k$

$$N(p^m\alpha, \beta, p^s\gamma, k) = 1 \,. \tag{C.25}$$

Thus, we can summarize our analysis as follows:

$$N(p^m\alpha, p^r\beta, p^s\gamma, k) = \begin{cases} p^k, & k \leqslant m, r, s \,, \\ p^m N(\alpha, p^{r-m}\beta, p^{s-m}\gamma, k-m), & m \leqslant k, r, s \,, \\ p^r, & r \leqslant k, m, s \,, \\ 0, & s < m, r, k \,. \end{cases} \tag{C.26}$$

The equations (C.7), (C.18) and (C.26) now give a complete equation to the number of solutions.

# D  Numerical values of mass shifts and decay widths

In this Appendix, we give the real and imaginary parts of the double residues

$$\underset{s=s_*}{\mathrm{DRes}} A^{\mathrm{P}}(s, t) \,, \tag{D.1}$$

giving the averaged mass shifts and decay widths, as explained in Section 7. In practice, we truncated the sum over $c$ at $c = 1000$. This is more than enough to achieve the quoted accuracy. The real parts up to $s_* \leqslant 16$ are given by:

$$\mathrm{Re}\,\underset{s=1}{\mathrm{DRes}} A^{\mathrm{P}} = 8.36799 \cdot 10^{-5} \,, \tag{D.2a}$$

$$\mathrm{Re}\,\underset{s=2}{\mathrm{DRes}} A^{\mathrm{P}} = -1.61091 \cdot 10^{-4}(1+t) \,, \tag{D.2b}$$

$$\mathrm{Re}\,\underset{s=3}{\mathrm{DRes}} A^{\mathrm{P}} = -9.05359 \cdot 10^{-6}(1+t)(2+t) \,, \tag{D.2c}$$

$$\mathrm{Re}\,\underset{s=4}{\mathrm{DRes}} A^{\mathrm{P}} = -1.62145 \cdot 10^{-5}(1.01378+t)(2+t)(2.98622+t) \,, \tag{D.2d}$$

$$\mathrm{Re}\,\underset{s=5}{\mathrm{DRes}} A^{\mathrm{P}} = -4.18445 \cdot 10^{-6}(0.839876+t)(1.91206+t)(3.08794+t)$$
$$\times (4.16012+t) \,, \tag{D.2e}$$

$$\mathrm{Re}\,\underset{s=6}{\mathrm{DRes}} A^{\mathrm{P}} = -4.53218 \cdot 10^{-7}(1.22631+t)(3.+t)(4.77369+t)(9.24078+6t+t^2) \,, \tag{D.2f}$$

$$\mathrm{Re}\,\underset{s=7}{\mathrm{DRes}} A^{\mathrm{P}} = -1.0992 \cdot 10^{-7}(0.713797+t)(1.66827+t)(2.87132+t)(4.12868+t)$$
$$\times (5.33173+t)(6.2862+t) \,, \tag{D.2g}$$

$$\mathrm{Re}\,\underset{s=8}{\mathrm{DRes}} A^{\mathrm{P}} = -1.17212 \cdot 10^{-8}(1.11566+t)(2.35085+t)(4+t)(5.64915+t)$$
$$\times (6.88434+t)(16.6573+8t+t^2) \,, \tag{D.2h}$$

$$\mathrm{Re}\,\underset{s=9}{\mathrm{DRes}} A^{\mathrm{P}} = -1.70412 \cdot 10^{-9}(0.802207+t)(1.72678+t)(2.77343+t)(3.89584+t)$$
$$\times (5.10416+t)(6.22657+t)(7.27322+t)(8.19779+t) \,, \tag{D.2i}$$

$$\mathrm{Re}\,\underset{s=10}{\mathrm{DRes}} A^{\mathrm{P}} = -1.46141 \cdot 10^{-10}(0.980373+t)(1.99513+t)(2.94495+t)(3.71252+t)$$
$$\times (5+t)(6.28748+t)(7.05505+t)(8.00487+t)(9.01963+t) \,, \tag{D.2j}$$

$$\operatorname{Re}_{s=11}\operatorname{DRes}A^{\mathrm{p}} = -1.57835\cdot 10^{-11}(0.97632+t)(1.94161+t)(3.08953+t)(4.58696+t)$$
$$\times (6.41304+t)(7.91047+t)(9.05839+t)(10.0237+t)$$
$$\times (32.5311+11t+t^2), \tag{D.2k}$$

$$\operatorname{Re}_{s=12}\operatorname{DRes}A^{\mathrm{p}} = -1.31773\cdot 10^{-12}(1.21376+t)(3.22948+t)(4.71099+t)(6+t)$$
$$\times (7.28901+t)(8.77052+t)(10.7862+t)(4.74244+4.10917t+t^2)$$
$$\times (99.4325+19.8908t+t^2), \tag{D.2l}$$

$$\operatorname{Re}_{s=13}\operatorname{DRes}A^{\mathrm{p}} = -1.06181\cdot 10^{-13}(0.72252+t)(1.72957+t)(2.95485+t)(4.27537+t)$$
$$\times (5.69148+t)(7.30852+t)(8.72463+t)(10.0452+t)$$
$$\times (11.2704+t)(12.2775+t)(47.6506+13t+t^2), \tag{D.2m}$$

$$\operatorname{Re}_{s=14}\operatorname{DRes}A^{\mathrm{p}} = -7.63715\cdot 10^{-15}(1.10103+t)(2.37048+t)(4.18465+t)(5.61478+t)$$
$$\times (7+t)(8.38522+t)(9.81535+t)(11.6295+t)(12.899+t)$$
$$\times (10.5741+6.20568t+t^2)(119.695+21.7943t+t^2), \tag{D.2n}$$

$$\operatorname{Re}_{s=15}\operatorname{DRes}A^{\mathrm{p}} = -5.37019\cdot 10^{-16}(0.894687+t)(1.77142+t)(2.71904+t)(3.91241+t)$$
$$\times (5.38858+t)(6.8943+t)(8.1057+t)(9.61142+t)(11.0876+t)$$
$$\times (12.281+t)(13.2286+t)(14.1053+t)(62.6029+15t+t^2), \tag{D.2o}$$

$$\operatorname{Re}_{s=16}\operatorname{DRes}A^{\mathrm{p}} = -3.41776\cdot 10^{-17}(1.09904+t)(3.31603+t)(4.65774+t)(5.99127+t)$$
$$\times (6.86396+t)(8+t)(9.13604+t)(10.0087+t)(11.3423+t)$$
$$\times (12.684+t)(14.901+t)(5.91033+4.85514t+t^2)$$
$$\times (184.228+27.1449t+t^2). \tag{D.2p}$$

The exact values of the imaginary parts were already given in the ancillary file `decaywidths.txt` of [14] for $s_* \leqslant 14$. For completeness, here we list their numerical evaluations obtained using the Rademacher method:

$$\operatorname{Im}_{s=1}\operatorname{DRes}A^{\mathrm{p}} = 2.20304\cdot 10^{-2}, \tag{D.3a}$$

$$\operatorname{Im}_{s=2}\operatorname{DRes}A^{\mathrm{p}} = 2.21936\cdot 10^{-2}(1+t), \tag{D.3b}$$

$$\operatorname{Im}_{s=3}\operatorname{DRes}A^{\mathrm{p}} = 1.14933\cdot 10^{-2}(1+t)(2+t), \tag{D.3c}$$

$$\operatorname{Im}_{s=4}\operatorname{DRes}A^{\mathrm{p}} = 3.91563\cdot 10^{-3}(1.00014+t)(2+t)(2.99986+t), \tag{D.3d}$$

$$\operatorname{Im}_{s=5}\operatorname{DRes}A^{\mathrm{p}} = 9.83397\cdot 10^{-4}(0.999232+t)(1.99945+t)(3.00055+t)(4.00077+t), \tag{D.3e}$$

$$\operatorname{Im}_{s=6}\operatorname{DRes}A^{\mathrm{p}} = 1.98909\cdot 10^{-4}(1.00063+t)(1.99819+t)(3+t)(4.00181+t)$$
$$\times (4.99937+t), \tag{D.3f}$$

$$\operatorname{Im}_{s=7}\operatorname{DRes}A^{\mathrm{p}} = 3.32953\cdot 10^{-5}(1.00098+t)(2.00117+t)(2.99979+t)(4.00021+t)$$
$$\times (4.99883+t)(5.99902+t), \tag{D.3g}$$

$$\operatorname{Im}_{s=8}\operatorname{DRes}A^{\mathrm{p}} = 4.78321\cdot 10^{-6}(0.999514+t)(2.00019+t)(3.00246+t)(4+t)$$
$$\times (4.99754+t)(5.99981+t)(7.00049+t), \tag{D.3h}$$

$$\operatorname{Im}\operatorname*{DRes}_{s=9} A^{\mathrm{p}} = 5.99899 \cdot 10^{-7}(1.0012 + t)(1.99945 + t)(3.00007 + t)(4.00202 + t)$$
$$\times (4.99798 + t)(5.99993 + t)(7.00055 + t)(7.9988 + t), \tag{D.3i}$$

$$\operatorname{Im}\operatorname*{DRes}_{s=10} A^{\mathrm{p}} = 6.68737 \cdot 10^{-8}(1.00084 + t)(2.00162 + t)(3.0018 + t)(4.00346 + t)$$
$$\times (5 + t)(5.99654 + t)(6.9982 + t)(7.99838 + t)(8.99916 + t), \tag{D.3j}$$

$$\operatorname{Im}\operatorname*{DRes}_{s=11} A^{\mathrm{p}} = 6.70698 \cdot 10^{-9}(1.00015 + t)(2.00059 + t)(3.00102 + t)(3.99762 + t)$$
$$\times (4.99746 + t)(6.00254 + t)(7.00238 + t)(7.99898 + t)(8.99941 + t)$$
$$\times (9.99985 + t), \tag{D.3k}$$

$$\operatorname{Im}\operatorname*{DRes}_{s=12} A^{\mathrm{p}} = 6.11032 \cdot 10^{-10}(1.00082 + t)(1.99978 + t)(3.00215 + t)(4.00551 + t)$$
$$\times (4.99931 + t)(6 + t)(7.00069 + t)(7.99449 + t)(8.99785 + t)$$
$$\times (10.0002 + t)(10.9992 + t), \tag{D.3l}$$

$$\operatorname{Im}\operatorname*{DRes}_{s=13} A^{\mathrm{p}} = 5.10267 \cdot 10^{-11}(1.00106 + t)(2.00149 + t)(2.99981 + t)(4.00025 + t)$$
$$\times (4.99981 + t)(5.99798 + t)(7.00202 + t)(8.00019 + t)$$
$$\times (8.99975 + t)(10.0002 + t)(10.9985 + t)(11.9989 + t), \tag{D.3m}$$

$$\operatorname{Im}\operatorname*{DRes}_{s=14} A^{\mathrm{p}} = 3.93214 \cdot 10^{-12}(1.00008 + t)(2.00104 + t)(3.00317 + t)(4.00097 + t)$$
$$\times (4.9995 + t)(6.00111 + t)(7 + t)(7.99889 + t)(9.0005 + t)$$
$$\times (9.99903 + t)(10.9968 + t)(11.999 + t)(12.9999 + t), \tag{D.3n}$$

$$\operatorname{Im}\operatorname*{DRes}_{s=15} A^{\mathrm{p}} = 2.8132 \cdot 10^{-13}(1.00085 + t)(1.99974 + t)(3.00035 + t)(4.00265 + t)$$
$$\times (4.9986 + t)(6.00371 + t)(7.00575 + t)(7.99425 + t)(8.99629 + t)$$
$$\times (10.0014 + t)(10.9973 + t)(11.9996 + t)(13.0003 + t)$$
$$\times (13.9991 + t), \tag{D.3o}$$

$$\operatorname{Im}\operatorname*{DRes}_{s=16} A^{\mathrm{p}} = 1.8781 \cdot 10^{-14}(1.00096 + t)(2.00162 + t)(3.00208 + t)(4.00441 + t)$$
$$\times (5.00036 + t)(5.9905 + t)(6.99625 + t)(8 + t)(9.00375 + t)$$
$$\times (10.0095 + t)(10.9996 + t)(11.9956 + t)(12.9979 + t)$$
$$\times (13.9984 + t)(14.999 + t). \tag{D.3p}$$

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
