# Peer review of "Evaluating one-loop string amplitudes"

_SciPost Physics, doi:SciPost Phys. 15, 119 (2023)_

## Round 1 · Referee Report · Eric D'Hoker (Referee 1) · 2023-6-6

Report
The paper provides a very detailed construction of the convergent sum for the annulus and Mobius one-loop open superstring amplitudes for gauge group SO(32). The general method of contour deformation used here follows Witten's general proposal for implementing an "i epsilon" prescription in string amplitudes, but also extends this proposal and renders it more concrete. While the imaginary part of the one-loop open string amplitude may be evaluated using alternative methods, its real part is calculated here for the first time, as far as I know. The integration of the open string modulus for the combined topologies is related to an integration over the Rademacher contour and is evaluated using some beautiful known methods of analytic number theory. The result is a concrete formula by which the amplitude can be evaluated numerically for real Mandelstam variables. This result provides novel insight into string amplitudes in the regime that was most difficult to access before this work, namely in an intermediate regime of energies^2 on the order of 1/alpha', away from both low and ultra-high energies.
My one comments is as follows. I am puzzled by the authors' statements concerning old results on the analytic continuation of the closed one-loop amplitude on page 22. Specifically, that the analytic continuation methods used then and there are difficult to extend beyond genus zero. Actually, if one allows the amplitudes to be considered as functions of complex momenta, then the integral representation for the one-loop four-point amplitude is convergent for all purely imaginary values of the Mandelstam variables and provides a perfectly fine starting point on which to build the analytic continuation. While the analytic continuation procedure may be difficult to implement (as the authors state) it was in fact implemented for the one-loop closed superstring four massless string amplitude precisely in those old papers. The procedure produced the imaginary part of the amplitude with four massless NS bosons, and explicit formulas for the decay widths of the massive string states that occur in the amplitude. What those old papers did not obtain is the real part of the amplitude.
In summary, the paper under consideration presents innovative results in the area of string amplitudes, it is well-written and detailed, and will be of lasting value. I strongly recommend it for publication in SciPost.
My one comments is as follows. I am puzzled by the authors' statements concerning old results on the analytic continuation of the closed one-loop amplitude on page 22. Specifically, that the analytic continuation methods used then and there are difficult to extend beyond genus zero. Actually, if one allows the amplitudes to be considered as functions of complex momenta, then the integral representation for the one-loop four-point amplitude is convergent for all purely imaginary values of the Mandelstam variables and provides a perfectly fine starting point on which to build the analytic continuation. While the analytic continuation procedure may be difficult to implement (as the authors state) it was in fact implemented for the one-loop closed superstring four massless string amplitude precisely in those old papers. The procedure produced the imaginary part of the amplitude with four massless NS bosons, and explicit formulas for the decay widths of the massive string states that occur in the amplitude. What those old papers did not obtain is the real part of the amplitude.
In summary, the paper under consideration presents innovative results in the area of string amplitudes, it is well-written and detailed, and will be of lasting value. I strongly recommend it for publication in SciPost.

---

## Round 1 · Referee Report · Carlo Angelantonj (Referee 2) · 2023-7-7

Report
First and foremost I would to apologise with the Authors for the delay in providing my report.
The manuscript deals with the evaluation of one-loop scattering amplitudes in the open sector of the type I superstring which is valid for any finite value of $\alpha '$ and not just in the low energy regime. The integration of the Riemann surfaces is typically ill-defined and requires a suitable implementation of the $i\epsilon$ prescription. To this end, The Authors suggest deform the integration contour a la Rademacher, both for planar and non-planar amplitudes. As a result, the essential singularity at $Im (\tau ) =0$ is not reached following vertical lines but horizontally which makes the evaluation consistent with causality and unitarity.
The Authors discuss the deformation of the contour and the steps required to perform the integration over the moduli parameter in great detail and apply it to the evaluation of two-point and four-point amplitudes, for which they provide (for the first time) concrete expressions which are valid at finite string tension, and which are amenable to direct numerical evaluation.
The paper is well written, technically correct, and provides new insights in the proper evaluation of open-string scattering amplitudes. I recommend it for publication on SciPost.
The manuscript deals with the evaluation of one-loop scattering amplitudes in the open sector of the type I superstring which is valid for any finite value of $\alpha '$ and not just in the low energy regime. The integration of the Riemann surfaces is typically ill-defined and requires a suitable implementation of the $i\epsilon$ prescription. To this end, The Authors suggest deform the integration contour a la Rademacher, both for planar and non-planar amplitudes. As a result, the essential singularity at $Im (\tau ) =0$ is not reached following vertical lines but horizontally which makes the evaluation consistent with causality and unitarity.
The Authors discuss the deformation of the contour and the steps required to perform the integration over the moduli parameter in great detail and apply it to the evaluation of two-point and four-point amplitudes, for which they provide (for the first time) concrete expressions which are valid at finite string tension, and which are amenable to direct numerical evaluation.
The paper is well written, technically correct, and provides new insights in the proper evaluation of open-string scattering amplitudes. I recommend it for publication on SciPost.

---

## Round 2 · Referee Report · Eric D'Hoker (Referee 1) · 2023-7-28

Report
I strongly recommend the paper for publication in its present form.

---

## Round 2 · Author Response

We would like to thank both referees for their valuable feedback. In the revised version, we describe previous results on computing the imaginary part of the one-loop amplitude more clearly in Sec. 3.1.

---

## Round 2 · List of Changes

In order to address the concern of Referee 1, we modified the text on p. 22 to more clearly explain previous contributions of Ref. [30-32]. The text now refers to specific results as follows:
"[...] This approach was used in the old literature on string amplitudes, see, e.g., [30–32], but is difficult to make practical beyond genus zero. The simple reason is that in order to perform analytic continuation, one needs an analytic expression to begin with and these are very hard to find for such an intricate object as a string scattering amplitude. However, such an analytic continuation was successfully carried out for the imaginary part of the one-loop amplitude in type II strings [32, Theorem 2] and heterotic strings [32, Theorem 4], and produced explicit expressions for the decay widths of massive string states [32, Theorem 5]."
"[...] This approach was used in the old literature on string amplitudes, see, e.g., [30–32], but is difficult to make practical beyond genus zero. The simple reason is that in order to perform analytic continuation, one needs an analytic expression to begin with and these are very hard to find for such an intricate object as a string scattering amplitude. However, such an analytic continuation was successfully carried out for the imaginary part of the one-loop amplitude in type II strings [32, Theorem 2] and heterotic strings [32, Theorem 4], and produced explicit expressions for the decay widths of massive string states [32, Theorem 5]."

---

## Editorial Decision

published